# Radical chemistry and ozone production at a UK coastal receptor site

Robert Woodward-Massey[1], Roberto Sommariva[2,4], Lisa K. Whalley[1,3], Danny R. Cryer[1], Trevor Ingham[1], William J. Bloss[4], Stephen M. Ball[2], Sam Cox[5,a], James D. Lee[6,7], Chris P. Reed[6,b], Leigh R. Crilley[4,c], Louisa J. Kramer[4,d], Brian J. Bandy[8], Grant L. Forster[9], Claire E. Reeves[8], Paul S. Monks[2], and Dwayne E. Heard[1]

[1]School of Chemistry, University of Leeds, Leeds, LS2 9JT, UK
[2]School of Chemistry, University of Leicester, University Road, Leicester, LE1 7RH, UK
[3]National Centre for Atmospheric Science, University of Leeds, Leeds, LS2 9JT, UK
[4]School of Geography, Earth and Environmental Sciences, University of Birmingham, Birmingham, B15 2TT, UK
[5]Research Software Engineering Team, University of Leicester, Leicester, LE1 7RH, UK
[6]Wolfson Atmospheric Chemistry Laboratories, Department of Chemistry, University of York, York, YO10 5DD, UK
[7]National Centre for Atmospheric Science, University of York, York, YO10 5DD, UK
[8]Centre for Ocean and Atmospheric Sciences, School of Environmental Sciences, University of East Anglia, Norwich, UK
[9]National Centre for Atmospheric Science, University of East Anglia, Norwich, NR4 7TJ, UK
[a]now at: Digital Research Service, University of Nottingham, Nottingham, NG7 2RD, UK
[b]now at: Faculty for Airborne Atmospheric Measurements, Cranfield University, Cranfield, MK43 0AL, UK
[c]now at: Department of Chemistry, York University, Toronto, M3J 1P3, Canada
[d]now at: Ricardo Energy & Environment, Harwell, Oxfordshire, OX11 0QR, UK

*Correspondence to*: Lisa K. Whalley (l.k.whalley@leeds.ac.uk) and Dwayne E. Heard (d.e.heard@leeds.ac.uk)

## Abstract

OH, HO$_2$, total and partially-speciated RO$_2$, and OH reactivity ($k'_{OH}$) were measured during the July 2015 ICOZA (Integrated Chemistry of OZone in the Atmosphere) project that took place at a coastal site in North Norfolk, UK. Maximum measured daily OH, HO$_2$, and total RO$_2$ radical concentrations were in the range 2.6–17 × 10$^6$, 0.75–4.2 × 10$^8$, and 2.3–8.0 × 10$^8$ molecule cm$^{-3}$, respectively. $k'_{OH}$ ranged from 1.7 to 17.6 s$^{-1}$ with a median value of 4.7 s$^{-1}$. ICOZA data were split by wind direction to assess differences in the radical chemistry between air that had passed over the North Sea (NW–SE sectors) or major urban conurbations such as London (SW sector). A box model using MCMv3.3.1 chemistry was in better agreement with the OH measurements, but it overpredicted HO$_2$ observations in NW–SE air in the afternoon by a factor of ~2–3, although slightly better agreement was found for HO$_2$ in SW air (factor of ~1.4–2.0 underprediction). The box model severely underpredicted total RO$_2$ observations in both NW–SE and SW air by factors of ~8–9 on average. Measured radical and $k'_{OH}$ levels and measurement-to-model ratios displayed strong dependences on NO mixing ratios with the results suggesting that peroxy radical chemistry is not well understood under high NO$_x$ conditions. The simultaneous measurement of OH, HO$_2$, total RO$_2$, and $k'_{OH}$ were used to derive experimental (i.e., observationally determined) budgets for all radical species as well as total RO$_x$ (= OH + HO$_2$ + RO$_2$). In NW–SE air, the RO$_x$ budget could be closed during the daytime within experimental uncertainty but the rate of OH destruction exceeded the rate of OH production, and the rate of HO$_2$ production greatly exceeded the rate of HO$_2$

destruction while the opposite was true for $RO_2$. In SW air, the $RO_x$ budget analysis indicated missing daytime $RO_x$ sources but the OH budget was balanced, and the same imbalances were found with the $HO_2$ and $RO_2$ budgets as in NW–SE air. For $HO_2$ and $RO_2$, the budget imbalances were most severe at high NO mixing ratios, and the best agreement between $HO_2$ and $RO_2$ rates of production and destruction rates was found when the $RO_2$ + NO rate constant was reduced by a factor of 5. A photostationary steady-state (PSS) calculation underpredicted daytime OH in NW–SE air by ~35%, whereas agreement (~15%) was found within instrumental uncertainty (~26% at $2\sigma$) in SW air. The rate of *in situ* ozone production ($P(O_x)$) was calculated from observations of $RO_x$, NO, and $NO_2$ and compared to that calculated from MCM-modelled radical concentrations. The MCM-calculated $P(O_x)$ significantly underpredicted the measurement-calculated $P(O_x)$ in the morning, and the degree of underprediction was found to scale with NO.

## 1. Introduction

The removal of trace gases in the troposphere is dominated by reactions with the hydroxyl radical (OH) during the daytime. At semi-polluted locations, OH formation is mainly initiated by the photolysis of ozone ($O_3$) and nitrous acid (HONO):

$$O_3 + h\nu \ (\lambda < 340 \ \text{nm}) \rightarrow O(^1D) + O_2 \tag{R1a}$$

$$O(^1D) + H_2O \rightarrow OH + OH \tag{R1b}$$

$$O(^1D) + M \rightarrow O(^3P) + M \tag{R1c}$$

$$HONO + h\nu \ (\lambda < 400 \ \text{nm}) \rightarrow OH + NO \tag{R2}$$

The OH oxidation of volatile organic compounds (VOCs) in the presence of oxygen results in the formation of organic peroxy radicals ($RO_2$), for example via H-atom abstraction:

$$OH + RH + O_2 \rightarrow RO_2 + H_2O \tag{R3}$$

or via addition to unsaturated VOCs. $RO_2$ radicals may also be formed from the photolysis of oxygenated VOCs (OVOCs). In the presence of NO, $RO_2$ radicals produce hydroperoxyl radicals ($HO_2$) and carbonyl species:

$$RO_2 + NO \rightarrow RO + NO_2 \tag{R4a}$$

$$RO + O_2 \rightarrow R'CHO + HO_2 \tag{R4b}$$

$HO_2$ also reacts with NO to reform OH:

$$HO_2 + NO \rightarrow OH + NO_2, \tag{R5}$$

thus completing the atmospheric reaction cycle known as the hydrogen oxide ($HO_x$ = OH + $HO_2$) cycle. $HO_2$ is also formed by formaldehyde (HCHO) photolysis and by reaction of OH with CO and HCHO. Non-photolytic sources of radicals include nitrate radical ($NO_3$) chemistry and the ozonolysis of alkenes. Chlorine atoms may also react with VOCs to generate $RO_2$ radicals. The subsequent photolysis of $NO_2$ formed in reactions (R4a) and (R5) results in the production of ozone in the troposphere:

$$NO_2 + h\nu \ (\lambda < 400 \ nm) \rightarrow O(^3P) + NO \tag{R6a}$$
$$O(^3P) + O_2 + M \rightarrow O_3 + M \tag{R6b}$$

The short lifetimes of OH, $HO_2$, and $RO_2$, collectively known as $RO_x$, make them ideal species to test our understanding of

80 tropospheric oxidation chemistry, particularly when measurements of OH reactivity (the inverse of the OH lifetime, $k'_{OH}$) are also available.

The marine boundary layer (MBL) accounts for a substantial fraction (71%) of the planetary boundary layer. Field measurements of OH and $HO_2$ radicals in the MBL have shown that in general, models are capable of simulating the observed concentrations to within ~30% or better. The majority of these studies were characterised by clean air masses with very low

to relatively low NO mixing ratios (Sommariva et al., 2004; Heard et al., 2006; Mao et al., 2009; Whalley et al., 2010; Beygi et al., 2011; Vaughan et al., 2012; Mallik et al., 2018), where observed OH and $HO_2$ concentrations were generally in the range ~3–8 × $10^6$ molecule $cm^{-3}$ and ~1–4 × $10^8$ molecule $cm^{-3}$, respectively. In the MBL, $HO_x$ radical production is normally dominated by the reaction of $O(^1D)$ with water vapour (reaction (R1)), but HCHO photolysis to $HO_2$ is often an important primary radical source (Ren et al., 2008; Stone et al., 2018). Similarly, owing to low primary VOC levels, OVOCs other than

HCHO can account for a significant proportion of OH reactivity (Sommariva et al., 2006; Mao et al., 2009; Whalley et al., 2010; Stone et al., 2018) , and their photolysis can also be important radical sources. $HO_x$ chemistry was shown to be sensitive to halogen chemistry in some studies (Bloss et al., 2005b; Whalley et al., 2010; Stone et al., 2018), particularly with regard to the partitioning between OH and $HO_2$ since BrO and IO radicals act to convert $HO_2$ to OH (Sommariva et al., 2006). Heterogeneous uptake of $HO_2$ on aerosols can be a significant $HO_x$ loss route under low NO conditions (Sommariva et al.,

2004; Sommariva et al., 2006; Whalley et al., 2010; Stone et al., 2018), but considerable uncertainty surrounds the treatment of heterogeneous processes such as the parameterisation of uptake coefficients ($\gamma_{HO2}$) (Song et al., 2020), which historically have sometimes been set to unrealistically high values to achieve measurement-model agreement.

To the authors' knowledge, there are no reported field campaigns in the MBL in which OH, $HO_2$, $RO_2$, and $k'_{OH}$ were all measured simultaneously. Similarly, there are only two studies (Novelli et al., 2014a; Mallik et al., 2018) in the MBL in which

OH measurements made by laser-induced fluorescence (LIF) used a technique that allows for the discrimination of OH measurement interferences (Mao et al., 2012). In this work, we present interference-free measurements of OH (Woodward-Massey et al., 2020) alongside $HO_2$, total and partially-speciated $RO_2$, and $k'_{OH}$ from a field campaign at a UK coastal receptor

site. This complete suite of $RO_x$ measurements allowed for more comprehensive testing of our understanding of MBL chemistry through comparisons to the predictions of a box model. The field campaign took place at a site subject to a variety of air mass types, at which previous field campaigns with (incomplete) $RO_x$ and/or $k'_{OH}$ measurements were conducted in 1994–1995 (Forberich et al., 1999; Grenfell et al., 1999; Penkett et al., 1999), 2002 (Fleming et al., 2006; Green et al., 2006), and 2004 (Smith, 2007; Lee et al., 2009b). Recently Tan et al. (2019) showed that, provided $RO_x$ observations are available along with simultaneous supporting measurements (trace gas mixing ratios, photolysis rates, etc.), experimental budgets can be derived for all measured radical species, previously done for OH only. Budget imbalances can be identified with such an approach, and would indicate problems with experimental input data, such as radical concentrations and rate constants, and/or an incorrect chemical mechanism. Here we adopt the approach of Tan et al. (2019) and show that large imbalances exist between experimental radical production and destruction rates, and suggest explanations for such differences. In addition, we use the measured radical data to calculate *in situ* ozone production rates, and compare these to those calculated from modelled radical concentrations.

## 2. The Integrated Chemistry of OZone in the Atmosphere (ICOZA) project

The ICOZA field campaign focussed on the chemistry surrounding the production of ozone, which is harmful to human health (Jerrett et al., 2009), damages vegetation (Krupa et al., 1998), and is a potent greenhouse gas (IPCC, 2023). The ICOZA campaign took place in June–July 2015 at the Weybourne Atmospheric Observatory (WAO), which is a Global Atmospheric Watch (GAW) regional station run by the University of East Anglia (UEA) on behalf of the National Centre for Atmospheric Science (https://weybourne.uea.ac.uk/). As shown in Figure S1 in the Supplementary Information, the site is located on the North Norfolk Coast, UK (52°57'02" N, 1°07'19"E), ~50 km NNW of Norwich and ~190 km NE of London. The site is situated 16 m above sea level and is surrounded by grass fields on three sides, with the fourth facing due north towards a gently-sloped pebble beach. The nearest major road is a rural road (A147) located ~800 m to the south.

As the site is situated on the North Sea coast, it is subject to clean air masses that have travelled over the North Sea and originating from polar regions, as well as more polluted air that has been influenced by emissions from major UK cities (e.g., London, Birmingham) ~12–24 h before arriving at the site (Lee et al., 2009b). Polluted continental air, containing aged (by up to 36 hours) anthropogenic emissions from continental Europe, may also be sampled (Lee et al., 2009b). In addition, the site is subject to emissions from local roads, as well as shipping influences (Cárdenas et al., 1998).

The campaign began on 1$^{st}$ July 2015, but radical measurements commenced on the afternoon of 29$^{th}$ June. The last radical observations were made during the early morning of 22$^{nd}$ July, before the campaign end date of 31$^{st}$ July.

### 2.1. Instrumentation

A list of the instrumentation involved in measurements of trace gases, aerosols, and photolysis frequencies during the ICOZA campaign is given in Table 1. Instruments sampled ambient air from a height of ~4 m from the roofs of individual shipping

containers (Universities of Leeds and Leicester), a van (Birmingham), and from either the roof (~5 m) of the main WAO building directly, or via a common glass manifold (glass, ~15 cm ID) located on a tower that reached ~10 m above the roof. Comparisons of $NO_x$ observations, measured using multiple instruments, indicated no significant heterogeneity in the air sampled from different positions of the site.

### 2.1.1. The Leeds ground-based FAGE instrument

OH, $HO_2$, and $RO_2$ radicals were measured using the fluorescence assay by gas expansion (FAGE) technique (Hard et al., 1984; Heard and Pilling, 2003). Only a brief description of the Leeds instrument is given here as it has been described in detail elsewhere (Creasey et al., 1997; Whalley et al., 2010; Whalley et al., 2013; Whalley et al., 2018; Woodward-Massey, 2018; Woodward-Massey et al., 2020) and was used to measure OH and $HO_2$ during a previous campaign at the WAO, namely the Tropospheric ORganic photoCHemistry experiment (TORCH) 2 campaign (Smith, 2007).

Ambient OH concentrations are measured using laser-induced fluorescence (LIF) spectroscopy. The inlet consists of a conical turret (4 cm length, 3.4 cm ID) with a 1.0 mm diameter pinhole through which ambient air is sampled at ~7 slm. The turret is mounted on top of a stainless steel fluorescence cell ($HO_x$ cell), which is held at ~1.5 Torr (~2 hPa) using a combination of a Roots blower (Leybold RUVAC WAU 10001) and a rotary pump (Leybold SOGEVAC SV2000). A wavelength tuneable solid-state laser (YAG pumped Ti Sapphire laser) with a pulse repetition frequency of 5 kHz is tuned to the OH $A^2\Sigma^+$ (v′ = 0) ← $X^2\Pi_{3/2}$(v″ = 0) electronic transition at $\lambda$ = 308 nm. Approximately 10–20 mW of laser light is supplied to the fluorescence cell using an optical fibre. OH fluorescence near 308 nm is detected with a micro-channel plate photomultiplier (MCP, Photek PMT325/Q/BI/G with 10 mm diameter photocathode), which is used together with a 50 ns gating unit (Photek GM10-50) and a 2 GHz 20 dB gain amplifier (Photek PA200-10). Due to failures of the MCP detectors used during ICOZA, channel photomultiplier (CPM, Perkin Elmer 993P) detectors were sometimes used for the $RO_x$ fluorescence cell (see below). Fluorescence signals from the MCP/CPM detectors are analysed using gated photon counting.

$HO_2$ is detected after conversion to OH by the addition of NO (BOC, 99.95%) delivered using a mass flow controller (MFC, MKS Instruments 1179A series). An advantage for ICOZA relative to TORCH 2 and other previous field campaigns is the addition of instrumental capability for observations of $RO_2$ radicals, using the $RO_x$LIF technique (Fuchs et al., 2008), as well as interference-free measurements of $HO_2$ (Fuchs et al., 2011; Whalley et al., 2013). The Leeds group first applied the $RO_x$LIF method to ambient $RO_2$ observations in London (Whalley et al., 2018) and has since deployed this approach in Beijing (Slater et al., 2020; Whalley et al., 2021). The $RO_x$LIF method relies on the reactions of $RO_2$ radicals with NO (BOC, 500 ppmv in $N_2$) and CO (BOC, 5% in $N_2$) in a flow tube held at ~30 Torr, which result in initial conversion of $RO_2$ to OH ($RO_2$ + NO → $HO_2$, $HO_2$ + NO → OH) and then to $HO_2$ (OH + CO → $HO_2$; very rapid conversion back to $HO_2$ results in minimal radical wall losses) that is finally detected as OH via addition of NO inside a second FAGE cell ($RO_x$ cell) that the $RO_2$ flow tube is coupled to. During fieldwork, the two FAGE cells are used to make sequential measurements of OH, $HO_2$, $HO_2^*$ ($HO_2$ plus an interference from $RO_2$ radicals derived from long-chain alkanes and alkene and aromatic species; see Whalley et al. (2013) for full details), and total $RO_2$ in the following data acquisition cycle: (1) the first cell ($HO_x$) measures OH while

simultaneously the second cell ($RO_x$) measures $HO_2^*$ (high NO flow, 50 sccm; $RO_2$ interference maximised), (2) the $HO_x$ cell measures $HO_2$ (low NO flow, 5 sccm; $RO_2$ interference minimised) while the $RO_x$ cell measures total $RO_2$. The $RO_x$LIF method allows for the speciation of total $RO_2$ into "complex" ($cRO_2$) and "simple" ($sRO_2$) $RO_2$ types (Whalley et al., 2013; Tan et al., 2017; Whalley et al., 2018). $cRO_2$ are those that readily convert to OH in $HO_2^*$ mode ($cRO_2 = HO_2^* - HO_2$; note that in other previous studies, $cRO_2$ have also been labelled as $RO_2^\#$ or $RO_2i$), and correspond to $RO_2$ radicals derived from alkenes, aromatics, and long-chain (> $C_3$) alkanes. $sRO_2$ concentrations are derived from the difference between total $RO_2$ and $cRO_2$ and correspond to $RO_2$ radicals derived from small-chain (< $C_4$) alkanes. For more details of the speciation of $sRO_2$ and $cRO_2$, the reader is referred to Whalley et al. (2013) and Whalley et al. (2018).

Background signals are normally obtained by scanning the laser wavelength to a position that is off-resonance from the OH transition line. In the case of OH, this yields the measurement referred to as OHwave (Mao et al., 2012). Alternatively, the OH background may be determined chemically, via addition of an OH scavenger (e.g., propane) prior to FAGE sampling, which results in an OH measurement known as OHchem (Mao et al., 2012). The recording of OHchem can be used to test for the presence of interferences in conventional OHwave detection. Prior to the ICOZA campaign, an inlet pre-injector (IPI) module (Novelli et al., 2014a) was constructed to facilitate OHchem measurements in the Leeds FAGE system (Woodward-Massey et al., 2020). The IPI module was first deployed for ambient measurements of OHchem during ICOZA. To test for interferences, two IPI sampling periods were conducted in the middle of the campaign, separated by a few days ($3^{rd}$–$8^{th}$ July and $12^{th}$–$16^{th}$ July). A comparison of OHwave (corrected for the small and well-characterised interference from $O(^1D) + H_2O$, with $O(^1D)$ deriving from laser photolysis of $O_3$ (Woodward-Massey et al., 2020)) and OHchem measurements yielded a slope (OHwave versus OHchem) of $1.16 \pm 0.06$, with the non-unity value suggesting the presence of a small unknown OHwave interference during ICOZA on the order of 10–20%, which is smaller than the overall measurement accuracy of 26% at $2\sigma$. The OH data presented in this work correspond to OHchem when such data were available, but OHwave otherwise, where all OHwave data have been corrected for the known interference from $O_3/H_2O$. No attempt has been made to correct the OHwave data for the presence of other unknown interferences, which must be considered as an additional uncertainty in our analyses.

The Leeds FAGE instrument was calibrated by supplying known radical concentrations to the instrument inlets. Radicals were delivered in an excess flow (~40 slm) of humidified synthetic air (BOC, BTCA 178) using a turbulent flow tube. OH and $HO_2$ were generated in a 1:1 ratio (Fuchs et al., 2011) by the photolysis of water vapour at 184.9 nm using a Hg(Ar) pen-ray lamp (LOT LSP035). For $RO_2$ calibrations, $CH_4$ (BOC, CP grade 99.5%) was added to form $HO_2$ and $CH_3O_2$ in a 1:1 ratio. To enable the calculation of radical concentrations, $N_2O$ (BOC, medical grade 98%) chemical actinometry (Edwards et al., 2003; Faloona et al., 2004) was performed before and after the campaign in order to determine the product of the lamp flux at 184.9 nm and the photolysis time in the flow tube. Multipoint calibrations were performed for all radical species at regular intervals during the campaign, approximately once per week. The calibration factors (i.e., sensitivities) obtained did vary somewhat due to instrumental issues, namely the need to switch between MCP/CPM detectors. One calibration factor was applied to periods in which an MCP was used, and one for periods in which a CPM was used, where both calibration factors were derived from the average sensitivities of multiple MCP or CPM calibrations. As a consequence of the detector changes,

limits of detection (LODs) also varied over the course of ICOZA, with campaign-median 5 min LODs ($\pm$ 1$\sigma$) of $(6.1 \pm 4.1) \times 10^5$, $(4.0 \pm 2.7) \times 10^6$, and $(5.0 \pm 1.2) \times 10^7$ molecule cm$^{-3}$ for OH, HO$_2$, and total RO$_2$, respectively, for a signal-to-noise ratio (SNR) of 2.

### 2.1.2. The Leeds OH reactivity instrument

The Leeds shipping container also housed an instrument used for the measurement of total OH reactivity, $k'_{OH}$. Full details may be found in Cryer (2016) and Stone et al. (2016) but the key features are described here. The instrument consists of an atmospheric pressure flow tube (85 cm length, 5 cm ID) coupled to an OH fluorescence cell, which was located on the roof of the Leeds container during the ICOZA field campaign. The low pressure in the fluorescence cell (~2 Torr) is provided by the same pumping system as the FAGE cells. The flow tube samples air (via ½" PFA tubing) from close to the FAGE inlets at ~16

slm using a vacuum pump (Agilent Technologies IDP-3 Dry Scroll Pump). The laser flash photolysis pump and probe technique is used here (Jeanneret et al., 2001; Sadanaga et al., 2004), which involves the 266 nm laser (Quantel USA CFR 200) photolysis (pump) of ambient O$_3$ to generate OH via the reaction of O($^1$D) with H$_2$O. The OH signal decay is then observed in real time by LIF (the probe). Fitting of the first-order exponential obtained yields $k'_{OH}$, after subtraction of the physical decay rate controlled by non-chemical losses of OH (e.g., diffusion). 308 nm probe light is generated using the laser

system described above. Previously, OH reactivity was measured at the WAO during TORCH 2 using a different method, the sliding injector technique with FAGE detection of OH (Ingham et al., 2009; Lee et al., 2009b).

### 2.1.3. Supporting measurements

Formaldehyde (HCHO) was measured using an LIF instrument developed in Leeds, full details of which may be found in Cryer (2016). The instrument is based on the design of Hottle et al. (2009) and uses a pulsed (300 kHz) tuneable fibre laser

(TFL3000, Novawave) to generate UV radiation at 353.370 nm, which excites the HCHO $5_{0,5} \leftarrow 5_{1,4}$ rotational transition of the 4 A$^1$A$_2$ $\leftarrow$ X$^1$A$_1$ vibronic band. As with FAGE, gas is sampled into a low-pressure detection cell (110–120 Torr), but broadband fluorescence is collected at red-shifted wavelengths ($\lambda \sim$ 390–550 nm). The fluorescence detected using a PMT (Sens-Tech P25PC photodetector module) and the signal recorded by gated photon counting (PMS400A, Becker and Hickl). The background is determined by moving the laser wavelength to an offline position ($\lambda$ = 353.360 nm). The compact HCHO

instrument is housed in a shock-insulated 19" rack inside a plastic case, which was situated in the main WAO building during ICOZA and sampled air through the common glass manifold at a height of ~15 m. HCHO was also measured using Hantzsch colourimetry (Nash, 1953), with reasonably good agreement between the two techniques as demonstrated by the fit [HCHO]$_{LIF}$ = 1.2 × [HCHO]$_{Hantzch}$ + 0.3 ppbv ($R^2$ = 0.77; data not shown).

   Photolysis frequencies ($J$) for a variety of species, including O$_3 \rightarrow$ O($^1$D), NO$_2$, H$_2$O$_2$, HONO, HNO$_3$, and HCHO, were

230 calculated using the actinic flux measured using a 2$\pi$ spectral radiometer and published absorption cross-sections and photodissociation quantum yields; $J$(O$^1$D) was also measured using a 2$\pi$ filter radiometer (Meteorologie Consult GmbH) (Bohn et al., 2008). A variety of other supporting instruments (Table 1) were brought to the WAO site. Observational data

were also provided by instruments permanently located at WAO (e.g., CO, NO$_x$, O$_3$, SO$_2$, VOCs, meteorological data; see Table 1).

## 2.2. Model description

In this work, radical concentrations were compared to the predictions of a zero-dimensional box model incorporating a kinetic and photochemical mechanism, the Master Chemical Mechanism (MCM, http://mcm.leeds.ac.uk/MCM)(Saunders et al., 1997; Jenkin et al., 2003; Saunders et al., 2003). The current version of the MCM was used, v3.3.1 (Jenkin et al., 2015). The MCM is a near-explicit chemical mechanism, which represents the oxidative degradation of methane and 142 primary emitted VOCs and incorporates ~17,000 reactions of ~6,700 closed shell and free radical species. A subset of the MCM with 4,258 species and 12,851 reactions was used instead of the full MCM, reflecting the suite of VOC measurements during ICOZA (Table 1), e.g., no measurements of $> C_6$ alkanes, and limited BVOC observations (discussed below).

The MCM model simulations were conducted using AtChem2 (https://github.com/AtChem/AtChem2; (Sommariva et al., 2020)). Three model scenarios were used for the interpretation of radical observations: MCM-base, MCM-carb, and MCM-hox. The base model, MCM-base, was constrained to all measured trace gases listed in Table 1, with the exception of radical species (including NO$_3$ radicals, due to limited measurement data for NO$_3$), and OH reactivity, Cl$_2$, HCHO, the sum of methyl vinyl ketone and methacrolein (MVK+MACR), xylenes, monoterpenes, and dimethyl sulphide (DMS). MCM-carb was additionally constrained to measured carbonyl species (HCHO, MVK+MACR), but was otherwise identical to the base model; MVK and MACR (both C$_4$H$_6$O, measured as a sum using proton transfer reaction–mass spectrometry (PTR–MS)) were assumed to be present in a 1:1 ratio. Similarly, MCM-hox was the same as the base model but was additionally constrained to FAGE-measured HO$_2$. In all simulations, the ratio of trimethylbenzene (TMB) isomers (i.e., C$_9$ aromatics, also indistinguishable by PTR–MS) was assumed to be 1:1:1. In all simulations, NO and NO$_2$ were constrained as separate species rather than as total NO$_x$.

Temperature, pressure, and RH were also constrained in the MCM models, along with spectral radiometer measurements of photolysis frequencies: O$_3 \rightarrow$ O($^1$D), NO$_2$, HONO, HNO$_3$, NO$_3$, HCHO, CHOCHO, CH$_3$CHO, CH$_3$COCH$_3$, CH$_3$NO$_3$, C$_2$H$_5$NO$_3$, 1-C$_3$H$_7$NO$_3$, 2-C$_3$H$_7$NO$_3$, and ClNO$_2$. For species with more than one photolytic decomposition channel, branching ratios were taken from the MCM, with the exception of CHOCHO (glyoxal, three channels) for which values were corrected with those used in the Tropospheric Ultraviolet and Visible (TUV) radiation model (Madronich, 1992). Photolysis frequencies that were not measured were calculated using the MCM parameterisation, scaled by a factor derived from measured and calculated $J$(NO$_2$) to account for cloud cover.

All measurement constraints were used at their original time resolution, as described in Sommariva et al. (2020). First-order physical losses of unmeasured, model-generated intermediates (e.g., unmeasured OVOCs, organic nitrates, peroxides, acids, and alcohols) through dry deposition were taken from Zhang et al. (2003), where an environment of deciduous trees and long grass/crops was assumed, representative of the immediate area around the WAO. The boundary layer height was estimated at 800 m and kept constant for the duration of the simulations. As examples, these constraints lead to deposition velocities of

~6.4, ~2.8, and ~2.3 cm s$^{-1}$, corresponding to first-order deposition lifetimes of ~4, ~10, and ~12 h, for HNO$_3$, H$_2$O$_2$, and HCHO, respectively. The lifetime of these model-generated secondary products was determined by their first-order loss rates of dry deposition, heterogeneous uptake (see below), and photolysis, and bimolecular reactions (e.g., with OH and Cl atoms).

In addition to dry deposition, physical losses to aerosols (i.e., heterogeneous uptake) were considered in all model scenarios, represented by the following first-order loss rate (Ravishankara, 1997):

$$k'_{loss} = \omega A \gamma / 4 , \hspace{4cm} \text{(E1)}$$

where $\omega$ is the mean molecular speed of the species being taken up, $A$ is the aerosol surface area measured using an aerodynamic particle sizer (APS, range: <0.5–20 μm), and $\gamma$ is the aerosol uptake coefficient. Heterogeneous uptake was considered for the following species: O$_3$, OH, HO$_2$, H$_2$O$_2$, HO$_2$NO$_2$, NO, NO$_2$, HONO, HNO$_3$, NO$_3$, N$_2$O$_5$, SO$_2$, SO$_3$, HCl, Cl, and ClNO$_2$. $\gamma_{HO2}$ was set to a constant value of 0.1 in all model scenarios, the same value used in analyses of the Clean air for London (ClearfLo) campaign (Whalley et al., 2018).

The model was run for 48 hours (spin-up time) then reinitialised with the values of all species at the end of this period and rerun for the whole campaign. This allowed radical species and other reactive intermediates to reach steady-state levels but prevented the build-up of secondary products. The model output data were averaged to 15 min for the comparisons featured in this work.

OH concentrations can also be calculated using a photostationary steady-state (PSS) approach, which uses field measured quantities only, providing a check on (1) the internal consistency of OH, HO$_2$, and $k'_{OH}$ observations and (2) whether the OH budget can be balanced using measured quantities. PSS OH was calculated using the following equation:

$$[OH]_{PSS} = P_{OH} / k'_{OH} , \hspace{4cm} \text{(E2)}$$

where $k'_{OH}$ is obtained directly from measured OH reactivity; $P_{OH}$ terms accounted for the photolysis of O$_3$ to O($^1$D) (R1a) and reaction with water vapour (R1b), photolysis of HONO (R2), reactions of HO$_2$ (R5) with NO and O$_3$, and alkene ozonolysis reactions:

$$P_{OH} = 2J(O^1D)[O_3]f + J(HONO)[HONO] + k_{HO2+NO}[HO_2][NO] + k_{HO2+O3}[HO_2][O_3] + \Sigma^i k_{O3+ALKi}[O_3][ALK_i]Y^{OH}_{ALKi} , \hspace{0.5cm} \text{(E3)}$$

where $f$ is the fraction of O($^1$D) atoms that react with H$_2$O to form OH (~10%), $J$(HONO) is the spectral radiometer-determined HONO photolysis rate, and the final term on the right-hand side accounts for the total OH formation from the ozonolysis of each measured alkene (ALK) $i$ with yield $Y^{OH}_{ALKi}$. Rate constants and yields were taken from MCMv3.3.1 (Saunders et al., 1997; Jenkin et al., 2003; Saunders et al., 2003; Bloss et al., 2005a; Jenkin et al., 2015).

### 2.3. Meteorological and chemical conditions encountered during ICOZA

The overall conditions encountered during the ICOZA campaign are summarised by the time series of meteorological (wind speed and direction, temperature, RH, photolysis frequencies) and chemical (mixing ratios of NO, $NO_2$, CO, HCHO, isoprene, MVK+MACR, $O_3$, HONO) parameters shown in Figure 1, which includes all available measurements at 15 min time resolution for the period 29th June – 22nd July 2015. As shown in Figure S2, the predominant wind sectors were W, SW, and S (i.e., ~180°–270°). In terms of air mass back-trajectories (Cryer, 2016), during ICOZA the WAO site was generally under the

influence of Atlantic air, which had been transported over the UK, likely encountering anthropogenic emissions from major conurbations (e.g., Birmingham, London, Leicester; Figure S1). However, there were some exceptions to this on certain days of the campaign. For example, at the start of the campaign on 1st July, air that had spent a considerable amount of time over northern mainland Europe was sampled, which coincided with a heatwave (temperature of up to 30°C) and an event where high mixing ratios of ozone were encountered. Similarly, 11th and 16th July were characterised by a strong European influence,

while on 9th July the site was subject to air masses originating from the North Sea.

During the ICOZA campaign, wind speeds were relatively strong, with a median of 5.5 m s$^{-1}$ and a maximum of 12.7 m s$^{-1}$, and tended to drop slightly in the morning. Temperatures generally increased through the day from ~15°C before sunrise to ~20°C in the late afternoon, with a campaign maximum of 29.8°C during the heatwave on 1st July. RH varied between ~40–90% and was strongly anticorrelated with temperature. Based on Figure S2, and given that the SW sector corresponds to air

that may have been transported over large urban areas (Figure S1), all ICOZA data (from 2nd July onwards) were split into two categories according to wind direction – SW winds (180°–270°), and all other winds (NW–SE, <165° and >285°) – as shown in Table S1 and Figure S3. It can be seen that temperatures were generally higher (and conversely RH lower) in SW air. In addition, increased cloud cover in SW air is evident from the slightly lower average values of $J(O^1D)$.

Overall, moderate levels of pollution were observed during the ICOZA campaign. For example, the campaign median NO

mixing ratio, for periods of overlap with FAGE radical observations, was 160 pptv with a maximum of 4650 pptv (15 min). NO generally peaked in the morning, with median values of ~500–1500 pptv at 08:00–10:00 Universal Time Coordinated (UTC = GMT = BST – 1), ~100–400 pptv in the afternoon, and <100 pptv at night (Figure S3). On average, NO mixing ratios were almost a factor of 2 higher in SW air than in NW–SE air (Table S1). $NO_2$ exhibited median and maximum levels of 2.2 and 10.4 ppbv, respectively, and followed an inverse diel profile to that of NO, peaking at night at ~3–4 ppbv with an afternoon

minimum of ~1–1.5 ppbv. Both NO and $NO_2$ exhibited significant short-term variability (Figure 1).

The highest ozone mixing ratios of ~110 ppbv were observed on 1st July (Figure 1), which as mentioned above, coincided with elevated temperatures. It should be noted that this day, although interesting as a case study, was not characteristic of the general chemical conditions (particularly ozone levels) of the ICOZA campaign and was thus omitted from the wind sector analysis discussed in this paper (this includes Fig. 4 and Figs. 6–14); VOC measurements were also not available at this time.

On average, ozone exhibits a classically-expected photochemical diel profile, with a minimum of ~25–30 ppbv around 06:00

UTC and a maximum of ~35–45 ppbv in the afternoon. Due to higher NO levels, $O_3$ mixing ratios were lower in SW air (Table S1).

The diel profile of HCHO for SW air is similar to ozone (Figure S3), which is typical for an environment where HCHO production is largely driven by the photochemical oxidation of VOCs (Ayers et al., 1997; Cryer, 2016), with a diel minimum
of ~800 pptv in the late morning and evening, and a maximum around 16:00 UTC in the range ~1000–1800 pptv. The diel profile of HCHO in NW–SE air is less pronounced, with lower mixing ratios indicating less integrated photochemical processing. The highest HCHO levels of 3990 pptv were observed during the late morning of 4[th] July, although unfortunately radical and other measurements are not available for this time, owing to instrumental issues caused by a power cut on the preceding night.

Levels of HONO were quite variable and reached a maximum of ~570 pptv during the night that followed the daytime ozone event discussed previously (1[st]–2[nd] July, Figure 1). In general, HONO mixing ratios tended to peak after sunset and midnight in NW–SE (~100 pptv) and SW air (~150 pptv), respectively. There is no obvious diel profile in CO measurements, and no clear difference between NW–SE and SW wind, with median levels of ~90–150 ppbv observed throughout the day but a few short-term spikes of up to ~420 ppbv. The flat diel profile observed for CO indicates that, for the most part, the WAO
site was not strongly impacted by fresh anthropogenic combustion emissions during the ICOZA campaign.

Isoprene levels were low during ICOZA, with a campaign median mixing ratio of 24 pptv and a maximum of 450 pptv. Diel profiles of isoprene were similar between NW–SE and SW air, and bear slight resemblance to that expected from biogenic emissions with a maximum of ~50 pptv in the afternoon/early evening. The isoprene oxidation products MVK and MACR, measured as a sum using PTR–MS, exhibited significantly higher levels of ~80–200 pptv in SW air compared to ~20–60 pptv
in NW–SE air. $PM_{2.5}$ levels exhibited no clear diel profile, with similar loadings between the two wind sector types.

## 2.4. Radical budget equations

Experimental budget analyses for OH, $HO_2$, and $RO_2$ as well as their sum, $RO_x$ were first described by Tan et al. (2019) for measurements made in the Pearl River Delta, China, although many previous studies have investigated the experimental budget
of OH only (e.g., Whalley et al., 2011). Given the short lifetimes of OH, $HO_2$, and $RO_2$ radicals (on the order of seconds to < 1 min), we can assume that their concentrations are in steady-state (Geyer et al., 2004) and hence expect their production and destruction rates to be equal at a location such as the WAO where incoming air is homogeneous. In this section, we describe the reactions involved in $RO_x$ initiation and termination as well as those that interconvert different $RO_x$ species (i.e., propagation). We then show how such reactions can be used to derive budget equations (i.e., production and destruction rates)
for all radical species. All reaction rate constants and branching ratios were taken from the Master Chemical Mechanism, MCMv3.3.1 (http://mcm.leeds.ac.uk/MCM/; (Jenkin et al., 2003; Jenkin et al., 2015)).

## 2.4.1. Budget for total $RO_x = OH + HO_2 + RO_2$

$RO_x$ production is driven by the photolysis of $O_3$ (R1), HONO (R2), and OVOCs (R7) as well as alkene ozonolysis reactions (R8):

$$OVOCs + hv \rightarrow HO_2, RO_2 + products \qquad (R7)$$

$$Alkenes + O_3 \rightarrow OH, HO_2, RO_2 + products \quad (R8)$$

Other photolabile radical reservoir species, such as $H_2O_2$, ROOH, $HNO_3$, and $RONO_2$ were not measured during ICOZA and therefore were not considered in $RO_x$ production. $H_2O_2$ and $CH_3OOH$, and some other peroxides have been measured at coastal locations. For example, mean concentrations of $H_2O_2$ and $CH_3OOH$ at Mace Head were 0.23-1.58 ppbv and 0.1-0.15 ppbv, respectively (Morgan and Jackson 2002). At coastal locations peroxide photolysis was shown to be a minor source of OH or $HO_2$ (via $CH_3O$) (Sommariva et al., 2004; 2006)). Specifically at Cape Grim, the rate of OH production from $CH_3OOH$ was less than 5% of the rate of production from $O(1D)+H_2O$ (Sommariva et al., 2004), and hence peroxides were not included in the radical budget analyses. The total $RO_x$ production rate may therefore be approximated using:

$$P_{ROx} = 2J(O^1D)[O_3]\,f + J(HONO)[HONO] + \Sigma^i(J(OVOC_i)[OVOC_i](Y^{HO2}_{OVOCi} + Y^{RO2}_{OVOCi}))$$
$$+ \Sigma^j(k^j_8[O_3][ALK_j](Y^{OH}_{ALKj} + Y^{HO2}_{ALKj} + Y^{RO2}_{ALKj})) \qquad\qquad (E4)$$

where $f$ is the fraction of $O(^1D)$ atoms that react with $H_2O$ to form OH, $Y^{HO2}_{OVOCi}$ and $Y^{RO2}_{OVOCi}$ are the $HO_2$ and $RO_2$ radical yields from the photolysis of OVOC $i$, respectively, and $Y^{OH}_{ALKj}$, $Y^{HO2}_{ALKj}$, and $Y^{RO2}_{ALKj}$ are the radical yields from the ozonolysis of alkene (ALK) $j$. Of the OVOCs measured during ICOZA, those included in equation (E4) were formaldehyde (that photolyses to form $HO_2$), acetaldehyde ($HO_2$ and $RO_2$), and acetone ($RO_2$). All measured alkenes were included in equation (E4). $RO_x$ termination is controlled by radical loss to $NO_x$ and the self- and cross-reactions of peroxy radicals:

$$OH + NO \rightarrow HONO \qquad\qquad\qquad\qquad\qquad\qquad\qquad\qquad\qquad\qquad (R9)$$
$$OH + NO_2 \rightarrow HNO_3 \qquad\qquad\qquad\qquad\qquad\qquad\qquad\qquad\qquad\qquad (R10)$$
$$RO_2 + NO \rightarrow RONO_2 \qquad\qquad\qquad\qquad\qquad\qquad\qquad\qquad\qquad\qquad (R11)$$
$$HO_2 + HO_2 \rightarrow H_2O_2 + O_2 \qquad\qquad\qquad\qquad\qquad\qquad\qquad\qquad\qquad (R12a)$$
$$HO_2 + HO_2 + H_2O \rightarrow H_2O_2 + H_2O + O_2 \qquad\qquad\qquad\qquad\qquad\qquad (R12b)$$
$$RO_2 + RO_2 \rightarrow products \qquad\qquad\qquad\qquad\qquad\qquad\qquad\qquad\qquad\qquad (R13)$$
$$RO_2 + HO_2 \rightarrow ROOH + O_2 \qquad\qquad\qquad\qquad\qquad\qquad\qquad\qquad\qquad (R14)$$

The total $RO_x$ destruction rate is thus given by:

$$D_{ROx} = (k_9[NO] + k_{10}[NO_2])[OH] + k_{11}[NO][RO_2] + 2(k_{12}[HO_2]^2 + k_{13}[RO_2]^2 + k_{14}[RO_2][HO_2]) \quad (E5)$$


In this work, $RO_2$ radicals are treated as a single species, with generalised rate constants taken from the MCMv3.3.1: at 298 K and 1 atm, $k_{11} = \beta \times 9.0 \times 10^{-12}$ cm$^3$ molecule$^{-1}$ s$^{-1}$, where $\beta$ is the RONO$_2$ yield which we have assumed to be a constant 5% for all $RO_2$ species (Orlando and Tyndall, 2012; Tan et al., 2019); $k_{13} = 3.5 \times 10^{-13}$ cm$^3$ molecule$^{-1}$ s$^{-1}$; and $k_{14} = 2.3 \times 10^{-11}$ cm$^3$ molecule$^{-1}$ s$^{-1}$.

In line with Tan et al. (2019), we do not explicitly consider equilibrium reactions of the type HO$_2$ + NO$_2$ ⇌ HO$_2$NO$_2$ and RO$_2$ + NO$_2$ ⇌ RO$_2$NO$_2$ (e.g, peroxyacetyl nitrate (PAN) formation and decomposition) in the budget analyses, and assume these processes result in no net gain or loss of the radical species. The reaction of acyl peroxy radical with NO$_2$ is the only way to form PAN in the MCM, and acyl peroxy constitutes 7-8% of the RO$_2$ pool. For typical temperatures of the campaign HO$_2$NO$_2$ and PAN (and other PANs) will be in equilibrium. Only at extremes in temperature would the equilibrium be skewed.

For example at Hudson Bay in the Arctic, the formation of HO$_2$NO$_2$ was identified as an important radical reservoir, reducing HOx concentrations during the day and enhancing them at night (Edwards et al, 2011).

## 2.4.2. Budget for OH

OH is formed by reactions (R1), (R2), and (R4) (primary production). In addition, there are secondary OH sources from radical
recycling reactions of HO$_2$:

HO$_2$ + NO → OH + NO$_2$ (R5)

HO$_2$ + O$_3$ → OH + 2O$_2$ (R15)

OH production rates are therefore given by:

$$P_{OH} = 2J(O^1D)[O_3]f + J(HONO)[HONO] + k_{HO2+NO}[HO_2][NO] + k_{HO2+O3}[HO_2][O_3] + \Sigma^i\, k_{O3+ALKi}[O_3][ALK_i]Y^{OH}{}_{ALKi} \quad (E3)$$

We do not consider the photolysis of hypohalous acids (HOX, e.g., HOI or HOBr) as a source of OH owing to the lack of IO
or BrO measurements during ICOZA needed to quantify this. However, given the levels of NO at WAO, we expect the HOX source to only be very minor compared to reaction (R11) owing to the absence of exposed macroalgae and thus low inputs of I and Br.

OH loss rates are obtained directly from measured [OH] and measured OH reactivity:

$D_{OH} = [OH]k'_{OH} \quad (E6)$

### 2.4.3. Budget for HO₂

As shown in Section 2.4.1, primary sources of $HO_2$ are OVOC photolysis (of HCHO and $CH_3CHO$, reaction (R3)) and alkene
ozonolysis (reaction (R4)). The reaction of OH with some VOCs can also lead to the prompt formation of $HO_2$ (e.g. from
isoprene and aromatics). The yield of $HO_2$ from OH oxidation of these species is explicitly contained in the MCM mechanism
and so prompt $HO_2$ formation is included. For the Beijing AIRPRO field campaign the formation of $HO_2$ from VOC + OH $\rightarrow$
$HO_2$ versus the formation of $RO_2$ from VOC+OH $\rightarrow$ RO2 was investigated (Whalley et al., 2021) – and $HO_2$ production was
significant, owing to the presence of VOCs like isoprene and aromatics in the Beijing in summer. Secondary $HO_2$ sources are
as follows:

$$OH + CO + O_2 \rightarrow HO_2 + CO_2 \tag{R16}$$

$$OH + HCHO \rightarrow HO_2 + CO + H_2O \tag{R17}$$

$$RO_2 + NO \rightarrow RO + NO_2 \tag{R4a}$$

$$RO + O_2 \rightarrow R'CHO + HO_2 \tag{R4b}$$

The total $HO_2$ production rate may therefore be calculated as:

$$P_{HO2} = \Sigma^i (J(OVOC_i)[OVOC_i] Y^{HO2}_{OVOCi}) + \Sigma^j (k^j_8[O_3][ALK_j] Y^{HO2}_{ALKj})$$
$$+ k_{16}[OH][CO] + k_{17}[OH][HCHO] + k_{4a}[RO_2][NO] \tag{E7}$$

where $k_4 = \alpha \times 9.0 \times 10^{-12}$ cm$^3$ molecule$^{-1}$ s$^{-1}$ at 298 K ($\alpha$ is the $HO_2$ yield and equal to 0.95).

$HO_2$ is lost through reactions R5, R12, R14, and R15 (we do not consider the reactions of IO or BrO for reasons given in
Section 2.4.2). Thus, the $HO_2$ destruction rate is given by:

$$D_{HO2} = (2k_8[HO_2] + k_{10}[RO_2] + k_{11}[NO] + k_{12}[O_3])[HO_2] \tag{E8}$$

### 2.4.4. Budget for RO₂

Analogous to $HO_2$, primary $RO_2$ sources are OVOC photolysis (of $CH_3CHO$ and $CH_3COCH_3$) and alkene ozonolysis. The
major secondary source of $RO_2$ radicals is the reaction of OH with VOCs and OVOCs:

$$OH + RH + O_2 \rightarrow RO_2 + H_2O \tag{R3}$$

The $RO_2$ production rate from reaction (R3) ($P^{sec}_{RO2}$) may be calculated using measured VOC and OVOC concentrations,
multiplied by their OH reaction rate constants and [OH] (i.e., $P^{sec}_{RO2} = [OH] \times \sum^i k_{OH + VOCi} [VOC_i]$). However, given we

showed that missing OH reactivity was significant, this method would underestimate $P^{sec.}_{RO2}$. Alternatively, we can calculate $P^{sec.}_{RO2}$ from measured OH reactivity, after corrections for the contributions of inorganic reactants (i.e., $NO_x$, CO, $SO_2$, etc.) and organics that do not produce $RO_2$ (i.e., HCHO):

$k'_{OH, corrected} = k'_{OH} - k'_{OH, inorganic} - k'_{OH, HCHO}$ (E9)

$P^{sec.}_{RO2} = [OH]k'_{OH, corrected}$ (E10)

The total $RO_2$ production rate is then calculated as:

$P_{RO2} = \Sigma^i(J(OVOC_i)[OVOC_i]Y^{RO2}_{OVOCi}) + \Sigma^j(k^j_8[O_3][ALK_j]Y^{RO2}_{ALKj}) + P^{sec.}_{RO2}$ (E11)

The reactions of the nitrate radical ($NO_3$) and chlorine atoms (Cl) with VOCs and OVOCs could also constitute a source of $RO_2$. $NO_3$ radical concentrations were measured during ICOZA but data coverage was poor; we have therefore omitted $NO_3$ radical reactions in our budget analyses. We note that this limitation should only impact the nighttime results. The impact of

Cl atom chemistry is discussed in Section 3.10.3.

RO$_2$ radicals are lost through reactions (R4a), (R11), (R13), and (R14). From these reactions, the total $RO_2$ destruction rate may be derived as:

$D_{RO2} = ((k_{4a} + k_{11})[NO] + 2k_{13}[RO_2] + k_{14}[HO_2])[RO_2]$ (E12)


# 3. Results

## 3.1. Radical and OH reactivity observations and comparison to model predictions

Figure 2 shows the full time series of OH, $HO_2$, and total $RO_2$ radical concentrations as well as OH reactivity (15 min means) observed during ICOZA, covering the period 29$^{th}$ June – 21$^{st}$ July 2015. Also shown are the MCM-base model results for all

radical species and $k'_{OH}$ for periods in which measurements of all key species used to constrain the model were available, and the PSS calculated OH concentrations. The radical observations follow their expected photochemical diel profiles, with maximum levels around solar noon (~12:00 UTC on cloud-free days) and low nighttime concentrations, approximately an order of magnitude smaller than during the daytime for OH and $HO_2$, and frequently scattered around zero. There was less of a day-night contrast for total $RO_2$, for which nighttime levels were almost always above the $RO_2$ LOD (~$5 \times 10^7$ molecule

cm$^{-3}$ (~2 pptv)). Unlike radical concentrations, OH reactivity does not appear to show any diel pattern, with a median value of 4.7 s$^{-1}$ but frequent spikes of up to ~10–15 s$^{-1}$ (range = 1.7–17.6 s$^{-1}$). OH reactivity values were much higher at the start of the

campaign (i.e., $1^{st}$–$2^{nd}$ July 2015), due to the aforementioned heatwave event that coincided with the transport of pollution from northern continental Europe (Cryer, 2016).

Daily maximum OH concentrations were in the range $2.6$–$17 \times 10^6$ molecule cm$^{-3}$ and $1.8$–$13 \times 10^6$ molecule cm$^{-3}$ for observations and PSS calculations, respectively, based on the 90$^{th}$ percentile of the daytime concentrations with daytime defined as $J(O^1D) > 5 \times 10^{-7}$ s$^{-1}$. The MCM-base modelled OH ($1.1$–$14 \times 10^6$ molecule cm$^{-3}$) is discussed in more detail below. Similarly, daily maximum observed HO$_2$ and total RO$_2$ levels were in the range $0.75$–$4.2 \times 10^8$ molecule cm$^{-3}$ and $2.3$–$8.0 \times 10^8$ molecule cm$^{-3}$, respectively, or $1.0$–$4.9 \times 10^8$ molecule cm$^{-3}$ and $0.53$–$2.8 \times 10^8$ molecule cm$^{-3}$ for MCM-base predictions. It is clear from these features that the PSS calculation can broadly capture the range in daily maximum OH levels, while the MCM-base model can generally reproduce peak HO$_2$ but significantly underpredicts midday total RO$_2$. Observed nighttime concentrations were on the order of $1$–$3 \times 10^5$, $2$–$3 \times 10^7$, and $1$–$2 \times 10^8$ molecule cm$^{-3}$ for OH, HO$_2$, and total RO$_2$, respectively (see Figure 2).

On shorter timescales it can be seen (Figure 2) that the level of agreement is more variable. For example, the PSS calculation tracks OH observations very tightly for extended periods, but severe underpredictions are often found around midday, with smaller but still significant underpredictions on some mornings. The MCM-base predicted OH levels generally follow changes in the measurements, but with a tendency towards overprediction during the daytime (median ~10%, see below). Similar to the PSS model capture of OH measurements, MCM-base modelled HO$_2$ concentrations show excellent agreement with measurements for much of the campaign (median daytime overprediction of ~3%). However, in contrast to the comparison between OH measurements and the PSS calculations, on other days the HO$_2$ observations were either under- or overpredicted, with roughly equal examples of each. For total RO$_2$ radicals, the level of agreement is poor (median daytime underprediction of ~80%), where, with a few exceptions (e.g., 14$^{th}$ July), the MCM-base model cannot reproduce temporal changes in RO$_2$ concentrations, and generally cannot capture their magnitudes with any reasonable degree of success, consistent with the discrepancy between the predicted and observed ranges in daily maxima. OH reactivity is almost always underpredicted (daytime median ~35%), with a few examples of short periods where the MCM-base model reactivity matches the observations.

Figure 3 shows the median diel profiles of observed and modelled radical concentrations and OH reactivity, split by wind direction as in Figure S3. All radicals display their characteristic photochemical diel profiles, peaking around midday (albeit with strong day-to-day variability), and their qualitative features (i.e., overall shapes) are generally well-captured by the various model schemes. Smaller but still significant (i.e., above the LOD for each species) concentrations were observed at night that are generally larger than the model predictions.

Measured diel profiles of OH concentrations were similar in NW–SE and SW air, reaching ~$2$–$4 \times 10^6$ molecule cm$^{-3}$, but with slight differences in the shape of their diel profiles. Overall, the models capture the observations reasonably well (i.e., generally within a factor of 2 during the daytime, although this is larger than the measurement uncertainty of 26% at 2$\sigma$), with the best agreement seen in SW air for the MCM-hox model (median difference ~15% during the daytime) and the PSS calculation (~20%). The PSS calculation underpredicts OH concentrations throughout the day in NW–SE air by ~35%, but

tracks the measurements very tightly in SW air with a slight tendency towards underprediction, suggesting. missing OH sources (see Equation (E2)). For OH, differences between the MCM-base and MCM-carb models are only minor (median difference NW–SE ~1%, SW ~2%), with a greater difference seen for the MCM-hox run.

Measured $HO_2$ levels and diel profiles were very similar between the two wind sector types, with peak levels of around ~1–1.5 × $10^8$ molecule $cm^{-3}$ in the afternoon and nighttime concentrations on the order of ~2 × $10^7$ molecule $cm^{-3}$. However, the MCM models predict very different behaviour in NW–SE and SW air. In NW–SE air, $HO_2$ levels are significantly underpredicted in the evening by a factor of ~2, but overpredicted by about a factor of ~2–3 in the afternoon. For SW air, $HO_2$ is still significantly underpredicted in the evening but agreement throughout the daytime is fairly reasonable (median ~1% difference between measured $HO_2$ and MCM-carb), with less substantial afternoon disagreement. In either wind sector type, there are strong differences, both positive and negative, between the MCM-base and MCM-carb models (range: −50% to +70%).

Total $RO_2$ observations reached similar maximum concentrations of ~5 × $10^8$ molecule $cm^{-3}$ in NW–SE and SW air, but exhibit different diel profiles. In NW–SE air, $RO_2$ levels peaked sharply just after midday, with concentrations of ~1–3 × $10^8$ molecule $cm^{-3}$ in the morning and late afternoon. In SW air, the profile is broader, with concentrations of ~2–4 × $10^8$ molecule $cm^{-3}$ sustained from mid-morning to the afternoon and maximum levels observed around 16:00 UTC. In contrast to OH and $HO_2$, the level of measurement/model agreement for total $RO_2$ is poor at all times of day, as might be expected based on their time series comparison (Figure 2). For example, in NW–SE air, the measurement/MCM-base model ratios range from ~2–5 in the afternoon to almost 40 in the early morning, with an average value of 8. Similar ratios are found in SW air, albeit with more substantial afternoon disagreement, with an average of 9. The models do capture the general shape of the diel profiles, not evident from the time series data in Figure 2, although, the models predict small secondary maxima in total $RO_2$ at night, which is not seen in the measurements; such behaviour was also found in London (Whalley et al., 2018). Constraining the model to the few measured carbonyls (MCM-carb) or $HO_2$ (MCM-hox) does little to improve the measurement-model agreement.

OH reactivity exhibits similar behaviour in the two wind sector types, with relatively flat diel profiles and levels of 3–6 $s^{-1}$. In NW–SE air, the model reactivity roughly tracks temporal changes in the measured reactivity (e.g., the afternoon decrease), but the reactivity is underpredicted by ~34% throughout the day (maximum ~49%). The contribution of model intermediates to model reactivity is ~35% on average, with an afternoon maximum of ~63%. In SW air, the measured OH reactivity profile is flatter but is also underpredicted throughout the day by ~37% on average (maximum ~46%). Model intermediates were less important than in NW–SE air, but accounted for a slightly greater proportion of model reactivity in the afternoon and evening of up to ~30–40%, compared to ~22% on average.

## 3.2. $RO_2$ speciation

The $RO_xLIF$ technique allows for "simple" ($sRO_2$) and "complex" ($cRO_2$) organic peroxy radicals to be measured separately, as discussed in Section 2.1.1. $RO_xLIF$ observations of speciated $RO_2$ radicals are compared to MCM-base model predictions

in Figure S4. On average, both observed and modelled sRO$_2$ account for ~60–100% of total RO$_2$ radicals. The overall levels of sRO$_2$ (~1–3 × 10$^8$ molecule cm$^{-3}$) and cRO$_2$ (~0–1.5 × 10$^8$ molecule cm$^{-3}$) are similar between each wind sector type. In NW–SE air, sRO$_2$ and cRO$_2$ display slightly different diel profile shapes, the latter being suppressed in the morning hours. sRO$_2$ are always significantly underpredicted by the model (average measurement/model ratio ~ 9), whereas there is agreement for cRO$_2$ around ~06:00 and ~18:00 UTC but disagreement overall (average ratio ~ 7). In SW air, both diel profiles are broader and the degree of underprediction in the afternoon is worse, with average values of ~10 and ~7 for sRO$_2$ and cRO$_2$, respectively. Similar to NW–SE air, agreement is also seen for cRO$_2$ in the early morning in SW air.

Figure S5 shows the daytime breakdown of RO$_2$ species predicted by the MCM-base model, split according to wind direction. The model predicts that the dominant species in both wind sector types was methylperoxy (CH$_3$O$_2$), with contributions of ~58% and ~55% (daytime median) in NW–SE and SW air, respectively. In NW–SE air, the next most important species is HYPROPO2 (CH$_2$(OH)CH(CH$_3$)O$_2$, formed from OH addition to propene) with a contribution of ~9%, followed by acetylperoxy (CH$_3$CO$_3$, ~7%), BUTDBO2 (CH$_2$(OH)CH(O$_2$)CH=CH$_2$, formed from OH addition to 1,3-butadiene, ~2%), and HOCH2CH2O2 (CH$_2$(OH)CH$_2$O$_2$, formed from OH addition to ethene, ~2%). Other RO$_2$ radicals contribute ~22% in total. In SW air, the contributions are fairly similar: HYPROPO2 ~6%, acetylperoxy ~8%, BUTDBO2 ~2%, and HOCH2CH2O2 ~2%. Other RO$_2$ radicals are slightly more important than in NW–SE air, with a total contribution of ~25%. Isoprene-derived peroxy radicals (with the most important being ISOPBO2 and ISOPDO2) contribute only ~2% and ~5% in NW–SE and SW air, respectively.

### 3.3. Observed and modelled OH versus $J(O^1D)$

The discussion of this can be found in the Supplementary Information.

### 3.4. Observed and modelled RO$_2$ versus HO$_2$

It can be seen from Figure 2 and Figure 3 that RO$_2$ observations are well correlated with HO$_2$. To assess the strength of this correlation, RO$_2$ is plotted against HO$_2$ in Figure 4 for both measurement and model results, with fit parameters summarised in Table S3. Observed RO$_2$ and HO$_2$ are indeed strongly correlated, with a stronger correlation in SW air ($R = 0.81$ versus $R = 0.63$ in NW–SE air). For NW–SE air, the correlation is much stronger for sRO$_2$ versus HO$_2$ ($R = 0.68$) than cRO$_2$ versus HO$_2$ ($R = 0.37$) (data not shown). The fit slopes suggest that in NW–SE air RO$_2$ and HO$_2$ coexisted in approximately a 1:1 ratio, while this was closer to 2:1 for SW air. The non-negligible intercepts of ~1–2 × 10$^8$ molecule cm$^{-3}$ suggest that there are some RO$_2$ sources that do not result in the concomitant production of HO$_2$, consistent with the time series data in Figure 2, which may be more relevant at night and possibly indicates a contribution from NO$_3$ chemistry. For the model results, the RO$_2$:HO$_2$ ratio was closer to 1:2 in both NW–SE and SW air during the daytime, but much higher (~12:1) during nighttime. The different slopes for day and nighttime data in the model cases are not seen in the observations. The increased slope for the model results during nighttime indicates slower RO$_2 \rightarrow$ HO$_2$ cycling due to lower NO levels compared to the daytime. However, small

amounts of RO$_2$ will be converted to HO$_2$, so a correlation still exists (either because there is a small amount of NO present or the RO$_2$+RO$_2$ self-reaction can form HO$_2$).

## 3.5 Observed and modelled OH, HO$_2$, RO$_2$, and $k'_{OH}$ versus NO

Radical levels are known to display a strong dependence on NO$_x$ concentrations since radical propagation is promoted by NO, and radical loss is often dominated by the reactions of radicals with NO and NO$_2$. In recent studies utilising the RO$_x$LIF

technique, it has become apparent that measurement-model ratios for RO$_2$ are particularly sensitive to NO (Tan et al., 2017; Tan et al., 2018; Whalley et al., 2018; Slater et al., 2020; Whalley et al., 2021), which has implications for the calculation of ozone production rates. The dependence of daytime ($J(O^1D) > 5 \times 10^{-7}$ s$^{-1}$) radical concentrations and OH reactivity values on NO mixing ratios is shown in Figure S7, split according to wind direction. For OH only, both measured and modelled concentrations were normalised to the campaign-average $J(O^1D)$ to remove the dependence on OH source strength (=

OH_Jnorm, e.g., (Tan et al., 2017)). This approach is justified by the almost linear dependence of OH on $J(O^1D)$ (Figure S6); similar trends were also found for un-normalised OH albeit with more scatter (data not shown). The corresponding measurement-model ratios for radical species are shown in Figure 5.

In NW–SE air, observed OH_Jnorm levels exhibit a classically-expected dependence on NO, increasing up to ~100 pptv NO before decreasing at higher NO (Figure S7). The MCM-base model reproduces the measured trend reasonably well.

However, the PSS model significantly underpredicts the observations at low NO, yielding measurement-model ratios of ~2–3 below ~200 pptv NO (Figure 5), which is greater than the estimated combined measurement-model uncertainty (~50%). In SW air, measured OH_Jnorm decreases with NO across the full NO range. The PSS model underpredicts the observations more severely at low NO (<300 pptv), yielding similar measurement-model ratios to those in NW–SE air. The MCM-base model slightly underpredicts the observations at low NO (by up to ~90%) but there is reasonable agreement (within ~40%) at

moderate to high NO (>300 pptv). The different behaviour for measured OH_Jnorm with respect to NO in NW–SE and SW air might be expected from the strong differences in scaling factors (A, Figures S6 and S2) in the analysis in Section 3.3, which reflect the dependence of OH on species such as NO$_x$ and VOCs.

Measured HO$_2$ in NW–SE air exhibits a weak decreasing trend with NO, with levels of ~0.7–1.3 × 10$^8$ molecule cm$^{-3}$ below ~1 ppbv NO and ~0.3 × 10$^8$ molecule cm$^{-3}$ above this threshold. In contrast, the model dependence on NO is much

stronger such that HO$_2$ levels are overpredicted by up to a factor of ~3 at low NO. In the highest NO bin, measured HO$_2$ is underpredicted by a factor of ~2. In SW air, both measured and modelled HO$_2$ decrease sharply with NO, from ~2 × 10$^8$ molecule cm$^{-3}$ at ~100 pptv NO to ~0.1–0.3 × 10$^8$ molecule cm$^{-3}$ above 1 ppbv. For this wind direction, the measurements and model results are in agreement across the full NO range.

Overall, measured OH and HO$_2$ are in reasonable agreement with the PSS calculation and the MCM-base model

prediction, respectively, at high NO. There is also good agreement between measured and MCM-base OH at moderate to high NO. However, RO$_2$ radicals are significantly underpredicted by the base model across all NO mixing ratios in both NW–SE and SW air. Observed and modelled RO$_2$ concentrations display a constant decrease with NO in either wind sector type.

Comparing the two sets of observations, the dependence is steeper in SW air. In both wind sector types, the model NO dependence of $RO_2$ is steeper than the corresponding measurement NO dependence, such that the measurement-model ratio increases from ~2–3 for NO < 100 pptv to ~10–30 for NO > 1 ppbv (Figure 5). Such discrepancies likely relate to the model underprediction of OH reactivity, the degree of which also scaled with NO (Figure S7), since this indicates missing $RO_2$ sources from the OH oxidation of missing VOCs. Missing OH reactivity, i.e., the difference between measured and modelled OH reactivity, reached values of ~2–3 $s^{-1}$ for NO > 1 ppbv, or ~30–45% of measured reactivity.

The increasing underprediction of $RO_2$ radicals as NO increases has been seen in all previous field campaigns in which $RO_2$ (distinct from $HO_2$) was measured using the $RO_xLIF$ technique (Fuchs et al., 2008; Whalley et al., 2013). $RO_2$ measurement-model ratios as a function of NO from these campaigns (Tan et al., 2017; Tan et al., 2018; Whalley et al., 2018; Slater et al., 2020; Whalley et al., 2021) are compared with ICOZA in Figure 6. It can be seen that the measurement-model discrepancy starts to appear at lower NO (i.e., <100 pptv) for ICOZA in comparison to the other campaigns, although the curves for ICOZA and AIRPRO summer display strong overlap in the ~100–600 pptv NO range. There is also some overlap between the curves for ICOZA and BEST-ONE (Tan et al., 2018), a winter campaign conducted at a suburban site near Beijing, at low/moderate NO (~100–200 pptv). Overall, the largest measurement-model ratios were found in London (Whalley et al., 2018) and central Beijing (Slater et al., 2020; Whalley et al., 2021), but at higher NO levels (>10 ppbv) than those seen in most other campaigns including ICOZA.

To further explore the $RO_2$ discrepancy found for ICOZA, the contribution of $sRO_2$ to total $RO_2$ is plotted as a function of NO for measurement and model results in Figure S8. For the measurements, the $sRO_2$ contribution increases with NO in both NW–SE and SW air from ~0.7 to values close to 1. In contrast, the model predicts a constant $sRO_2$ fraction of ~0.7, in accordance with the dominance of $CH_3O_2$ (Figure S5). The reasons for the strong dependence of the measured $sRO_2$ fraction on NO are unclear, but may be due to the NO-mediated propagation of $cRO_2$ to $sRO_2$ as VOCs are increasingly fragmented into smaller and less complex $RO_2$ species. Alternatively, $cRO_2$ formation may be facilitated by low $NO_x$ levels, e.g., due to autoxidation chemistry (Crounse et al., 2013; Jokinen et al., 2014; Bianchi et al., 2019).

## 3.6. Missing $k'_{OH}$ versus OVOCs and temperature

Missing OH reactivity has been found in many previous field studies in which OH reactivity was measured and compared to calculated reactivity or model simulations (Kovacs et al., 2003; Ren et al., 2003; Di Carlo et al., 2004; Sinha et al., 2008; Lee et al., 2009b; Lou et al., 2010; Mao et al., 2012; Nolscher et al., 2012; Edwards et al., 2013; Brune et al., 2016; Whalley et al., 2016; Kumar et al., 2018). Missing OH reactivity is normally attributed to either unmeasured primary VOCs (e.g., BVOCs), or unmeasured VOC oxidation products (i.e., OVOCs). To test which was responsible for the missing reactivity observed for ICOZA, missing OH reactivity (measured – modelled) was binned against various chemical concentrations and temperature. These data are shown in Figure S9. It can be seen that missing reactivity exhibits strong correlations ($R^2 \geq 0.83$) with several measured OVOCs, such as acetaldehyde, acetone, and methanol (all constrained in MCM-base). This finding suggests that the missing reactivity is due to unmeasured VOC oxidation products that were not well simulated by the base model. The only

OVOCs measured and constrained in the base model were acetone, acetaldehyde, and methanol and as such many OVOCs were missing, e.g., the oxidation products of >$C_2$ VOCs. Weaker correlations ($R^2 \leq 0.7$) were found for isoprene (maximum = 418 pptv) and the PTR-MS measured sum of monoterpenes (maximum = 105 pptv), such that unmeasured primary BVOCs are unlikely to be the root of the missing reactivity. BVOC emissions are known to display an exponential dependence on temperature (Guenther et al., 1993). It is therefore expected that missing reactivity should scale exponentially with temperature if missing biogenic species are responsible (Di Carlo et al., 2004). As shown in Figure S9, this was not the case for ICOZA and the dependence is clearly linear, albeit over a relatively small temperature range of ~12–24°C. This is further evidence that the missing reactivity for ICOZA is due to OVOCs, not a primary biogenic species. It is hypothesised that the correlation with temperature is due to increased VOC oxidation rates at high temperature that results in greater OVOC production. Missing reactivity is also reasonably well correlated with toluene ($R^2 = 0.84$, data not shown), such that unmeasured aromatic VOCs could also be responsible, as suggested by Lee et al. (2009b).

When missing OH reactivity is calculated using the MCM-carb model, which is additionally constrained to measured HCHO and MVK+MACR, all the correlations in Figure S9 remain ($R^2 \geq 0.83$), with the exception of temperature ($R^2 = 0.61$). Therefore, species other than HCHO and MVK+MACR must be responsible for the missing OH reactivity. In recent years, OVOC emissions have increased in importance in the UK, with ethanol now the largest contributor to non-methane VOCs in terms of mass emissions (Lewis et al., 2020). More generally, alcohols are now the largest contributors to ozone production (~30%) in terms of their photochemical ozone creation potentials (POCPs). It is therefore critical that in future field campaigns, alcohols such as ethanol and isopropanol are measured to evaluate their impacts on radical budgets and ozone production.

## 3.7. Experimental radical budget balance

### 3.7.1. Budget for total $RO_x$

Figure 7 shows median diel profiles of the rates of $RO_x$ production and destruction calculated using equations (E1–E2), split according to wind direction. In NW–SE air, which was encountered for ~40% of the data, both $P(RO_x)$ and $D(RO_x)$ peak at ~0.7–0.8 ppbv h$^{-1}$ around 12:00 UTC with a fairly symmetrical profile either side of midday (solar noon ~12:00 UTC based on cloud-free days). Within uncertainty, $P(RO_x)$ and $D(RO_x)$ are equal for much of the day, indicating budget closure, apart from around midnight. In SW air (~60% of the data), $P(RO_x)$ and $D(RO_x)$ peak at ~1.2–1.4 ppbv h$^{-1}$ around 10:00 UTC, where $D(RO_x)$ displays a broader profile than that in NW–SE air. $P(RO_x)$ is always smaller than $D(RO_x)$, by greater than the measurement uncertainty in the hours 06:00–09:00 UTC, late afternoon, and evening, suggesting missing $RO_x$ sources in SW air on the order of ~0.2–0.6 ppbv h$^{-1}$ at these times. Since $NO_3$ + VOC reactions were omitted from the budget analysis, it is suggested that $NO_3$ radical reactions, acting as a net $RO_x$ source, would likely reduce the gap between $P(RO_x)$ and $D(RO_x)$ at night.

### 3.7.2. Budget for OH

Figure 8 displays median diel profiles of OH rates of production and destruction calculated using equations (E3–E4). In contrast to the $RO_x$ budget, in which rates of production and destruction were in balance for most of the 24 h diel cycle in NW–SE air, $P(OH)$ is almost always smaller than $D(OH)$ in NW–SE air, which, since $D(OH)$ is calculated directly from measured OH reactivity, indicates missing OH sources of up to ~2–3 ppbv h$^{-1}$. In addition, $D(OH)$ exhibits two diel peaks at ~10:00 UTC (~2.5 ppbv h$^{-1}$) and ~16:00 UTC (~3.5 ppbv h$^{-1}$), whereas $P(OH)$ peaks only once at ~1.5 ppbv h$^{-1}$ in the morning and then decreases through midday and over the course of the afternoon. It should be noted that the peak at ~16:00 UTC is the 1-hour median of many 15-minute data points, corresponding to different days, and is not driven by a single high value in the averaging. In SW air, OH rates of production and destruction are reasonably well balanced throughout the day, with $P(OH)$ slightly smaller than $D(OH)$ by ~0.2 ppbv h$^{-1}$ on average, but differences of up to ~1 ppbv h$^{-1}$ (~14:00 UTC) can be seen.

### 3.7.3. Budget for HO₂

In contrast to the $RO_x$ and OH budgets, the $HO_2$ budgets calculated using equations (E5–E6) (Figure 9) are out of balance throughout the daytime in both NW–SE and SW air, with $HO_2$ rates of production greatly exceeding rates of destruction by up to an order of magnitude in the morning. $P(HO_2)$ peaks around ~10:00 UTC at ~8 and ~14 ppbv h$^{-1}$ in NW–SE and SW air, respectively. At this time, known $HO_2$ sinks amount only to ~1 ppbv h$^{-1}$. $D(HO_2)$ reaches diel maxima of only ~1 and ~2 ppbv h$^{-1}$ in NW–SE and SW air, respectively. The imbalance between $P(HO_2)$ and $D(HO_2)$ cannot be accounted for by the measurement uncertainty in $D(HO_2)$ of ~44% (derived from calibration accuracy and reproducibility), and would imply the very rapid build-up of $HO_2$ to multi-ppbv levels, which was not observed.

### 3.7.4. Budget for RO₂

The diel profiles of $RO_2$ rates of production and destruction calculated using equations (E7–E10) (Figure 10) bear close resemblance to those of $HO_2$ but with opposite sign imbalances, i.e., for $RO_2$, destruction greatly exceeds known production processes. In NW–SE air, $D(RO_2)$ peaks at ~7 ppbv h$^{-1}$ around ~10:00 UTC, at which time known $RO_2$ sources amount to only ~0.6 ppbv h$^{-1}$. Maximum $P(RO_2)$ occurs around midday at almost 1 ppbv h$^{-1}$, which is a factor of three slower than $D(RO_2)$ at the same time. $RO_2$ destruction is even faster in SW air, reaching ~13 ppbv h$^{-1}$ around 09:00–10:00 UTC, at which time $P(RO_2)$ is only ~0.6–1.5 ppbv h$^{-1}$. $RO_2$ production rates were almost twice as fast in SW air compared to NW–SE air, with a diel maximum of ~2 ppbv h$^{-1}$ around ~14:00 UTC.

### 3.8. Dependences on NO mixing ratios

To summarise thus far, in NW–SE air during daytime the total $RO_x$ budget is balanced but OH is missing a source, and $HO_2$ production rates greatly exceed $HO_2$ destruction rates while the opposite is true for $RO_2$. In SW air, evidence for missing $RO_x$

sources is found in the morning and late afternoon, while the daytime OH budget is balanced, and the same problems with the $HO_2$ and $RO_2$ budgets in NW–SE air are also found (i.e., calculated $RO_2 \rightarrow HO_2$ conversion is perhaps too fast in both wind sectors).

As radical levels and measurement-model ratios are strongly dependent on NO mixing ratios, it is expected that the budget imbalances may also have been influenced by NO. As shown in Figure S10, this was indeed the case, with the difference between the rate of destruction and the rate of production displaying a strong dependence on NO for $RO_x$, $HO_2$, and $RO_2$.

      $D(RO_x) - P(RO_x)$ increases with NO in NW–SE air, from virtually zero (i.e., budget balance) at <600 pptv NO to almost 1 ppbv h$^{-1}$ at ~2000–3000 pptv NO. This suggests missing $RO_x$ sources and/or overestimated $RO_x$ loss rates under high $NO_x$

conditions. However, in SW air, the difference between destruction and production rates exhibits a U-shaped dependence on NO. $D(RO_x) - P(RO_x)$ is ~1 ppbv h$^{-1}$ at ~100–200 pptv NO, scattered around zero in the ~300–600 pptv NO region, and increases again to ~0.5 ppbv h$^{-1}$ at 1000–2000 pptv NO. This may suggest that in SW air, the radical chemistry is well-understood at moderate $NO_x$, but that there are missing $RO_x$ sources and/or overestimated $RO_x$ loss rates at both low and high $NO_x$. It is unclear why the budget is balanced at low $NO_x$ in NW–SE air, but not SW air, but may relate to differences in VOC

composition between the two wind sectors.

      For OH, the rate of destruction minus the rate of production does not exhibit any obvious trend with NO level, with values of ~0–2 ppbv h$^{-1}$ across the entirety of NO space encountered during ICOZA, in both NW–SE and SW air. Since $D(OH)$ is constrained by measured OH reactivity, this suggests the presence of missing OH sources, which are independent of NO. One possibility is that OH radicals were formed from the reactions of $HO_2$ or $RO_2$ with species other than NO, discussed in further

detail in Section 3.10. Although this contrasts with the lack of NO-dependence found for $D(OH) - P(OH)$, their ratios $D(OH)/P(OH)$ do show a decreasing trend with NO (data not shown), consistent with the presence of missing OH sources under low $NO_x$ conditions.

      For the $HO_2$ and $RO_2$ budgets, the NO trends are the same in NW–SE and SW air. $D(HO_2) - P(HO_2)$ is close to zero at low NO, but becomes more negative with increasing NO, reaching –(12–15) ppbv h$^{-1}$ at >1000 pptv NO. Similarly, for $RO_2$,

the budget is closed at low NO but $D(RO_2) - P(RO_2)$ reaches up to +(13–16) ppbv h$^{-1}$ at high NO. Thus, the $HO_2$ and $RO_2$ budget balances show virtually the same trends with NO in magnitude, but with opposite sign. This is strong evidence that the rate of $RO_2 \rightarrow HO_2$ propagation has been substantially overestimated and is discussed in further detail in Section 3.11.

### 3.9. Radical sources and sinks

### 3.9.1. $RO_x$ initiation and termination

Figure S15 in the Supplementary Information shows a time series of the experimentally determined radical budgets for the ICOZA campaign, which demonstrates the variability of the total rate of production and the total rate of destruction for OH, $HO_2$, $RO_2$ and $RO_x$ (=sum of OH, $HO_2$ and $RO_2$). Figure 11 displays average diel profiles of the contributions of known $RO_x$ sources and sinks, split according to wind direction. Table 2 summarises these data by presenting the median daytime (defined

as $J(O^1D) > 5 \times 10^{-7}$) percentage contributions of individual $RO_x$ sources and sinks in NW–SE and SW air. In NW–SE air, $RO_x$ initiation had roughly equal contributions from $O^1D + H_2O$ and HONO photolysis (~37%) on average, where HONO photolysis dominated $RO_x$ initiation in the early morning (~05:00–08:00 UTC) but was less important over the rest of the day. In contrast, HONO photolysis was dominant (median 44% versus 29% for $O^1D + H_2O$) in the more polluted SW air throughout the day. This might be expected based on the mixing ratios of HONO in each wind sector type, with median values of 52 and 97 pptv in NW–SE and SW air, respectively. The contributions from carbonyl (HCHO, acetaldehyde, and acetone) photolysis (~23–25%) and ozonolysis (~3%) were about the same in each wind sector type.

In terms of $RO_x$ termination, the main contributors in both wind sector types were calculated to be alkyl nitrate formation, $RO_2 + HO_2$ reactions, and the reaction of OH with $NO_2$ to yield $HNO_3$. In NW–SE air, these three loss processes were of equal importance on average (~30%), with alkyl nitrate formation dominant around ~09:00 UTC and $RO_2 + HO_2$ reactions dominant in the afternoon. The contributions from $HO_2 + HO_2$, $RO_2 + RO_2$, and OH + NO were all small on average (<4%). The contributions from alkyl nitrate and $HNO_3$ formation were greater in SW air (almost 40% on average), whereas $RO_2 + HO_2$ reactions were less important (~14%), driven by differences in $NO_x$ levels between the two wind sectors. Again, alkyl nitrate formation was most important in the morning, but also contributed substantially throughout the afternoon. $HO_2 + HO_2$ and $RO_2 + RO_2$ reactions were almost negligible (~1%), but the contribution from OH + NO (~6%) was greater than in NW–SE air (~3%).

### 3.9.2. OH production and $k'_{OH}$

The breakdown of OH production and its comparison to measured OH destruction ([OH] $\times k'_{OH}$) is given in Figure 12, again split by wind direction. These data are summarised in Table 3, which shows the median daytime contributions of the known OH sources. Similarly, Figure 13 gives the breakdown of OH reactivity and comparison to measured $k'_{OH}$, also summarised in Table 3.

OH production was dominated throughout the daytime by the secondary source $HO_2 + NO$ in both NW–SE (~50% on average) and SW (~70%) air. In NW–SE air, the next most important OH sources were the primary sources $O^1D + H_2O$ and HONO photolysis, with average contributions of ~23% each. Similar to the $RO_x$ budget (Section 3.9.1), HONO photolysis (~18%) was more important than $O^1D + H_2O$ (~12%) as an OH source in the more polluted SW air. Radical recycling from $HO_2 + O_3$ (<3%) and radical initiation from ozonolysis (<1%) were of only minor importance in both wind sector types.

In terms of OH loss (Table 3), the most important OH reactant was CO (NW–SE daytime median: ~42% of calculated OH reactivity, SW: ~27%), followed by $NO_2$ (~20%, ~26%), reflecting the overall dominance of inorganic reactants to calculated OH reactivity. We restate that the calculated reactivity is approximately half that of the measured value.but In terms of organic OH reactivity, carbonyls (~13%, ~21%; mostly (~57%) HCHO) and alkenes and alkynes (~6–8%; mostly (~62%) propene) were the most important species. The dialkenes isoprene and 1,3-butadiene made small contributions to OH reactivity (~4–6%), whereas the contributions from aromatics, alkanes, and methanol were all minor (≤3%). Missing OH reactivity was similar in magnitude in both wind sectors (~50%). Monoterpenes (MTs) were not included in the calculation of OH reactivity

as their sum (measured using proton transfer reaction–mass spectrometry, PTR–MS) was generally below the LOD; if we use these data, the maximum contribution of MTs was only ~0.4 s$^{-1}$ (median 0.04 s$^{-1}$, compared to measured $k'_{OH}$ ~ 4.7 s$^{-1}$), using the rate constant for OH + limonene.

## 3.10. Attempting to balance the radical budgets

In this section, we describe various attempts to try and balance the radical budgets through making modifications to the calculation of experimental budgets. Such modifications include: (1) the addition of generic radical recycling processes, (2) reduction of the rate of RO$_2$ → HO$_2$ conversion, (3) inclusion of heterogeneous HO$_2$ uptake, and (4) addition of chlorine chemistry.

### 3.10.1. Reducing the rate of RO$_2$ → HO$_2$ conversion

Whalley et al., (2018)presented field measurements of HO$_2$ and RO$_2$ radicals in London. HO$_2$ levels were significantly overpredicted by an MCM model during the daytime, particularly in air that had passed over central London. It was found that HO$_2$ concentrations could be reasonably well simulated if the fraction of RO$_2$ radicals that propagated to HO$_2$ (i.e., the branching ratio $\alpha$ in reactions (R11) and (R4)) was reduced. To achieve good agreement, $\alpha$ was reduced to 0.15, compared to $\alpha$ ~ 0.5 in the base model, a factor of ~3 reduction.

In the present work, $\alpha$ was set to 0.95 based on literature values of the branching ratios for alkyl nitrate formation ($\beta$) of ~5% (Orlando and Tyndall, 2012; Tan et al., 2019). However, even with such a low RONO$_2$ branching ratio, $P$(RONO$_2$) values of up to ~0.7 ppbv h$^{-1}$ (Figure S11) are already very high considering previous measurements of RONO$_2$ at Weybourne were on the order of tens of pptv (Worton et al., 2010). Therefore, it is not thought that changing the value of $\alpha$ is appropriate for ICOZA. Instead, we have artificially reduced the rate of RO$_2$ → HO$_2$ conversion by changing the total rate constant of reactions (R11) and (R4a) (originally $9.0 \times 10^{-12}$ cm$^3$ molecule$^{-1}$ s$^{-1}$ at 298 K). The impact of reducing the RO$_2$ + NO rate constant by a factor of 5 on the HO$_2$ and RO$_2$ budgets is shown in Figure S11. It can be seen that the HO$_2$ and RO$_2$ budgets are now reasonably well balanced in the afternoon, but still $P$(HO$_2$) > $D$(HO$_2$) and $P$(RO$_2$) < $D$(RO$_2$) by ~1–2 ppbv h$^{-1}$ in the morning. It should be noted that no evidence exists for such low RO$_2$ + NO rate constants, with published $k$(298 K) values in the range ~8–20 $\times 10^{-12}$ cm$^3$ molecule$^{-1}$ s$^{-1}$ and associated uncertainties of ~15–35% (Orlando and Tyndall, 2012), although the kinetics of relatively few RO$_2$ species with NO have been studied directly.

### 3.10.2 Inclusion of heterogeneous HO$_2$ uptake

Up to now, the heterogeneous uptake of HO$_2$ has not been considered in the HO$_2$ budget calculations. We discuss the impact of this chemistry on the experimental HO$_2$ budget in the Supplementary Information, which includes Figure S12.

 ### 3.10.3. Inclusion of chlorine atom initiated oxdiation chemistry

We discuss the impact of chlorine atom initiated oxidation chemistry on the experimental $HO_2$ budget in the Supplementary Information, which includes Figure S13.

## 3.11. Ozone production

### 3.11.1. Calculated $P(O_x)$ and comparison to MCM model predictions

The *in situ* ozone production rate, $p(O_3)$, may be defined in terms of the rate of net NO $\rightarrow$ $NO_2$ conversion (Cazorla et al., 2012), i.e., $p(O_x)$ where $O_x = O_3 + NO_2$:

$$p(O_3) \approx p(O_x) = k_{11}[HO_2][NO] + k_{RO2+NO}[RO_2][NO] \times \alpha \qquad \text{(E14)}$$

Here, $\alpha$ is the branching ratio for the channel generating $HO_2 + NO_2$ formation (reaction (R4)) and $k_{RO2+NO}$ is the sum of rate constants for reactions (R11) and (R4). The chemical loss rate of ozone, $l(O_3)$, may be derived from the rate of radical-$NO_x$ termination reactions and the loss of $O_3$ to $HO_2$, approximated by:

$$l(O_3) \approx l(O_x) = k_6[OH][NO_2][M] + k_{RO2+NO}[RO_2][NO] \times \beta + k_{12}[HO_2][O_3] , \qquad \text{(E15)}$$

where $\beta$ ($= 1 - \alpha$) is the branching ratio for $RONO_2$ formation (reaction (R7)). The net ozone production rate, $P(O_3)$, is then obtained from the difference between equations (E13) and (E14):

$$P(O_3) \approx P(O_x) = p(O_x) - l(O_x) \qquad \text{(E14)}$$

Calculation of $P(O_3)$ ($\approx P(O_x)$) from FAGE observations of $HO_2$ and $RO_2$ radicals was one of the main aims of the ICOZA project.

Median diel profiles of the rate of net ozone production, $P(O_x)$, calculated from measured and modelled OH, $HO_2$, and $RO_2$ radical concentrations are shown in Figure S14. Here, $P(O_x)$ was calculated from equations (E12–E14) with the same values of $k_{RO2+NO}$ and $\alpha$ ($= 0.95$) applied to both observations and model predicted concentrations of total $RO_2$ (i.e., model $P(O_x)$ was not calculated from the rate constants and yields for individual $RO_2$ species). $k_{RO2+NO}$ was set to the generic value used in the MCM ($k_{RO2+NO} = 2.7 \times 10^{-12}$ exp(360/T) = $9.0 \times 10^{-12}$ cm$^3$ molecule$^{-1}$ s$^{-1}$ at 298 K; for reference, $k_{CH3O2+NO} = 7.7 \times 10^{-12}$ cm$^3$ molecule$^{-1}$ s$^{-1}$ at 298 K).

In NW–SE air, $P(O_x)$ derived from measurements using the FAGE instrument peaks at ~16 ppbv h$^{-1}$ at 09:30 UTC when NO and peroxy radical levels are both high, before decreasing sharply in the afternoon to ~0.7–1.4 ppbv h$^{-1}$. Model-calculated

$P(O_x)$ also peaks at 09:30 UTC but at a ten-fold lower value of ~1.6 ppbv h$^{-1}$. The afternoon decrease is less severe than for FAGE-calculated $P(O_x)$, resulting in good agreement between FAGE- and model-calculated $P(O_x)$ in the afternoon. In SW air, FAGE-calculated $P(O_x)$ displays a broader morning peak in the hours ~07:00–10:00 UTC of ~10–15 ppbv h$^{-1}$. In comparison to NW–SE air, afternoon FAGE-calculated $P(O_x)$ was greater with values of ~5–8 ppbv h$^{-1}$. Daytime model-calculated $P(O_x)$ is in the range 0.3–2.3 ppbv h$^{-1}$, peaking at 14:30 UTC, and underpredicts the observations throughout the daytime, in contrast to NW–SE air. $P(O_x)$ will be impacted by a change in the rate coefficient for $RO_2+NO$, owing to the change in $RO_2$ and $HO_2$ budgets, as shown Figure S11.

### 3.11.2. Ozone production regime – $L_n / Q$

The ratio of the rates of radical loss to $NO_x$ ($L_n$, reactions (R5–R7)) to total radical initiation ($Q = P(RO_x) \approx D(RO_x)$) has been proposed as a simple metric to assess whether ozone production is $NO_x$ or VOC limited (Kleinman et al., 1997; Kleinman et al., 2001). A ratio above 0.5 suggests that ozone production is VOC limited, while values below 0.5 indicate that ozone production is in the $NO_x$ limited regime. This metric has been used to assess ozone production sensitivity in previous urban campaigns (Mao et al., 2010; Griffith et al., 2016).

In the present work, calculated $D(RO_x)$ was generally slightly greater than $P(RO_x)$ (Figure 7) and calculated $L_n$ often exceeded $P(RO_x)$, leading to $L_n / Q$ ratios of greater than 1. For this reason, we have used the ratio of radical loss to $NO_x$ to total radical destruction ($L_n / D(RO_x)$) to assess ozone production sensitivity for the ICOZA campaign. Median diel profiles of daytime (06:00–21:00 UTC) $L_n / D(RO_x)$ calculated from measured radicals in NW–SE and SW air are shown in Figure 14. In NW–SE air, ozone production was generally VOC limited (i.e., $L_n / D(RO_x) > 0.5$), but with some $NO_x$ limited ozone production around midday. In contrast, ozone production was VOC limited throughout the daytime in the more polluted SW air.

### 3.11.3. $P(O_x)$ dependence on NO mixing ratios

Figure 15 shows that both FAGE- and model-calculated $P(O_x)$ are strongly dependent on NO, with similar trends in NW–SE and SW air. FAGE-calculated P($O_x$) shows a consistent increase with NO in both NW–SE and SW air, with values of <1 ppbv h$^{-1}$ below 100 pptv NO and up to ~17 ppbv h$^{-1}$ at ~2–3 ppbv NO. In contrast, model-calculated P(Ox) starts to fall off a little above 1 ppbv NO in NW–SE air, but generally increases with NO in SW air, but the latter is largely due to a single point at 2 ppb NO. Below ~500 pptv NO, FAGE- and model-calculated $P(O_x)$ are in reasonable agreement within combined uncertainties. However, above this threshold, FAGE-calculated $P(O_x)$ is much greater than model-calculated $P(O_x)$, with measurement-to-model ratios of up to ~5–15 for NO ~2–3 ppbv. $NO_x$ levels were not high enough to show any onset of a plateau in FAGE-calculated $P(O_x)$.

## 4. Discussion

### 4.1. Comparison to previous coastal field campaigns

Table 4 summarises previous measurements of OH, $HO_2$, $HO_2$+$RO_2$ and OH reactivity at the WAO site, and also at other selected locations in the MBL. A more detailed summary up until 2012 can be found in Stone et al., 2012. For other ground and ship-based campaigns in the MBL, measured noontime OH concentrations were mostly in the range ~4–6 × $10^6$ molecule $cm^{-3}$, and generally the observations have been found to agree with model predictions to within ~30% on average during the daytime in the MBL (Sommariva et al., 2004; Sommariva et al., 2006; Whalley et al., 2010; Beygi et al., 2011; Van Stratum

et al., 2012). During the NASA airborne Atmospheric Tomography study (ATom), OH concentrations in the MBL were on the order of ~1–4 × $10^6$ molecule $cm^{-3}$ and a model was able to reproduce them to generally within 40% (Brune et al., 2020). In NAMBLEX $HO_2$ levels were overpredicted by up to a factor of 2 (Sommariva et al., 2006). These results are almost identical to the findings of ICOZA despite the substantial differences in chemical conditions (e.g., the much lower anthropogenic influence and the role of halogen species during NAMBLEX). For other campaigns, $HO_2$ concentrations were generally above

~2 × $10^8$ molecule $cm^{-3}$ (Sommariva et al., 2004; Whalley et al., 2010; Beygi et al., 2011), higher than the range observed during ICOZA (~0.5–2 × $10^8$ molecule $cm^{-3}$), and the observations have mostly been overpredicted. During ATom, $HO_2$ concentrations in the MBL were on the order of ~1–5 × $10^8$ molecule $cm^{-3}$ and were well captured by a model (agreement within 40%) (Brune et al., 2020). The mean missing OH reactivity in terms of the difference between measured OH reactivity and that calculated using trace gases only was 1.9 $s^{-1}$ (39%) during TORCH 2 (cf. 2.4 $s^{-1}$, 48% for ICOZA). A box model

using MCMv3.1 chemistry (Bloss et al., 2005a) was used to simulate OH reactivity, reducing the missing reactivity to 1.4 $s^{-1}$ or 29% (cf. 1.7 $s^{-1}$, 36% for ICOZA). Lee et al. (2009b) speculated that the missing reactivity may be due to a potentially large number of unmeasured, high molecular weight aromatic compounds, but that this could also be due to missing OVOCs, as we have suggested based on the data in Figure S9. To date, only a handful of studies have measured OH reactivity at coastal locations (examples in Table 4) but there have been airborne campaigns conducted in the marine boundary layer. An airborne

OH reactivity instrument was deployed during flights over the Pacific Ocean for the Intercontinental Chemical Transport Experiment-B (INTEX-B) campaign (Mao et al., 2009). In the boundary layer (i.e., < 2 km altitude), measured OH reactivity was ~4 $s^{-1}$ on average, while that calculated from measured reactants was only ~1.5–2 $s^{-1}$ (i.e., ~50–60% missing), similar to ICOZA. During the NASA Atmospheric Tomography (ATom) campaign involving flights over the Atlantic and Pacific Oceans, measured OH reactivity at < 2 km was on the order of ~2 $s^{-1}$, with missing reactivity on the order of ~0.5–1 $s^{-1}$ (~25–

50% missing) (Thames et al., 2020). The authors suggested that, based on correlations of missing OH reactivity with HCHO, DMS, butanal, and sea surface temperature, there were unmeasured/unknown VOCs/OVOCs associated with oceanic emissions, in agreement with our findings.

## 4.2. Differences and similarities between the NW–SE and SW wind sectors

Many aspects are fairly similar between the two wind sector types, for example measured OH, HO$_2$, RO$_2$, and OH reactivity levels. Perhaps the most striking difference between the two wind sector types is the model performance for OH and HO$_2$ (Figure 3). In NW–SE air, measured OH is underpredicted by the PSS calculation by ~35% on average, but reasonable agreement is found in SW air (within 20% on average). Similarly, HO$_2$ is overpredicted by both the MCM-base and MCM-carb models by a factor of 2–3 during the afternoon in NW–SE air, but reasonable agreement is found between measured HO$_2$ and the MCM-carb model for daytime SW air. In contrast, the model underprediction of RO$_2$ is more severe in SW air compared to the NW–SE sector, suggesting that the good agreement found for HO$_2$ may be fortuitous (i.e., if the model was able to reproduce RO$_2$, then RO$_2$ + NO → HO$_2$ reactions would likely lead to the model overpredicting HO$_2$). The underprediction of OH and overprediction of HO$_2$ in NW–SE air, only occurs at low-NO$_x$ (Figure S7 and Figure 5). Possible reasons for these discrepancies are discussed in Section 4.4.

## 4.3. Differences between models

The carbonyls HCHO and MVK+MACR were not constrained in the MCM-base model because of several gaps in the time series of these measurements. The MCM-base model performance in simulating these carbonyls was assessed, where it was found that there was reasonable agreement for MVK+MACR on a diel average basis, but that HCHO concentrations were significantly overpredicted in the afternoon (data not shown). The differences in the calculated concentrations of these OVOC compounds is the cause of the differences between the MCM-base and MCM-carb simulations of radical species (Figure 3). Similarly, HO$_2$ concentrations were generally overpredicted by both the MCM-base and MCM-carb models, and therefore constraining to HO$_2$ (MCM-hox model) has impacts on the model OH and RO$_2$ concentrations (Figure 3). Since the MCM-base model also underpredicted OH reactivity, there are additional differences between the PSS calculation of OH and the MCM model predictions.

For OH, differences between the MCM-base and MCM-carb runs are relatively minor and both positive and negative (Figure 3). There are two competing effects here: HCHO is a minor OH sink (~6% and thus an overprediction of HCHO would lead to lower OH in MCM-base relative to MCM-carb. In contrast, the base model overprediction of HCHO leads to a greater HO$_2$ source strength that would drive higher OH levels in MCM-base relative to MCM-carb. Similarly, HO$_2$ was overpredicted in MCM-base and MCM-carb such that constraining the model to HO$_2$ (MCM-hox) resulted in lower model OH levels. Finally, due to the MCM model underprediction of OH reactivity, PSS calculated OH levels were lower than the MCM model concentrations. At low NO (Figure S7), the better agreement found for the OH/MCM-base case compared to the OH/PSS case is driven by the MCM overprediction of HO$_2$, i.e., the agreement for OH does not necessarily mean the OH chemistry is well understood at low NO. Peak afternoon RO$_2$ concentrations were similar for the MCM-base and MCM-carb simulations (Figure 3). However, the reduced OH in MCM-hox results in reduced afternoon RO$_2$ levels.

## 4.4. Overprediction of $HO_2$ under low-$NO_x$ conditions in NW–SE air

In several previous field campaigns, $HO_2$ concentrations were overpredicted under low-$NO_x$ conditions (Sommariva et al., 2004; Sommariva et al., 2006; Kanaya et al., 2007; Griffith et al., 2013; Whalley et al., 2018). Extremely low NO levels of < 3 pptv were observed during the Southern Ocean Photochemistry Experiment (SOAPEX-2), which took place at Cape Grim in austral summer 1999. $HO_2$ observations were overpredicted by ~40%, but improved agreement could be found by inclusion of $HO_2$ uptake with an uptake coefficient ($\gamma_{HO2}$) of unity (Sommariva et al., 2004). $HO_2$ uptake was considered in the present

work, using $\gamma_{HO2} = 0.1$. As discussed above, $HO_2$ measurements were overpredicted by a factor of 2 during NAMBLEX, for which the model analysis was performed for days with low $NO_x$ levels (NO < 30 pptv) (Sommariva et al., 2006). Agreement was improved when the model was constrained to measured OVOCs (acetaldehyde, methanol, and acetone in the case of NAMBLEX), similar to the improvement in measurement-model agreement seen for the MCM-carb run (constrained to HCHO and MVK+MACR) in the present work (Figure 3). Additionally, at Mace Head, seaweed beds are exposed at low tide that

represent a significant source of reactive halogen species such as $I_2$ and $CH_2I_2$ (Carpenter et al., 1999; Carpenter et al., 2003; Mcfiggans et al., 2004). Halogen oxides (XO, where X = Br, I) are able to convert $HO_2$ to OH:

$$HO_2 + XO \rightarrow HOX + O_2 \tag{R18}$$
$$HOX + h\nu \rightarrow OH + X, \tag{R19}$$

where hypohalous acids (HOX) may also undergo heterogeneous loss to aerosols. In a steady-state analysis, (Bloss et al., 2005b) found that up to 40% of $HO_2$ could be lost to IO under low-$NO_x$ conditions, for measured IO levels of 0.8–4.0 pptv (Commane et al., 2011). In the full modelling study (Sommariva et al., 2006), constraining the model to BrO and IO resulted in similar decreases in model $HO_2$, depending on the uptake coefficients used for HOI and HOBr. Reactive iodine species were

960 not measured during ICOZA, and their influence is expected to be negligible due to the lack of seaweed beds at the WAO site. However, it is possible that there was a source of reactive bromine through sea salt aerosol chemistry (Keene et al., 2009). We therefore speculate that inclusion of reactive halogens could simultaneously reduce the underprediction of OH and the overprediction of $HO_2$ under low-$NO_x$ conditions in NW–SE air. To our knowledge there have been no measurements of $I_2$, BrO, or IO at the WAO (John Plane, personal communication).

OH and $HO_2$ were measured at Rishiri Island, Japan, in September 2003 (Kanaya et al., 2007). Daytime $HO_2$ levels were overpredicted by almost a factor of 2. In addition to halogen chemistry and $HO_2$ uptake, the authors also considered the possibility that $HO_2 + RO_2$ reactions were faster than previously thought. Increasing the rate of $HO_2 + RO_2$ reactions would result in increased $RO_2$ destruction rates, therefore worsening the agreement between $RO_2$ destruction and production rates, which should be in balance. For this reason, we do not think that faster-than-expected $HO_2 + RO_2$ reactions are the cause of

the overprediction of $HO_2$ levels under low-$NO_x$ conditions in NW–SE air during ICOZA.

During the Clean air for London (ClearfLo) campaign in summer 2012, $HO_2$ concentrations were overpredicted by a box model using MCMv3.2 by up to a factor of 10 at low NO (< 1 ppbv) (Whalley et al., 2018). The model $HO_2$ was somewhat reduced but the observations could still not be reconciled after inclusion of both $HO_2$ aerosol uptake (using $\gamma_{HO2} = 1$) and autoxidation chemistry (Bianchi et al., 2019), which is now known to play a significant role in the gas phase oxidation of both BVOCs (Crounse et al., 2011; Ehn et al., 2014; Jokinen et al., 2014; Berndt et al., 2016; Zha et al., 2017) and anthropogenic VOCs (AVOCs) (Wang et al., 2017; Molteni et al., 2018; Mehra et al., 2020; Wang et al., 2020). Whalley et al. (2018) found that good agreement between the model and $HO_2$ measurements could be found if the rate of $RO_2 + NO \rightarrow HO_2$ propagation was reduced, in their case by reducing the branching ratio for alkyl nitrate formation.

## 4.5. Underprediction of $RO_2$ under high-$NO_x$ conditions

Under moderate and high-$NO_x$ conditions (above ~200–300 pptv NO), reasonable measurement-model agreement is found for OH and $HO_2$, i.e., generally to within a factor of 2 (Figure S7 and Figure 5). However, total $RO_2$ concentrations are much more significantly underpredicted, by as much as a factor of ~30 at the highest NO levels encountered (above ~2 ppbv), coincident with increased missing OH reactivity. As shown in Figure 6, this phenomenon has been observed in several other field campaigns (Tan et al., 2017; Tan et al., 2018; Whalley et al., 2018; Slater et al., 2020; Whalley et al., 2021). Tan et al. (2017) found that an additional primary $RO_2$ source from chlorine chemistry could explain a small portion (10–20%) of the missing $RO_2$ in their study. Whalley et al. (2018) found that chlorine chemistry increased modelled $RO_2$ for the ClearfLo campaign by ~20% in the morning when $NO_x$ levels were high, in comparison to $RO_2$ underpredictions of greater than factor of 10. Since the major Cl atom precursor $ClNO_2$ was measured during ICOZA (Sommariva et al., 2018) and constrained in all model scenarios, $ClNO_2$ photolysis to form Cl atoms and the subsequent reactions of Cl with VOCs is not thought to be the source of the missing $RO_2$ in the present study. However, as the chlorine chemistry in MCMv3.3.1 is limited to reactions with alkanes, additional chlorine chemistry (e.g., reactions with alkenes, OVOCs, etc.) may be needed to fully assess the role of chlorine during ICOZA.

Since missing OH reactivity was also found at high $NO_x$ conditions, some of the missing $RO_2$ may be due to the reactions of OH with unmeasured VOCs. However, evidence is also found for missing $RO_2$ sources at high $NO_x$ using calculations constrained to measured OH reactivity. Thus the missing OH reactivity cannot fully explain the missing $RO_2$. It is possible that the missing $RO_2$ found for ICOZA is not due to a missing $RO_2$ source, but an overestimated $RO_2$ sink. As discussed above, reducing the rate of $RO_2 + NO \rightarrow HO_2$ propagation (i.e., a reduced $RO_2$ sink) helps to resolve budget imbalances for $HO_2$ and $RO_2$, similar to that found for $HO_2$ by Whalley et al. (2018). We found the largest improvement to the experimental budget balance in the morning, when $NO_x$ levels were at their highest. Reducing the $RO_2 + NO$ rate constant by a factor of 5 is not consistent with accepted laboratory measurements for which uncertainties in the range ~15–35% are reported (Orlando and Tyndall, 2012). It is therefore imperative that more laboratory studies are conducted to measure $RO_2 + NO$ rate constants with a wide variety of $RO_2$ types.

Recently, Whalley et al. (2020) presented measurements of OH, $HO_2$, $RO_2$, and OH reactivity in summertime Beijing. $RO_2$ concentrations were underpredicted by a box model with MCMv3.3.1 chemistry, most severely at high $NO_x$ ( Figure 6). Missing OH reactivity was also identified. The measurement-model agreement for $RO_2$ was significantly improved after the model inclusion of an α-pinene derived $RO_2$ radical, C96O2 (MCM nomenclature), formed at a rate set equal to the level of missing OH reactivity. This complex $RO_2$ species does not generate $HO_2$ directly from its reaction with NO, but instead the RO radical formed preferentially isomerises (via a H-shift) to form another $RO_2$ radical in the presence of $O_2$, and undergoes multiple $RO_2 + NO \rightarrow R'O_2$ reactions before eventually forming $HO_2$. Such autoxidation chemistry has the net effect of reducing the rate of $RO_2 \rightarrow HO_2$ propagation, and effectively extends the lifetime of $RO_2$ radicals, resulting in higher concentrations. Based on the results in the present work, it is possible that similar chemistry occurred during the ICOZA campaign, although it is unlikely that a BVOC was involved because of the low biogenic influence at the WAO site. However, aromatic species, more relevant to ICOZA, have also been shown to undergo autoxidation (Wang et al., 2017; Mehra et al., 2020).

## 4.6. Budget analyses

Overall, the results show that during the daytime, the budget of total $RO_x$ is virtually closed in both wind sectors (Figure 7), although there is evidence for relatively small missing $RO_x$ sources in the early morning and late afternoon in SW air. However, the budgets for individual $RO_x$ species are out of balance, most severely for $HO_2$ (Figure 9) and $RO_2$ (Figure 10). The dependence of these imbalances on $NO_x$ levels (Figure S10) implies that there are quantitative limitations to our understanding of the processes that interconvert $RO_x$ species, in particular those that convert $RO_2$ to $HO_2$. The worst agreement between experimental $HO_2$ and $RO_2$ production and destruction rates is found at high $NO_x$, suggesting that it is under these conditions that our understanding is most incomplete, or that the uncertainties, perhaps associated with any interferences in the measurements, have been underestimated.

First, the $RO_2 + NO$ rate constant was reduced to assess the impact on experimental $HO_2$ and $RO_2$ budgets. Of all the hypotheses tested in the present work, this resulted in the best agreement between $HO_2$ and $RO_2$ production and destruction rates (Figure S11). However, the $RO_2 + NO$ rate constant had to be reduced by a factor of 5 to achieve this, for which no evidence exists, given that accepted laboratory measurements of these rate constants for specific peroxy species have not reported values as low and have uncertainties in the range ~15–35% (Orlando and Tyndall, 2012). We reiterate therefore imperative that more laboratory studies are conducted to measure $RO_2 + NO$ rate constants with a wide variety of $RO_2$ types. Given that organic OH reactivity was dominated by OVOCs and alkenes and alkynes (Table 3), it is perhaps here where efforts should focus. On the other hand, missing OH reactivity is also significant and may be due to unmeasured OVOCs rather than BVOCs. Alternatively, previous experiments at the WAO have suggested that a multitude of aromatic species may be responsible for missing OH reactivity (Lee et al., 2009b). We do note however, that different $RO_2$ species have different NO rate constants, while in this work we were only able to treat $RO_2$ radicals as a single species with a single NO rate constant, which introduces a bias in our analyses. Reduced $RO_2 + NO$ rate constants could help to reconcile measurement-model

discrepancies seen for $RO_2$ at high NO in other campaigns (Tan et al., 2017; Tan et al., 2018; Whalley et al., 2018; Slater et al., 2020). In addition, the reduced rate constants would result in longer $RO_2$ lifetimes with respect to NO. This has implications for autoxidation chemistry (Bianchi et al., 2019): longer $RO_2$ lifetimes would allow more time for unimolecular autoxidation

reactions to compete with the bimolecular NO reaction, resulting in more efficient formation of highly-oxidised molecules (HOMs) under high $NO_x$ conditions. This may help to explain why HOMs species are not completely removed at higher $NO_x$ seen in some laboratory experiments (e.g., Zhao et al. (2018); Mehra et al. (2020)). However, the only rate constant for a highly-oxidised $RO_2$ radical with NO that has been measured (Berndt et al., 2015) was found to be ~3–4 times faster than the rate constant used in the present work.

The simultaneous measurement of $RO_x$, NO, and $NO_2$ allowed for the calculation of *in situ* ozone production rates (Figure S8). Using FAGE-measured radicals, daily integrated (06:00–21:00 UTC) ozone productions of 38 and 80 ppbv in NW–SE and SW air, respectively, were calculated. The daily integrated ozone productions calculated from MCM-modelled radicals are much lower at 9 and 15 ppbv, respectively. These values may be compared to those calculated from measured ozone using equation (E14) at 18 and 20 ppbv, respectively. The large difference between FAGE-calculated ozone production and that

calculated from measured ozone suggests that most of the ozone produced *in situ* was transported downwind of the WAO site. Daytime ozone production was shown to be close to the transition between $NO_x$-limited and VOC-limited regimes in NW–SE air, which may be considered representative of the background conditions of northern Europe. Ozone production was generally VOC-limited in the more polluted SW air, although still relatively close to the transition point (Figure 14). It was also shown that FAGE-calculated ozone production rates scaled with NO in both wind sectors (Figure 15). Taken together, these results

suggest that both $NO_x$ and VOC emissions reductions in source regions (e.g., London, Birmingham) would help to mitigate ozone pollution at this UK coastal receptor site.

The results of our work may be compared to those of Tan et al. (2019), who first used the experimental budget approach for a campaign in the Pearl River Delta (PRD), China. Pollution levels were much higher during the PRD campaign compared to those encountered at the WAO – for example much greater OH reactivities of up to 80 s$^{-1}$ were measured (c.f. 18 s$^{-1}$ for

ICOZA), and NO mixing ratios were higher (diurnal maximum of ~4 ppbv versus ~0.8–1.4 ppbv for ICOZA). Despite this, measured radical concentrations were fairly similar, with maximum diel median concentrations of $4.5 \times 10^6$ molecule cm$^{-3}$ for OH (c.f. 2–4 × 10$^6$ molecule cm$^{-3}$ during ICOZA), $3 \times 10^8$ molecule cm$^{-3}$ for $HO_2$ (c.f. 1–1.5 × 10$^8$ molecule cm$^{-3}$), and 2 × 10$^8$ molecule cm$^{-3}$ for $RO_2$ (c.f. ~5 × 10$^8$ molecule cm$^{-3}$). In the PRD, maximum loss rates for OH, $HO_2$, and $RO_2$ reached up to 10–15 ppbv h$^{-1}$, similar to the loss rates observed for $RO_2$ in SW air during ICOZA (Figure 10). The loss rate of total $RO_x$

peaked at midday at ~3 ppbv h$^{-1}$, compared with ~0.8–1.2 ppbv h$^{-1}$ for ICOZA (Figure 7), where the difference is likely due to the higher pollution levels found in the PRD (i.e., increased radical loss to $NO_x$). Within experimental uncertainties, the $RO_x$ budget was balanced, similar to that observed for ICOZA. Evidence for a missing afternoon OH source was presented (with an inferred source strength of 4–6 ppbv h$^{-1}$), which was also the case for NW–SE air during ICOZA (up to ~2 ppbv h$^{-1}$, Figure 8). However, in the PRD, the $HO_2$ budget was closed within experimental uncertainty, and the closure of the $RO_2$ budget could

be greatly improved when the rate of $RO_2$ production was calculated from measured OH reactivity, although a missing

afternoon $RO_2$ sink was still present. This is in contrast to our results, from which a significant missing $HO_2$ sink (Figure 9) and a missing $RO_2$ source (Figure 10) on the order of 10 ppbv h$^{-1}$ may be inferred. In the PRD, the strongest differences between calculated $RO_2$ production and destruction rates were found at low NO (<1 ppbv), with budget closure at high NO. However, during ICOZA, the difference between $RO_2$ (and $HO_2$) production and destruction rates was most severe at high NO (Figure S10).

More recently, Whalley et al. (2020) also assessed the experimental radical budget for $RO_x$ and OH reactivity observations made in summertime Beijing. A missing OH source was identified under the low NO (<0.5 ppbv) conditions experienced in the afternoon, similar to that for ICOZA NW–SE air, but with a much higher inferred source strength on the order of ~15 ppbv h$^{-1}$. Identical to ICOZA, their budget analysis indicated that the $HO_2$ and $RO_2$ budgets were both out of balance but with opposite sign, where the ratios of production to destruction rates displayed a strong dependence on NO concentration; under the highest NO (~100 ppbv) conditions, $P(HO_2)$ exceeded $D(HO_2)$ by ~50 ppbv h$^{-1}$ (cf. ~10–15 ppbv h$^{-1}$ for ICOZA at ~2 ppbv NO), whilst $D(RO_2)$ exceeded $P(RO_2)$ by the same magnitude. The agreement between experimental production and destruction rates for $HO_2$ and $RO_2$ was much improved after reducing the rate of $RO_2 \rightarrow HO_2$ propagation (by reducing $\alpha$ from 0.95 to 0.10), similar to our approach of reducing the $RO_2$ + NO rate constant (Figure S11). Whalley et al. (2020) suggested that some complex $RO_2$ species (e.g., from BVOC or aromatic VOC oxidation) do not directly generate $HO_2$ after reaction with NO, but instead the RO radicals formed autoxidise (via H-shifts) to form new $RO_2$ species that undergo further reaction with NO before eventually forming $HO_2$. This type of chemistry serves to reduce the rate of $RO_2 \rightarrow HO_2$ propagation and could help to explain the differences between experimental production and destruction rates of $HO_2$ and $RO_2$ found for ICOZA.

## 5. Conclusions

OH, $HO_2$, and $RO_2$ radicals and OH reactivity ($k'_{OH}$) were measured at a UK coastal receptor site during the July 2015 ICOZA intensive field campaign. Maximum measured daily OH, $HO_2$, and total $RO_2$ radical concentrations were in the range 2.6–17 $\times 10^6$, 0.75–4.2 $\times 10^8$, and 2.3–8.0 $\times 10^8$ molecule cm$^{-3}$, respectively. $k'_{OH}$ ranged from 1.7 to 17.6 s$^{-1}$ with a median value of 4.7 s$^{-1}$. ICOZA data were split by wind direction to assess differences in the radical chemistry between air that had passed over the North Sea (NW–SE sectors) or over major urban conurbations such as London (SW sector). A PSS calculation underpredicted daytime OH in NW–SE air by ~35% on average, whereas agreement was found within instrumental uncertainty (~26% at $2\sigma$) in SW air. The OH levels predicted by a box model using MCM chemistry were in better agreement with the measurements. However, for $HO_2$, the base MCM model overpredicted the observations in NW–SE air in the afternoon by a factor of ~2–3, whereas reasonable agreement was found for $HO_2$ in SW air when the model was constrained to measured carbonyls (HCHO, MVK+MACR). In contrast, for total $RO_2$, the model severely underpredicted the observations in both NW–SE and SW air, with measurement/model ratios ranging from ~2–5 in the afternoon to almost 40 in the early morning. The model predicted that the dominant $RO_2$ species in both wind sector types was $CH_3O_2$. $k'_{OH}$ observations were underpredicted by ~34% and ~37% in NW–SE and SW air, respectively.

Measured OH levels were well correlated with $J(O^1D)$, where power law fits to measured OH versus $J(O^1D)$ yielded power terms similar to previous coastal field campaigns. Good correlations were also observed between measured total $RO_2$ and measured $HO_2$, and the fit slopes indicated that the $RO_2$:$HO_2$ ratio was close to 1:1 in NW–SE air and ~2:1 in the more polluted SW air. The slopes of modelled $RO_2$ versus modelled $HO_2$ were different between day and nighttime data, which was not seen in the observations.

Measured radical and $k'_{OH}$ levels and measurement-to-model ratios displayed strong dependences on NO mixing ratios. For OH, the PSS calculation could capture the observations at high NO (> 1 ppbv), but underpredicted the observations at low NO (< 200–300 pptv) by a factor of ~2–3, suggesting missing OH sources. The MCM-base model performed better in terms of reproducing the observed dependence of OH on NO but there was still a tendency towards underprediction at low NO. The MCM-base model overpredicted $HO_2$ concentrations at low NO in NW–SE air by a factor of ~3, whereas in SW air, the measurements and model results were in agreement across the full NO range. For $RO_2$, measurement-to-model ratios scaled with NO, from ~2–3 for NO < 100 pptv to ~10–30 for NO > 1 ppbv, a trend found in all previous field campaigns in which $RO_2$ was measured using the $RO_x$LIF technique. This suggests that peroxy radical chemistry is not well understood under high $NO_x$ conditions. Missing OH reactivity, i.e., the difference between measured and modelled $k'_{OH}$, also scaled with NO. The strong correlation of missing OH reactivity with several OVOCs suggests that the missing reactivity was due to unmeasured VOC oxidation products that were not well simulated by the model, rather than a primary VOC species (e.g., a BVOC).

The simultaneous measurement of OH, $HO_2$, $RO_2$, and $k'_{OH}$ allowed for experimental (i.e., observationally determined) budgets to be derived for all radical species as well as total $RO_x$. Data were separated according to wind direction: SW winds (180°–270°), and all other winds (NW–SE, <165° and >285°). In NW–SE air, the $RO_x$ budget could be closed during the daytime within experimental uncertainty but OH destruction exceeded OH production by ~2–3 ppbv h$^{-1}$, and $HO_2$ production greatly exceeded $HO_2$ destruction while the opposite was true for $RO_2$. In SW air, the $RO_x$ budget analysis indicated missing daytime $RO_x$ sources on the order of ~0.2–0.6 ppbv h$^{-1}$ but the OH budget was balanced, and the same behaviour was found with the $HO_2$ and $RO_2$ budgets as in NW–SE air. Differences between radical destruction and production rates were found to exhibit species-dependent trends with respect to NO mixing ratios; the budget imbalances were most severe for $HO_2$ and $RO_2$ at high NO (> 1000 pptv), with differences of –(12–15) ppbv h$^{-1}$ and +(13–16) ppbv h$^{-1}$, respectively.

In NW–SE air, the dominant daytime $RO_x$ sources were $O^1D$ + $H_2O$ and HONO photolysis (~37% each) with significant contributions from carbonyl photolysis (~23%), while the major $RO_x$ sinks were the reactions $RO_2$ + NO → $RONO_2$ (~28%), $RO_2$ + $HO_2$ (~33%), and OH + $NO_2$ (~33%). The major OH source was the secondary source $HO_2$ + NO (~50%) with significant contributions from $O^1D$ + $H_2O$ and HONO photolysis (~23% each), while in terms of OH loss, the most important reactions were OH + CO (~42%) and OH + $NO_2$ (~20%). In the more polluted SW air, $RO_x$ initiation was dominated by HONO photolysis (~44%) with similar contributions from $O^1D$ + $H_2O$ (~29%) and carbonyl photolysis (~25%), while $RO_x$ termination was mainly controlled by the reactions $RO_2$ + NO → $RONO_2$ (~38%) and OH + $NO_2$ (~39%). The rate of OH production was dominated by $HO_2$ + NO (~70%), while OH loss was controlled by reactions with CO (~27%), $NO_2$ (~26%), and carbonyls (~21%).

After finding that the radical budgets were out of balance, most severely for $HO_2$ and $RO_2$, several modifications were made to the calculation of experimental budgets to try and reconcile this: (1) the addition of generic radical recycling processes, (2) reduction of the rate of $RO_2 \rightarrow HO_2$ conversion, (3) inclusion of heterogeneous $HO_2$ uptake, and (4) addition of chlorine chemistry. The best agreement between $HO_2$ and $RO_2$ production and destruction rates was found for (2), in which we reduced the $RO_2 + NO$ rate constant by a factor of 5. It is therefore recommended that more studies are conducted to measure $RO_2 + NO$ rate constants, in particular for more complex, functionalised $RO_2$, and to explain a lower-than-expected $RO_2$-to-$HO_2$ propagation rate, further study the fate of RO radicals is also recommended, particularly those which may be involved in autoxidation.

The rate of *in situ* ozone production ($P(O_x)$) was calculated from observations of $RO_x$, NO, and $NO_2$ and compared to that calculated from MCM-modelled radical concentrations. The MCM-calculated $P(O_x)$ significantly underpredicted the measurement-calculated $P(O_x)$ in the morning by up to a factor of 10, and the degree of underprediction was found to scale with NO. Using the ratio of the rates of radical loss to $NO_x$ to total radical loss ($L_n / D(RO_x)$), it was shown that in NW–SE air, daytime ozone production was close to the transition between $NO_x$-limited and VOC-limited regimes. However, in the more polluted SW air, ozone production was generally VOC-limited.

The strong NO-dependences of the $HO_2$ and $RO_2$ budget imbalances reveal a systematic limitation to our understanding of peroxy radical cycling chemistry, which directly impacts our ability to calculate ozone production rates correctly. Future tropospheric ozone abatement strategies rely on the accurate simulation of ozone chemistry. It is therefore crucial that further studies seek to explain the budget imbalances found in this work.

## Data availability

The data used in this study are available from the corresponding authors upon request (l.k.whalley@leeds.ac.uk and d.e.heard@leeds.ac.uk) and are also archived on CEDA (https://archive.ceda.ac.uk/).

## Author contributions

WJB was the principal investigator of the ICOZA project and was responsible for organisation of the Weybourne field intensive. RWM, LKW, DRC, TI, and DEH were responsible for measurements of radicals, OH reactivity, HCHO, and photolysis frequencies (*J* values also provided by RS, SMB, and PSM). LRC, LJK, and WJB made measurements of HONO and aerosol surface area. RS, SMB, and PSM provided $Cl_2$ and $ClNO_2$ data. JL and CR measured NO, $NO_2$, and HONO. BJB was responsible for the long-term operation of the Weybourne Atmospheric Observatory and provided $O_3$, CO, and HCHO data. GLF was responsible for measurements of VOCs. RS and SC developed the AtChem modelling framework. RS

conducted the MCM model simulations. RWM, LKW, and RS analysed the data. RWM wrote the manuscript with input from all co-authors.

## Acknowledgements

We thank the science team of the ICOZA project. RWM and DRC are grateful to the NERC for funding PhD studentships. RWM, LKW, DRC, TI, and DEH would like to thank the University of Leeds electronic and mechanical workshops. RWM is grateful to Hans Osthoff (University of Calgary) for the provision of Igor functions, and to Chunxiang Ye (Peking University) and Samuel Seldon (University Of Leeds) for useful discussions. We thank Lloyd Hollis and Roland Leigh (University of Leicester) for assistance with the spectral radiometer and chemical ionization mass spectrometer. We thank Sam Cox for his help with the AtChem modelling framework.

## Financial support

This research has been supported by the NERC (grant nos. NE/K012029/1, NE/K012169/1, and NE/K004069/1).

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

 **Tables**

**Table 1.** List of species observed and their corresponding measurement techniques for the ICOZA campaign. For descriptions of simple and complex $RO_2$, see Section 2.1.2. For some species (e.g., NO, $NO_2$, HONO, HCHO) more than one measurement technique was available.

| Observation(s) | Technique | Sampling platform | Institution | Reference(s) |
|---|---|---|---|---|
| OH, $HO_2$, total $RO_2$, "simple" and "complex" $RO_2$ | Fluorescence assay by gas expansion (FAGE) | FAGE container roof | Leeds | Whalley et al. (2013); Whalley et al. (2018); Woodward-Massey et al. (2020) |
| OH reactivity | Laser flash photolysis–laser-induced fluorescence spectroscopy (LFP-LIF) | FAGE container roof | Leeds | Stone et al. (2016) |
| $J(O^1D)$ | Filter radiometry | FAGE container roof | Leeds | Bohn et al. (2016) |
| Photolysis frequencies | Spectral radiometry (two instruments) | FAGE and Leicester containers | Leeds/Leicester | Bohn et al. (2008) |
| HCHO | Laser-induced fluorescence (LIF) | WAO manifold | Leeds | Cryer (2016) |
| HONO | Long path absorption photometry (LOPAP) | Birmingham van roof | Birmingham | Heland et al. (2001); Crilley et al. (2021) |
| Aerosol surface area | Aerodynamic particle sizer (APS) | Birmingham van roof | Birmingham | Chen et al. (1985) |
| $Cl_2/ClNO_2$ | Chemical ionisation mass spectrometry | Leicester container roof | Leicester | Sommariva et al. (2018) |
| NO ($NO_2$) | Chemiluminescence (LED $NO_2$ converter) | WAO roof | York | Lee et al. (2009a) |
| $NO_2$ | Cavity-attenuated phase-shift spectroscopy (CAPS) | WAO manifold | York | Kebabian et al. (2008) |
| HONO | Differential photolysis with chemiluminescence detection of NO | WAO roof | York | Reed et al. (2016) |
| $O_3$ | UV absorption | WAO manifold | UEA | - |
| CO | MgO reduction with UV detection | WAO manifold | UEA | Robbins et al. (1968) |
| HCHO | Hantzsch colourimetry | WAO manifold | UEA | Nash (1953) |
| VOCs (up to $C_6$ alkanes/alkenes, acetylene, benzene, toluene) | Gas chromatography with flame ionisation detection (GC-FID) | WAO roof | UEA | - |
| VOCs ($C_8/C_9$ aromatics, $\Sigma$monoterpenes), OVOCs (methanol, acetaldehyde, acetone, acetic acid, MVK+MACR[a], MEK[b]), acetonitrile, DMS[c] | Proton transfer reaction–mass spectrometry (PTR-MS) | WAO roof | UEA | Murphy et al. (2010) |

[a]Sum of methyl vinyl ketone and methacrolein.

 [b]Methyl ethyl ketone.

[c]Dimethyl sulfide.

**Table 2.** Median daytime (defined as $J(O^1D) > 5 \times 10^{-7}$ s$^{-1}$, approximately 06:00–18:00 UTC) $RO_x$ source and sink contributions, split according to wind direction (NW–SE = <165° and >285°; SW = 180°–270°).

| $RO_x$ source | NW–SE (%) | SW (%) | $RO_x$ sink | NW–SE (%) | SW (%) |
|---|---|---|---|---|---|
| Ozonolysis | 3.1 | 2.8 | $RO_2 + NO \rightarrow RONO_2$ | 28.1 | 38.2 |
| $J_{carbonyls}$ | 22.9 | 24.5 | $HO_2 + HO_2$ | 1.8 | 0.9 |
| $J_{HONO}$ | 36.5 | 44.0 | $RO_2 + HO_2$ | 32.6 | 14.2 |
| $O^1D + H_2O$ | 37.5 | 28.7 | $RO_2 + RO_2$ | 1.5 | 1.1 |
| | | | $OH + NO_2$ | 32.6 | 39.4 |
| | | | $OH + NO$ | 3.4 | 6.2 |

**Table 3.** Median daytime OH source and sink contributions, split according to wind direction. OH reactivity contributions are derived from calculated OH reactivity, not measured. OH sink groupings are based on MCM classifications.

| OH source | NW–SE (%) | SW (%) | OH sink | NW–SE (%)[a] | SW (%)[b] |
|---|---|---|---|---|---|
| Ozonolysis | 0.8 | 0.5 | Aromatics[c] | 0.4 | 0.3 |
| $O^1D + H_2O$ | 23.2 | 11.5 | HONO[c] | 0.7 | 0.7 |
| $J_{HONO}$ | 22.6 | 17.7 | Methanol[c] | 1.7 | 3.1 |
| $HO_2 + O_3$ | 2.4 | 0.9 | Alkanes[c] | 2.5 | 1.6 |
| $HO_2 + NO$ | 51.0 | 69.4 | NO | 2.7 | 4.5 |
| | | | Unclassified[c] | 2.7 | 1.8 |
| | | | $O_3$[c] | 3.0 | 2.3 |
| | | | Dialkenes (isoprene + 1,3-butadiene) | 4.2 | 5.5 |
| | | | Alkenes + alkynes | 7.9 | 6.4 |
| | | | Carbonyls | 13.0 | 20.9 |
| | | | $NO_2$ | 19.5 | 26.2 |
| | | | CO | 41.6 | 26.7 |

[a]Median missing reactivity = 2.2 s$^{-1}$ (48% of measured).

[b]Median missing reactivity = 2.4 s$^{-1}$ (49% of measured).

[c]Lumped together as "others" in Figure 13.

**Table 4.** Previous measurements of OH, HO$_2$, sum of HO$_2$+RO$_2$ and OH reactivity at WAO and other selected locations in the MBL.

| Campaign, location and date | Concentration of NO | OH peak concentration / molecule cm$^{-3}$ | HO$_2$ peak concentration / molecule cm$^{-3}$ | HO$_2$+RO$_2$ peak concentration/ molecule cm$^{-3}$ | OH reactivity / s$^{-1}$ | References |
|---|---|---|---|---|---|---|
| ICOZA, WAO[a], June/July 2015 | mean 0.38 ppbv range ~0–7.5 ppbv | ~4 × 10$^6$ (noon) | 1 × 10$^8$ (~2 pm) | ~4-6 × 10$^8$ (noon) 2 × 10$^8$ (night) | 5.0 s$^{-1}$ (mean) 17.6 s$^{-1}$ (max) | This work |
| TORCH[b] 2, WAO, May 2004 | mean 0.62 ppbv range ~0–50 ppbv | ~4 × 10$^6$ (noon) | ~8 × 10$^7$ | | 4.9 s$^{-1}$ (mean) 9.7 s$^{-1}$ (peak) | Smith, 2007; Lee et al., 2009b |
| WAO, 1995 | | ~4–7 × 10$^6$ (measured using DOAS) | ~4–7 × 10$^6$ (measured using DOAS) | | | Forberich et al., 1999; Grenfell et al., 1999 |
| NAMBLEX[c], Mace Head, 2002 | Range 7-70 pptv | ~4 × 10$^6$ | 0.9–2.1 × 10$^8$ | | | Heard et al., 2006 |
| DOMINO[d], SW Spain, Nov–Dec 2008 | | | 1.5 × 10$^8$ | ~2–12 × 10$^8$ (day) ~20 × 10$^8$ (nighttime spikes) (PERCA) | | Van Stratum et al., 2012; Andres-Hernandez et al., 2013; Sinha et al., 2012 |
| Various at WAO, including INSPECTRO[f], Sept 2002 | | | | ~2–5 × 10$^8$ (day) ~5–7 × 10$^7$ (night) (PERCA[e]) | | Penkett et al., 1999; Fleming et al., 2006; Green et al., 2006 |
| EASE 97, Mace Heard, Apr-May 1997 | | | | ~1 × 10$^8$ (night) (PERCA) | | Salisbury et al., 2001 |
| TexAQS, Ship-based off coast near Houston 2006[g] | | | | ~30 × 10$^8$ (nighttime, polluted) ~2–10 × 10$^8$ (nighttime, clean) (PERCA) | | Sommariva et al., 2011 |
| Corsica, Summer 2013 | | | | | ~5 s$^{-1}$ (mean), ~17 s$^{-1}$ (max) with CRM[h] | Zannoni et al., 2017 |

[a] Weybourne Atmospheric Observatory
[b] Tropospheric ORganic photoCHemistry experiment
[c] North Atlantic Marine Boundary Layer Experiment
[d] Diel Oxidants Mechanisms In relation to Nitrogen Oxides
[e] peroxy radical chemical amplification
[f] Influence of clouds on the spectral actinic flux in the lower troposphere
[g] Texas Air Quality Study
[h] Comparative reactivity method

**Figure 1**. Time series of meteorological parameters (wind speed and direction, temperature, RH, photolysis frequencies) and trace gases (NO, NO₂, CO, HCHO, isoprene, MVK+MACR, O₃, HONO) measured during ICOZA (29th June – 23rd July 2015). All data presented are 15 min averages. UTC = Universal Time Coordinated.

**Figure 1.** Time series of meteorological parameters (wind speed and direction, temperature, RH, photolysis frequencies) and trace gases (NO, NO$_2$, CO, HCHO, isoprene, MVK+MACR, O$_3$, HONO) measured during ICOZA (29[th] June – 23[rd] July 2015). All data presented are 15 min averages. UTC = Universal Time Coordinated.

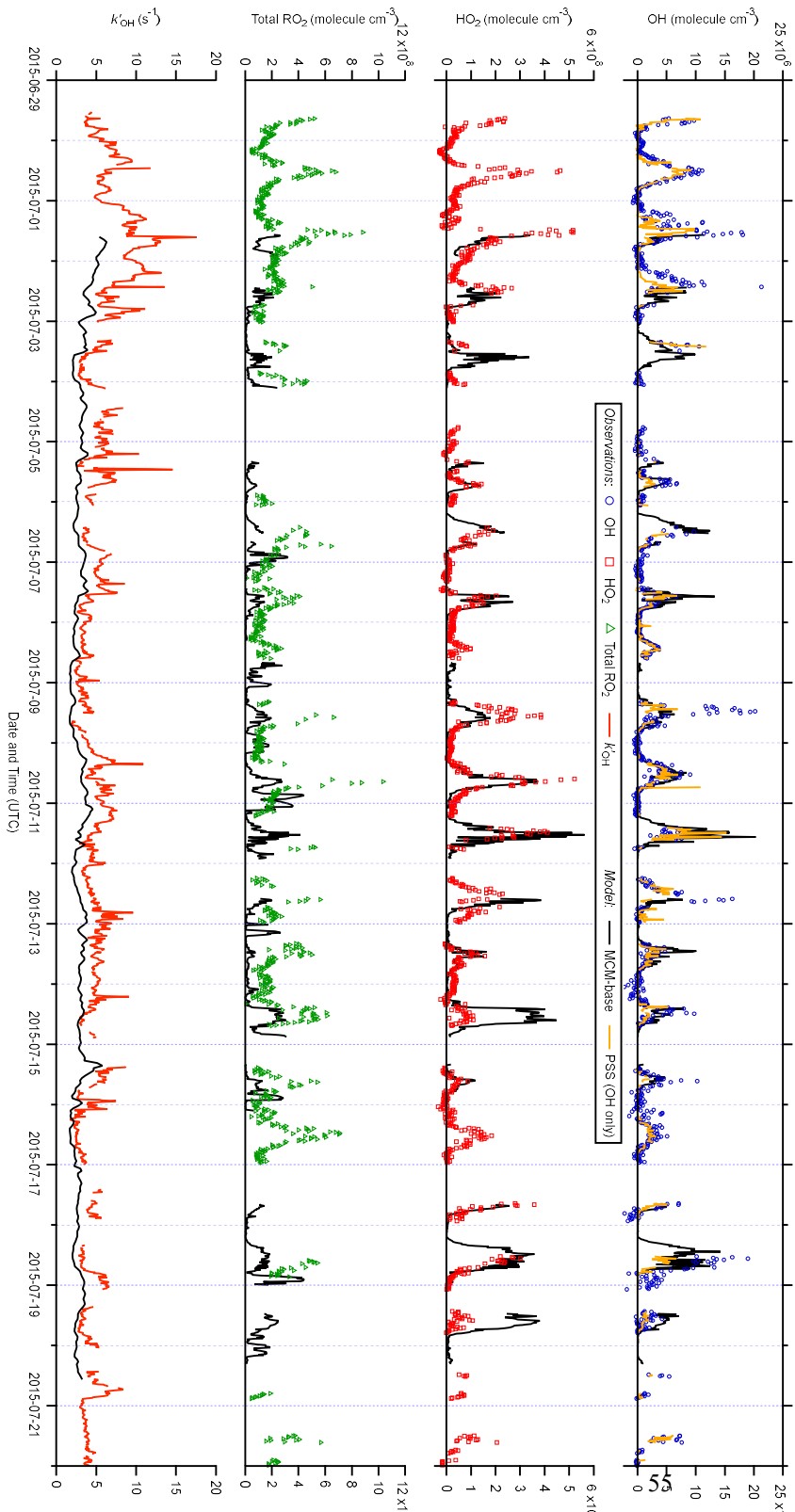

**Figure 2.** Time series of OH, HO$_2$, and total RO$_2$ measurements and comparison to MCM-base model and photostationary steady state (PSS) predictions. All data are at 15 min time resolution except for model OH reactivity (1 h). Error bars omitted for clarity. See Figure 1 for the time-series of wind direction.

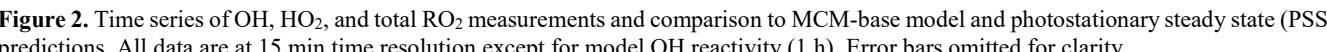

**Figure 2.** Time series of OH, $HO_2$, and total $RO_2$ measurements and comparison to MCM-base model and photostationary steady state (PSS) predictions. All data are at 15 min time resolution except for model OH reactivity (1 h). Error bars omitted for clarity.

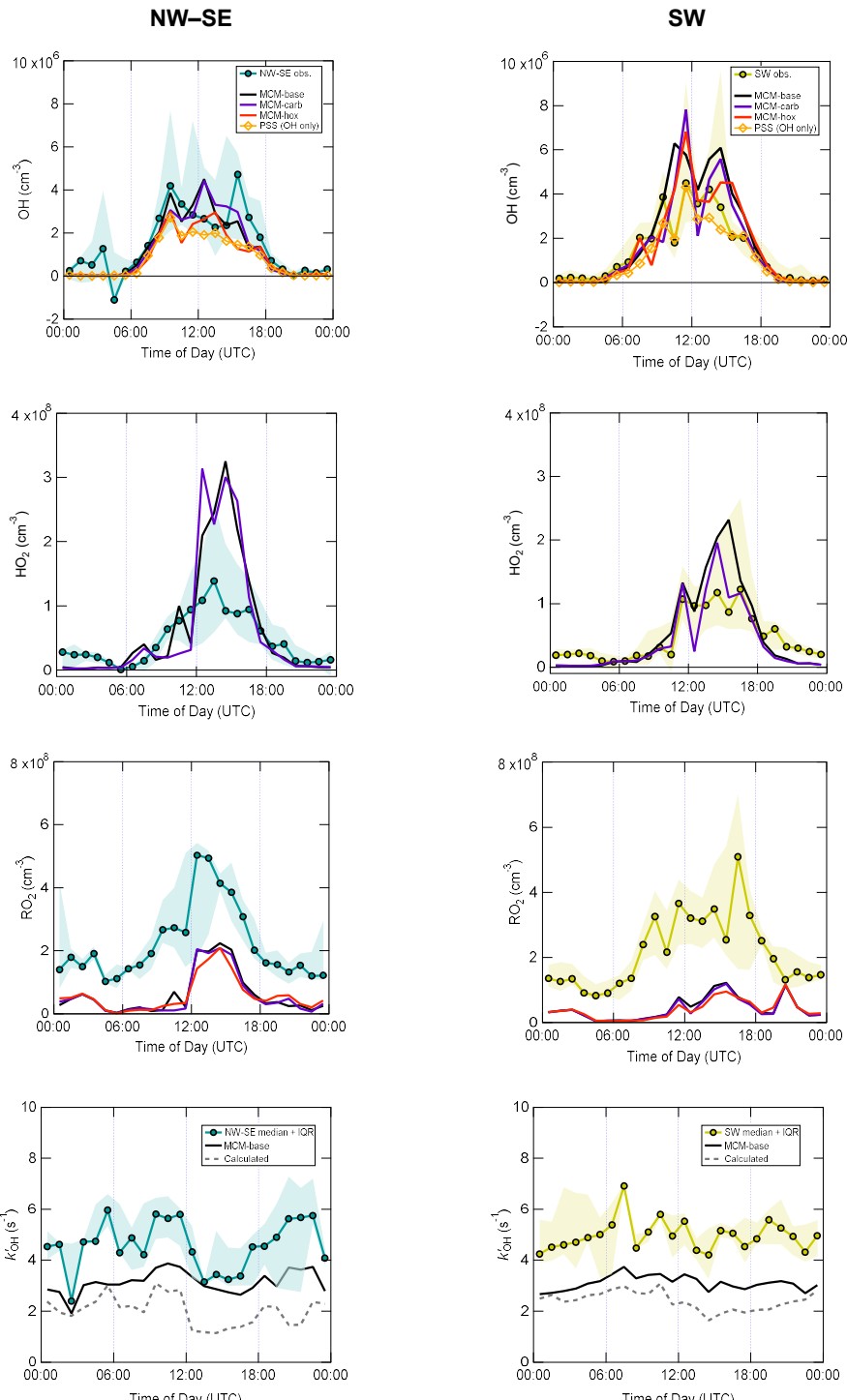

**Figure 3.** Hourly median diel profiles of OH, HO$_2$, total RO$_2$, and $k'_{OH}$ and comparison to MCM-base and PSS model predictions, split according to wind direction (left NW–SE, right SW). Shaded areas correspond to 25$^{th}$ and 75$^{th}$ percentiles of the data in each time bin.

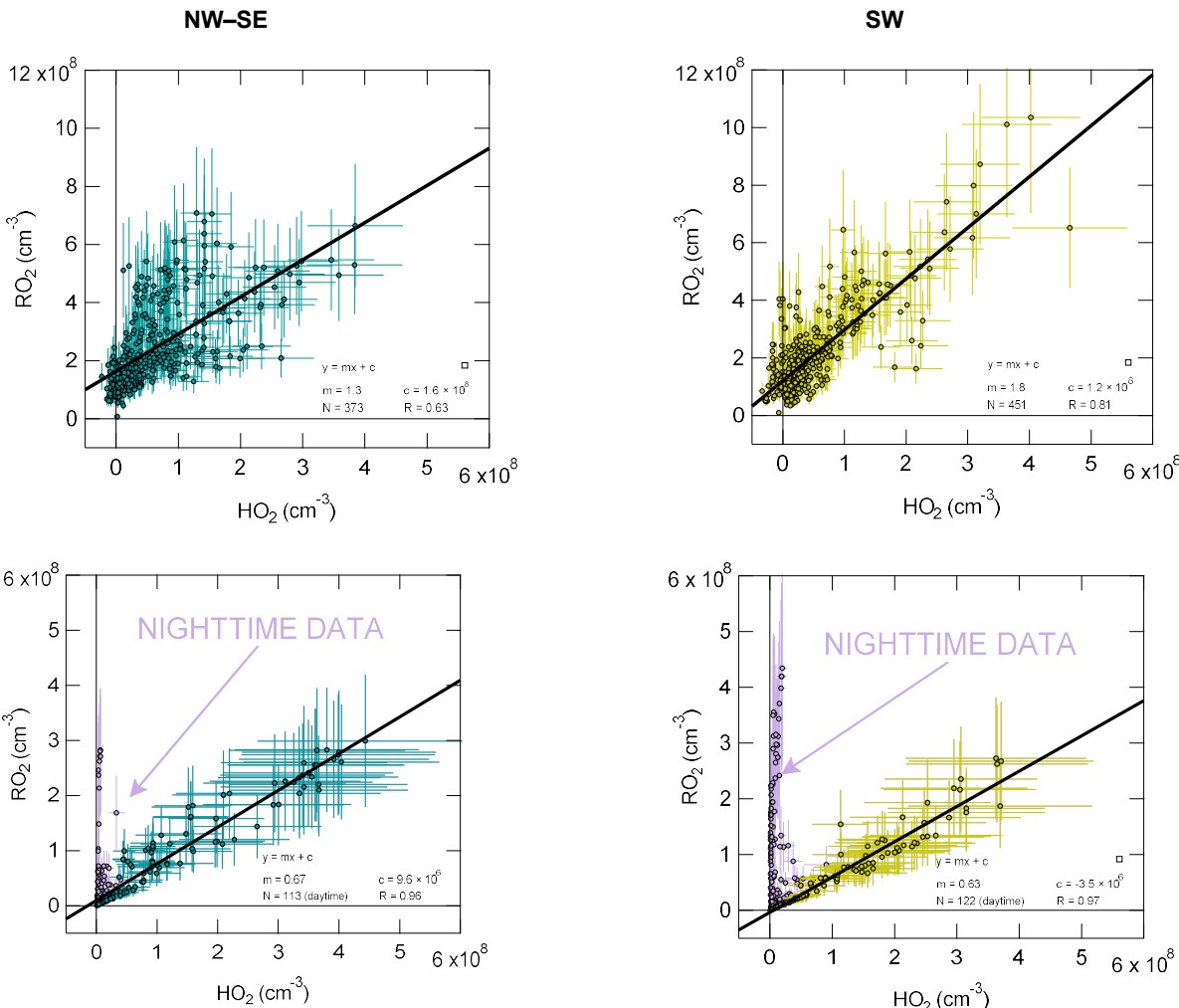

**Figure 4.** Observed total $RO_2$ versus observed $HO_2$ (top) and modelled total $RO_2$ versus modelled $HO_2$ (bottom), split according to wind direction (left NW–SE, right SW). Solid black lines correspond to linear least squares fits. For the model results, nighttime data exhibit a different $RO_2$ versus $HO_2$ slope, highlighted in purple; these data were not included in fits.

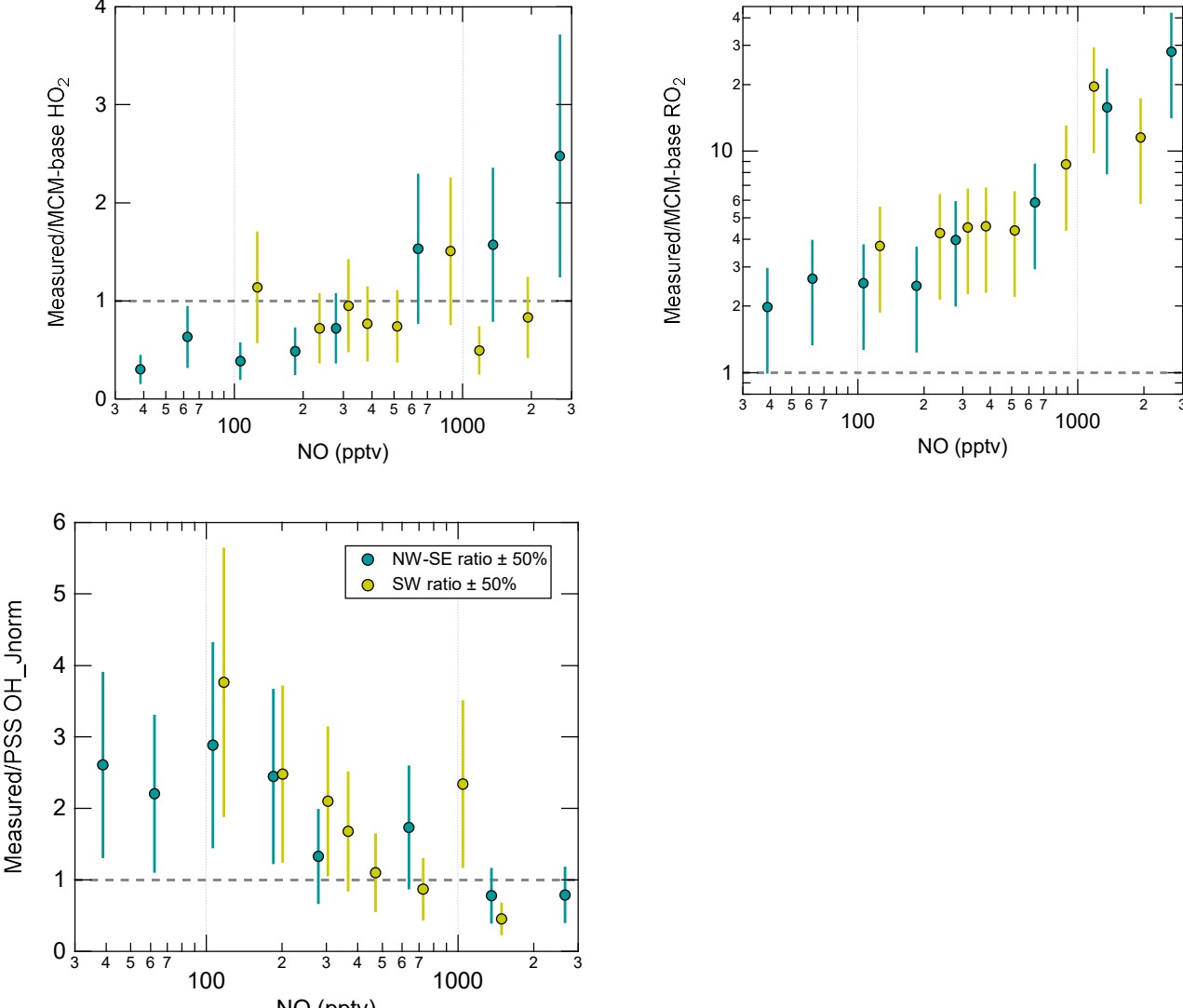

**Figure 5.** NO-dependence of the measurement-model ratios for radical species. Error bars correspond to an estimated combined measurement-model error of 50%. For OH, the reference model is the PSS calculation, and for HO$_2$ and RO$_2$ this is MCM-base. Note the $y$-log scale for RO$_2$.

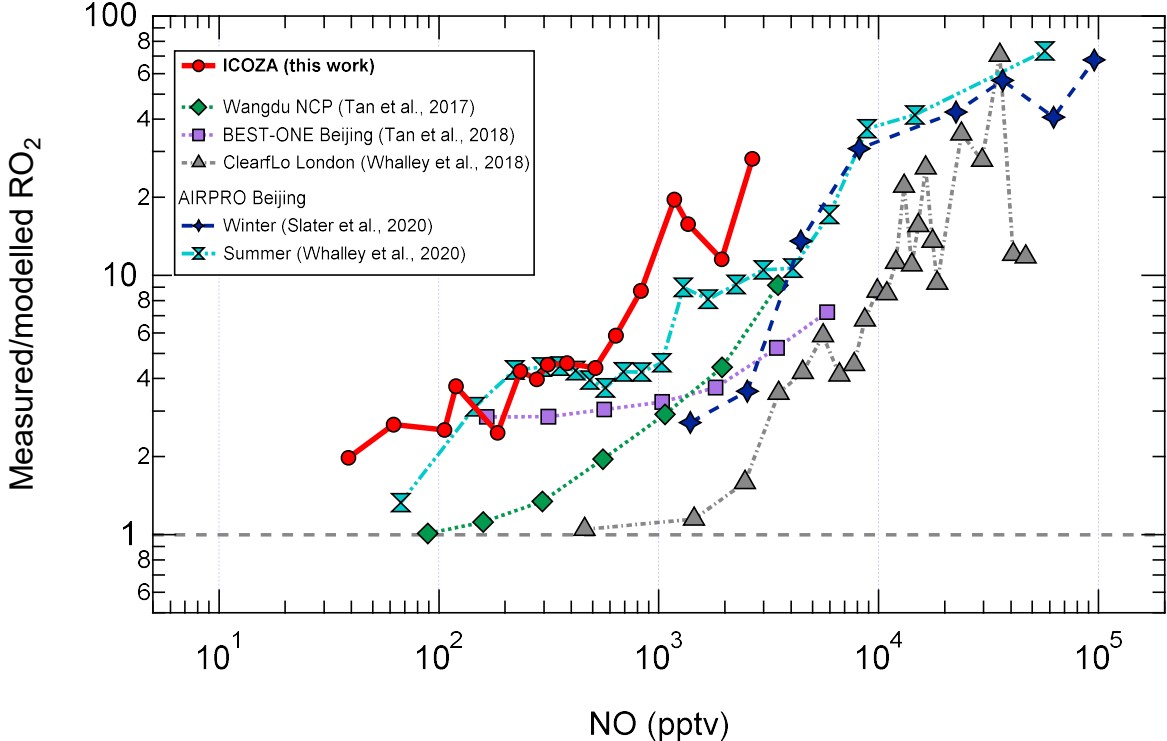

**Figure 6.** Comparison of RO$_x$LIF-measured RO$_2$ measurement-model ratios as a function of NO.

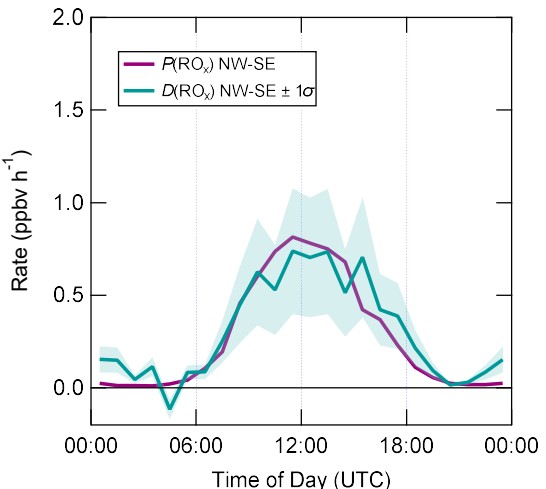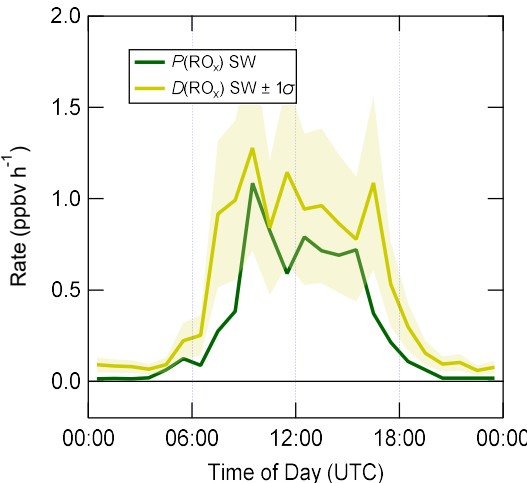

**Figure 7.** Median diel profiles of RO$_x$ production and destruction, split according to wind direction (NW–SE = <165° and >285°; SW = 180°–270°). Shaded area on $D$(RO$_x$) corresponds to the estimated 1$\sigma$ uncertainty of 35% (derived from calibration accuracy and reproducibility), not shown for $P$(RO$_x$) for clarity.

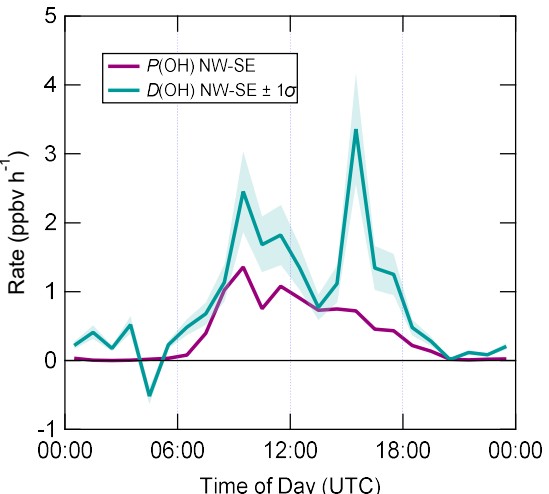 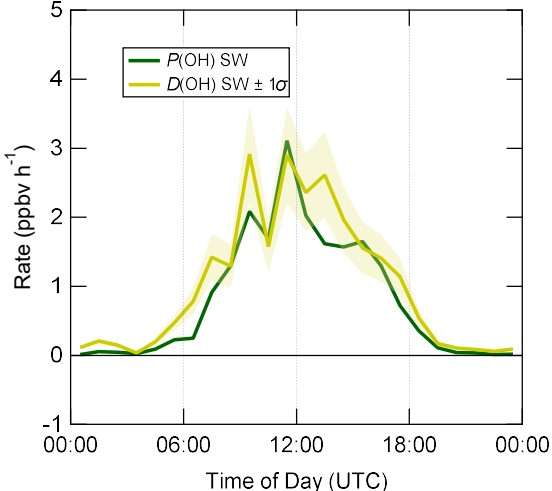

**Figure 8.** Median diel profiles of OH production and destruction, split according to wind direction. Shaded area on *D*(OH) corresponds to the estimated 1σ uncertainty of 24%, not shown for *P*(OH) for clarity.

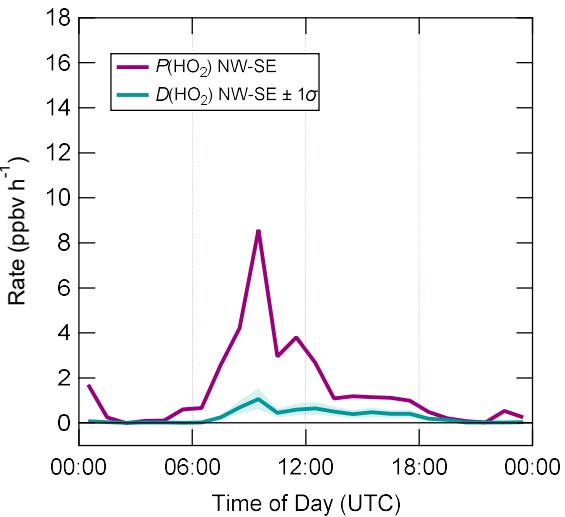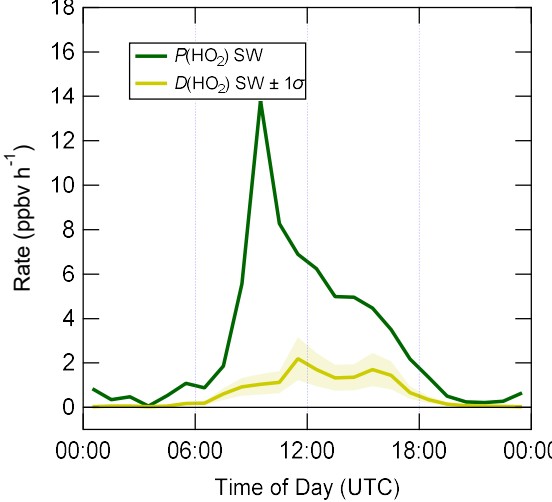

**Figure 9.** Median diel profiles of HO$_2$ production and destruction, split according to wind direction. Shaded area on $D$(HO$_2$) corresponds to the estimated $1\sigma$ uncertainty of 32%, not shown for $P$(HO$_2$) for clarity.

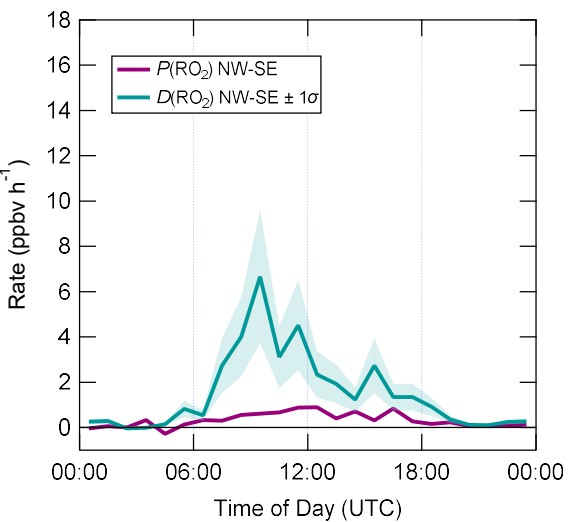 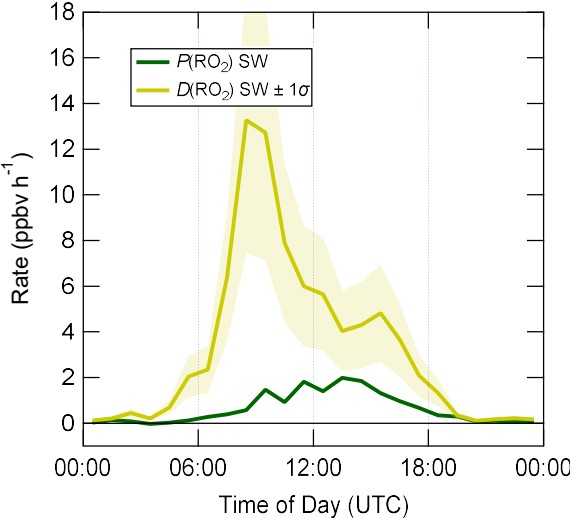

**Figure 10.** Median diel profiles of RO₂ production and destruction, split according to wind direction. Shaded area on $D(RO_2)$ corresponds to the estimated $1\sigma$ uncertainty of 32%, not shown for $P(RO_2)$ for clarity.

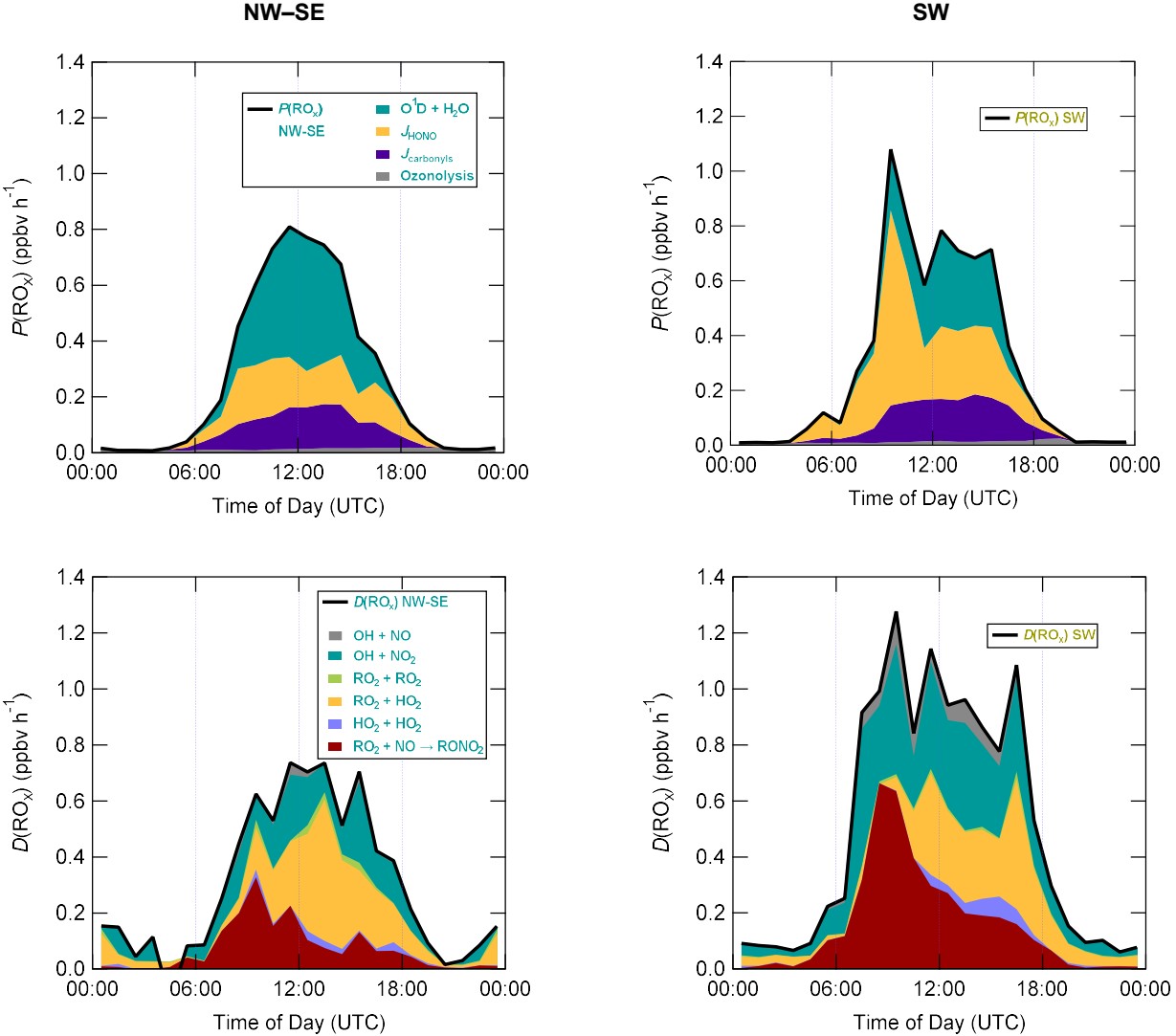

**Figure 11.** Median diel profiles of known RO$_x$ sources (top) and sinks (bottom), split according to wind direction. Average daytime contributions are given in Table 1. For interpretation of colours, please see the figure legend.

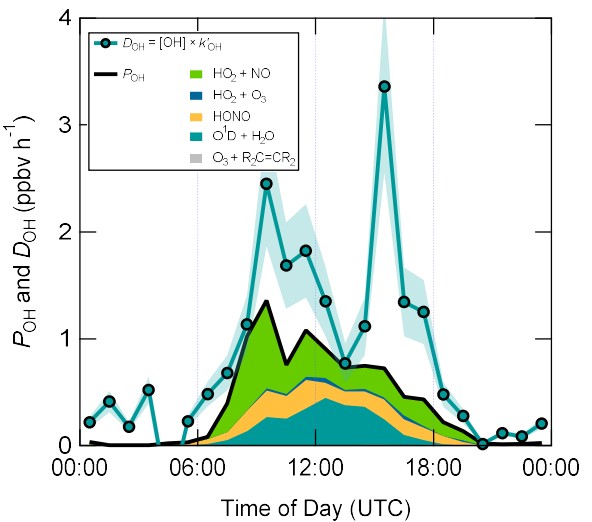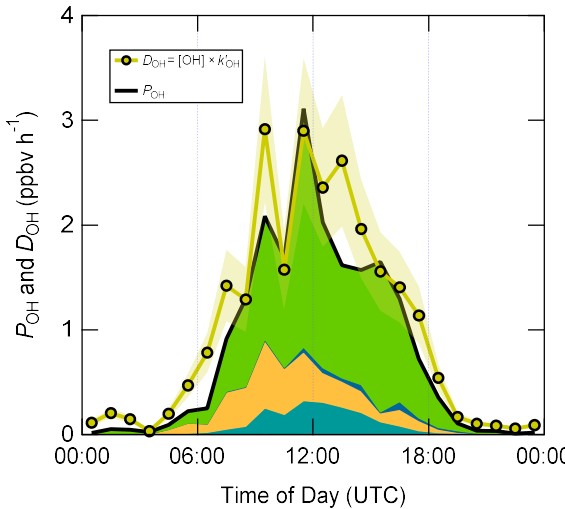

**Figure 12.** Median diel profiles of known OH sources and comparison to measured OH destruction, split according to wind direction (left: NW–SE, right: SW). Average daytime contributions are given in Table 2. For interpretation of colours, please see the figure legend.

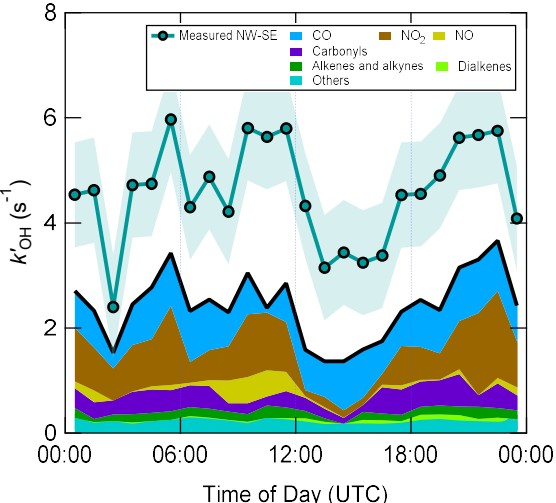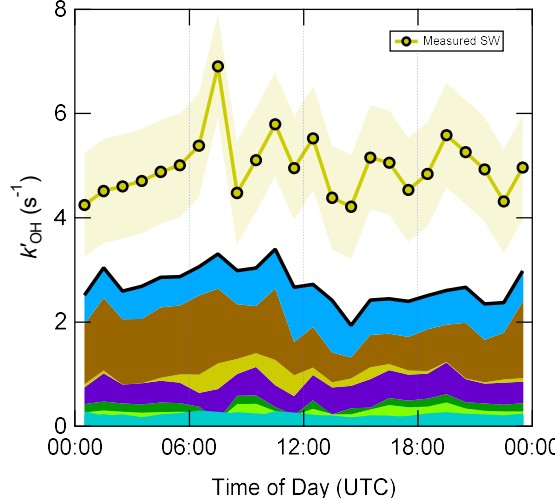

**Figure 13.** Median diel profiles of the OH reactivity calculated from measured reactants and comparison to measured OH reactivity, split according to wind direction. Average daytime contributions are given in Table 2. For interpretation of colours, please see the figure legend. Reactants in the "others" class are listed in Table 2. The shaded area on measured $k'_{OH}$ corresponds to the $1\sigma$ precision of ~1 s$^{-1}$. Model intermediates are not included here but their contributions are discussed in the text.

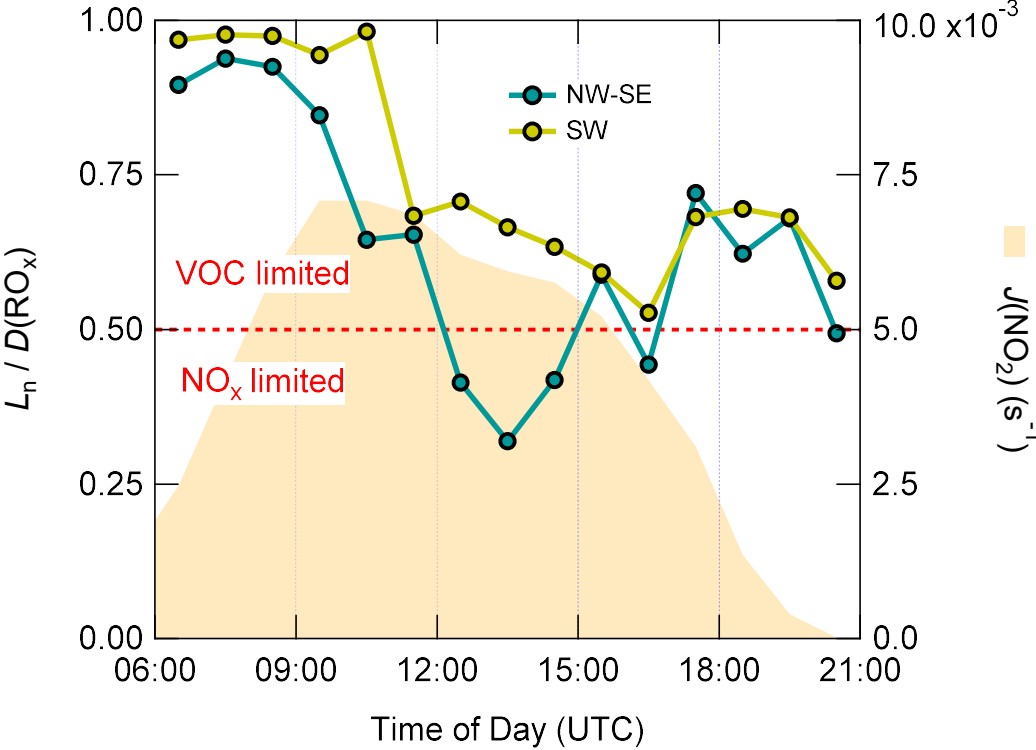

**Figure 14.** Daytime (06:00–21:00 UTC) ozone production regime based on the ratio $L_n$ / $D(RO_x)$ (left axis). $J(NO_2)$ is shown on the right axis. The dashed line corresponds to the transition between VOC and $NO_x$ limited ozone production ($L_n$ / $D(RO_x)$ = 0.5).

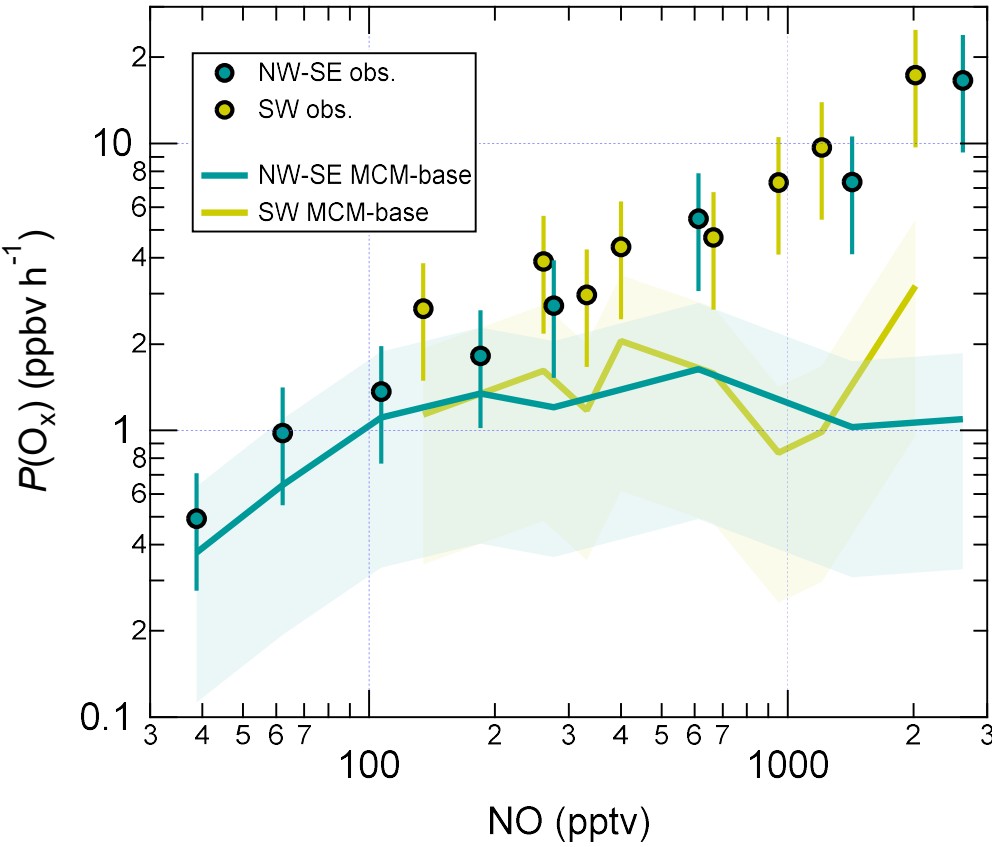

**Figure 15.** $P(O_x)$ as a function of NO for measured and MCM-base model $HO_2$ and $RO_2$. Error bars and shaded areas correspond to estimated $1\sigma$ uncertainties of 40% and 70% for measured and model $P(O_x)$, respectively. Note $y$- and $x$-log scales. $P(Ox)$ will be impacted by a change in the rate coefficient for $RO_2+NO$, owing to the change in $RO_2$ and $HO_2$ budgets, as shown Figure S11.

