# Peer review of "Radical chemistry and ozone production at a UK coastal receptor site"

_Atmospheric Chemistry and Physics, 2022_

## Referee Comment (RC1)

This paper presents the experimental budget analysis of OH, $HO_2$, $RO_2$, and the sum of all three radicals (ROx) at a site on the east coast of UK in summer 2015 as a part of the ICOZA campaign. The data was split into two subsets, i.e. SW and NW-SE, according to the wind sectors. The ROx budget was closed within experimental uncertainty, indicating no significant missing primary sources or termination processes. The OH destruction rate was slightly larger than its production rate, indicating a missing OH production process. The most severe imbalance occurred in the $HO_2$ and $RO_2$ budget analysis. The $HO_2$ production rate exceeded the destruction rate by the similar rate as the $RO_2$ destruction rate exceeding the production rate. Such imbalance elevated with NO mixing ratios.

To reconcile the imbalance in $HO_2$ and $RO_2$ budget, the authors performed several sensitivity tests. The most efficient case was reducing the $RO_2+NO$ reaction rate constant by a factor of 5. However, the change in the reaction rate constant contradict with literature reports.

The gross in situ ozone production rate was also calculated from observed and modelled radical concentrations. Large discrepancy was found in the ozone production rate derived from measured and modelled radical concentrations.

The results are of interest to the atmospheric chemistry community, to show key processes of ROx radical chemistry and unveil missing chemical processes. The paper is well structured and nicely written. I am in favor of publication after some of the issue being addressed. Also, the author may consider to concise the writing so that the main message(s) can be more prominent.

General comments:

1. The uncertainty analysis is missing. The authors stated the experimental uncertainty of radical destruction rates. But what about the production rates? Was it possible that the ROx budget in the SW air was also balanced if uncertainty of P(ROx) was considered. It would be useful to add one vertical bar at the time when the discrepancy was largest. The paper discussed the possible mechanism behind the experimental budget imbalance, which was fine. But the imbalance in the $HO_2$ and $RO_2$ budget could also be explained by

measurement interreference, for example, the $RO_2$ measurement interference. An evaluation of possible interreference and its contribution to close/enlarge the imbalance of radical budget should be added in the revised version to complete the discussion.

2. I would suggest to add the time series of radical budget in supplement at least. So the readers could see the variability of the experimentally determined budget.

3. The selection of the amounts of X, Y, Z looks quite arbitrary. It would be good to reconstruct the context to give the reason why such amounts of X, Y, Z were used. The amount of X was chosen to be 100 ppt as suggested in another paper, which conducted at a very different environment. The simplification of X mechanism was not really correct as the X species will have impact on OH, $HO_2$, and $RO_2$ radical budgets. The $RO_2 \rightarrow OH$ conversion is possible, such as $RO_2$ unimolecular isomerization, which should be nominated to a different term other than 'X'. Although I don't expect a big change to the $HO_2$ and $RO_2$ budget, it's more comparable to previous papers.

4. As the authors proposed that reducing the reaction rate of $RO_2$ to $HO_2$ propagation was the most likely explanation to reconcile the $HO_2$ and $RO_2$ budgets, should the P(Ox) also be calculated with the reduced reaction rate constant?

5. If $HO_2$ uptake and Cl chemistry was not important radical processes, I would suggest moving the relevant figures to the supplement and reduce the context further.

6. The first part of the discussion seems to be a repetition of what have been done and belongs to the conclusion. It should be shortened massively.

7. The comparison to previous works looks interesting and informative. Could it be able to summarize into a table with the three studies side by side?

8. The citations are not properly used. For example, Mehra et al. 2020, Slater et al. 2020, Whalley et al., 2020, the discussion version is cited while the final version is published. I also noticed the same problem occurs in the accompany paper. I do not have time go through the list, but I think mistake in the reference is not acceptable in any kind of scientific journals. Please carefully check all the reference list.

Technical comments:

1. Throughout the paper, it should be the rates of production and destruction. Most of the sentences miss the word 'rate(s)'.

2. Line 106. It's not a good argument that $H_2O_2$, ROOH and etc. were not measured so they are not considered in the radical budget. I think the contribution of these species is not important. Please give a rough estimate of these species and contribution to the radical budget first and say why they are not included.

3. Line 135. 'do not'

4. Line 212. Could you give an estimation on how much radical can be produced from $NO_3$+VOC?

5. Line 231. The number is different from the figure.

6. Line 247-250. I don't get the picture why to split the data into two sectors. How significant are these two air masses different? I cannot really see from the radical budget analysis.

7. Line 258. Be more specific on what is the difference in VOC.

8. Line 264-266. Better rephrase this sentence to be clearer.

9. Line 302-310. The contribution of different species is better scaled to measured $k_{OH}$ rather than calculated $k_{OH}$. The same applies to Table 2.

10. Line 306. 'Small but significant' sounds like a statistic phrasing, which may not fit in the current sentence.

11. Line 350. Is there a reason or reference to support this speculation?

12. Line 358. wrong citation format.

13. Line 403-404. Better to rephrase it to be clearer.

14. Line 408. '$HO_2$+NO' looks confusing.

15. Line 411 E13. The RONO2 formation should not be a $O_3/NO_2$ loss. I guess the authors are referring to PAN formation, i.e. $RO_2+NO_2$?

16. Line 430 and later on. 'FAGE-calculated' looks like a lab jargon. Maybe better to use derived from measurements or others.

17. Line 449-450. Not really needed this sentence.

18. Line 478-481. Better to rephrase it to be clearer.

19. Line 491. What does the 'complete' mean?

20. Line 502. How the ozone increase was calculated?

---

## Author Response (AR1)

Atmos. Chem. Phys. Discuss., referee comment RC1
https://doi.org/10.5194/acp-2022-207-RC1, 2022
**Comment on acp-2022-207**

Anonymous Referee #1

Referee comment on "Radical chemistry at a UK coastal receptor site - Part 1: observations of OH, HO₂, RO₂, and OH reactivity and comparison to MCM model predictions" by Robert Woodward-Massey et al., Atmos. Chem. Phys. Discuss., https://doi.org/10.5194/acp-2022-207-RC1, 2022

*We thank the referee for their careful reading of the manuscript and for their helpful suggestions. Our responses are given in italics below each comment, together with any changes to the text. The main change is that the two papers have been merged into one paper, and a lot of material has been moved into the SI.*

The authors report measurements of radicals at a coastal site in the UK. In general, campaigns with a full set of radical measurements are sparse, so that further exploration of radical concentrations in different chemical conditions are valuable. The paper has a companion paper investigating the chemical budgets of radicals. In this manuscript the authors focus on the description of measurements and model-measurement comparisons. A large part of the manuscript is very descriptive, also in the discussion part, which puts the results into the context of results from other campaigns reported literature. Little new results are shown in the sense of improving the understanding radical chemistry in the atmosphere. Therefore, this manuscript rather fits a measurement report instead of a research article. It should be considered to change the manuscript category.

*We have now merged the Part 1 and Part 2 companion papers into a single merged paper, with quite a bit of material moved to the SI. We feel that combining the results of the two papers does provide new understanding of the chemistry of the atmosphere over a range of NOx, and combines a unusually broad range of measurements of radicals (OH, HO2, RO2), OH reactivity with full supporting measurements to constrain the model and to quantify the radical budgets.*

The authors need to improve the manuscript by a concise writing. It is not clear, if separte papers for the measurements / model reszlts and chemical budget would have been required, if the authors had carefully planned to focus the writing of new findings and had cut on lengthy descriptions of figures that can be easily grasped by seeing the figures. In addition. there are some sections, in which it is not clear, if there is a deeper meaning of the analysis that is shown or if the analysis has only be done, because a similar analysis has been done in other papers and therefore, these sections could have been omitted. The manuscript as it is written

now clearly suffers from having the interpretation of model results and of the chemical budgets separated in 2 papers due to the close connection between both. Specifically the PSS calculations for OH shown in this paper is essentially the same as doing a chemical budget presented in the other paper. Merging the 2 papers would clearly be the best to present the insights into radical chemistry and likely also possible, because parts of the papers are similar, since the same data set is analysed and results from each paper is described in the other paper, and descriptive and unnecessary parts can be omitted.

*On reflection we feel that separate papers are not warranted for the radical measurements/model comparisons and the radical budget, for the reasons outlined by the referee. Hence we have merged the two papers, and with conciseness in mind, have reduced the material considerably looking for overlaps between the two papers, and have moved a lot of material into the SI to support the results in the main paper.*

The presentation quality of figures also needs considerable improvements. Font sizes in most of the figures contain are too small to be readable and scaling of axis are not appropriate. Light colours of text as used in the current figures are not suitable for reading (e.g. yellow).

*Where possible we have resized some of the figures to mitigate against this, and where possible altered the figures, and moved quite a few figures to the SI where the figures are presented at a considerably larger size.*

Additional specific comments:

L50: For this type of paper, just showing H-abstraction to form RO2 may oversimplifying the chemistry. Overall, some of the text-book like introduction may not be required.

*We have removed a considerable amount of the text-book like material, in that way it should then not be considered selective via omission of e.g. addition processes.*

L84: I assume that you mean "their photolysis can also be important radical sources"

*Yes that is what is meant. We have used the suggested rewording.*

L94: The conclusion in Novelli et al. is not that Criegee intermediates are the reason for interferences observed in the field, because reactant concentrations in their work were much higher than atmospheric concentration.

*We have removed this reason for the observed interferences.*

L144: Are you sure that the purity of NO was 99.95%? Typically the best purity that is available is only 99.5%.

*Checking the BOC specifications, the purity is now quoted as N2.8, which is 99.8%. This has been changed in the materials section.*

Fig. 2 It may be a good idea to improve visibility by splitting the figure into 2 panels by time.

*We feel that the data being shown all together provides an overview of the campaign in one panel. We have now positioned the figure so it is landscape on a single page so easier to see.*

L330: The statement about HONO is rather short and does not really reflect the high variability that is observed. On some days, values during the day were even higher than during the night.

*We agree that HONO is quite variable – and have added a statement to that effect.*

L384: Looking at the entire time series, the second peak that appears in the median diel profile looks more like an artefact of the median calculation than a real feature of the diel profile as it sounds in this statement.

*We agree, it is just a single point on the median diel profile. We have removed the sentence regarding the second peak.*

L411: I would avoid giving information in the text that is repeating what can be seen in the legend of the figure

*We have removed the repetition, and tried to do this where this may occur elsewhere.*

Fig 7: Here, it may make sense to have the same scale of the y-axis for sRO2 and cRO2.

*There is quite a difference between sRO2 and cRO2 (factor of 3) and we feel that some detail will be lost if the y-axes are the same.*

Fig. 8: Labels of the pie chart are not easy to read. Names may need further explanation in the figure caption. Numbers of fraction could be useful as well as the total median RO2 concentration.

*We have made these bigger and defined them in the caption (using the definitions in Section 3.2 of Paper 1). We have also given the total median RO2 in the caption for the 2 wind directions. We feel the fractions can be estimated by eye using the figure, and the numerical values are given in the text.*

L430 ff: Is the relative abundance of specific RO2 radicals consistent with measured OH reactants? This should be discussed.

*That is a good idea but it is very difficult for some RO2 because they can have multiple sources. Even for the simplest CH3O2, this may derive from several VOCs, not just CH4 but*

*following further degradation of larger RO2. We do indicate the major VOCs which the RO2 derive from, but in this section.*

Section 3.2: RO2: It looks like an offset between measured and modelled simple RO2. Can you exclude that there is an unaccounted instrumental offset?

*We can never completely exclude the possibility of an unaccounted instrumental offset, but our laboratory measurements for a range of VOCs (as described in Whalley et al., 2013) did not reveal any offset. We have assumed that all RO2 are converted to HO2 in the ROxLIF reactor and then to OH in FAGE fluorescence cell (following NO addition) with the same efficiency as CH3O2. If other RO2 species convert less efficiently, this would mean we are underpredicting the total RO. The only way that the measurement can overestimate RO2 is if something is decomposing to form RO2 in the instrument. – Fuchs et al. (2008) estimated HO2NO2 to contribute 1.7% of the measured HO2, CH3O2NO2 contributes 6% to RO2 (assuming an ambient NO2 concentration of 10 ppbv). For PAN, Fuchs et al. (2008) expect 0.1 pptv RO2 per ppbv PAN.*

*H. Fuchs, F. Holland and A. Hofzumahaus, Measurement of tropospheric RO2 and HO2 radicals by a laser-induced fluorescence instrument, Review of Scientific Instruments, 79, 084104 (2008).*

*Whalley, L. K., Blitz, M. A., Desservettaz, M., Seakins, P. W., and Heard, D. E.: Reporting the sensitivity of laser-induced fluorescence instruments used for $HO_2$ detection to an interference from $RO_2$ radicals and introducing a novel approach that enables $HO_2$ and certain $RO_2$ types to be selectively measured, Atmos. Meas. Tech., 6, 3425–3440, https://doi.org/10.5194/amt-6-3425-2013, 2013.*

Section 3.3: This analysis does not give much insights as it is done here. More discussion and comparison with previous findings with interpretation of different and similar results would be needed.

*There is a comparison with previous findings later in Section 4.1. We have signposted this from section 3.3. Also, combining the two papers will help – as the budget analysis for the radicals highlights processes that may be missing – which links to the model-measurement comparison.*

Section 3.4: Again, there is little interpretation or discussion of the correlation and it is not clear what is learned from this analysis. What is the meaning of the different slopes? What is expected for what reason?

*It is quite unusual to see plots of RO2 versus HO2 owing to the scarcity of simultaneously measured data so these correlation plots, and the corresponding ones for modelled values are quite novel. However, we agree there could be more discussion, for example regarding the modelled correlation plot at night, which is very different. Also, combining the two papers into one allows the budget analysis for HO2 and RO2, as RO2 degradation provides a source of HO2.*

L465: The offset does not necessarily indicate that some RO2 sources (= type of RO2 radicals) do not form HO2 as it sounds in the statement. It can also be that the lifetimes of HO2 and RO2 were much different or RO2 loss channels did not lead to HO2 formation, but the RO2 from all sources may still generally form HO2. If there were mainly RO2 sources in the night but little HO2 present, why would you expect that there is a correlation between RO2 and HO2, when the reaction of RO2+NO as most important pathway to HO2 is not relevant in the night?

*We agree and have incorporated the above into the discussion. There is a correlation but the gradient is much greater (RO2:HO2) at night suggesting nighttime RO2 converts to HO2 much less efficiently at night when NO is low, compared with the daytime. However, small amounts of RO2 will be converted to HO2, so a correlation still exists (either because there is a small amount of NO present or the RO2+RO2 self-reaction can form HO2).*

Section 3.6 / Figure 15: Would you expect that an exponential behaviour of BVOC emissions is visible for the range of temperature that is experienced in the campaign? Can you make an estimate, how much RO2 concentrations will change, if you assume additional VOCs in the model to account for the gap between measured and modelled OH reactivity?

*Over the range of T in the plot (13-24 degrees) we would not expect to see an exponential behaviour , but there clearly is a dependence. In the companion budget analysis paper, we discuss the additional RO2 that is made from missing OH reactivity, which is given by missing kOH/DRO2, where DRO2 is the destruction rate of RO2. As we have now merged the two papers, discussion of how RO2 concentrations will change is covered.*

Section 4.4: It would be good to have some numbers of reactive halogen species that would be required to explain observations.

*We state that for Mace Head during NAMBLEX where it was found that up to 40% of HO2 could be lost to IO under low-NOx conditions, for measured IO levels of 0.8–4.0 pptv 675 (Commane et al., 2011). There have been no measurements of IO, HOI or I2, nor BrO, at Weybourne, and there are no exposed sea-weed beds like at Mace Head, and so it is not expected that halogen levels are so high. However, the levels that were seen at Mace Head are the sort of concentrations that would be needed to explain the observations.*

[Figure]

Atmos. Chem. Phys. Discuss., referee comment RC2
https://doi.org/10.5194/acp-2022-207-RC2, 2022 ©
Author(s) 2022. This work is distributed under the
Creative Commons Attribution 4.0 License.

**Comment on acp-2022-207**
Anonymous Referee #2

Referee comment on "Radical chemistry at a UK coastal receptor site - Part 1: observations of OH, $HO_2$, $RO_2$, and OH reactivity and comparison to MCM model predictions" by Robert Woodward-Massey et al., Atmos. Chem. Phys. Discuss., https://doi.org/10.5194/acp-2022-207-RC2, 2022

*We thank the referee for their careful reading of the manuscript and for their helpful suggestions. Our responses are given in italics below each comment, together with any changes to the text. The main change is that the two papers have been merged into one paper, and a lot of material has been moved into the SI.*

This paper presents measurements of OH, HO2, RO2, and total OH reactivity at a coastal site during the 2015 ICOZA (Integrated Chemistry of OZone in the Atmosphere) campaign. The authors compare the measurements to predictions by both a photostationary state model as well as a zero-dimensional model based on the Master Chemical Mechanism (MCM 3.3.1). The authors find that in general the MCM model was able to reproduce the measured OH concentrations during the campaign, but overpredicted the measured concentrations of HO2 under lower NOx conditions when air arrived to the site from the northwest-southeast sectors, while underpredicting the measurements when more polluted air arrived to the site from the southwest sector. The authors also found that the model underpredicted the measured RO2 concentrations for both lower and higher NOx air that arrived from all sectors. The authors also find that the measured total OH reactivity was consistently greater than that calculated by the model.

The measurements described add to a growing dataset that suggest that our understanding of radical chemistry under a range of conditions may be incomplete, and as a result are of interest to the atmospheric chemistry community. The results are consistent with several previous measurements, and the authors suggest several possible reasons for the model discrepancies based on these previous results, including missing halogen chemistry and autooxidation of RO2 radicals reducing the rate of conversion to HO2 radicals. Unfortunately, the impact of these proposals on their model results are not included in this paper, as they are discussed in the companion paper. While the companion paper focuses on the impact of their proposed mechanisms on the radical budgets, this paper would benefit from some additional discussion of the impact of the proposed mechanisms on the modeled radical concentrations.

*We have now merged the Part 1 and Part 2 companion papers into a single merged paper, with quite a bit of material moved to the SI. In this way the impact of the proposed*

*mechanisms on both the modelled radical concentrations and also the radical budgets can be combined. There is certainly overlap on the impact on the modelled concentrations and the radical budgets.*

Specifically, the authors should consider including their model results when they reduced the rate of the RO2 +NO propagation rate as discussed in sections 4.4 and 4.5 as it appears that a reduction in this rate, perhaps due to the competition of RO2 autooxidation with radical propagation, improves the agreement with the measured HO2 and RO2 concentrations. While including an expanded discussion of the model results would add to an already lengthy manuscript, the authors should also consider condensing and or moving some of the discussion of previous measurements into a supplement.

*By combining the two papers into a single merged paper, we have been able to condense the material considerably, and have moved a significant amount of material to the SI. The reduced rate means that RO2 will have a longer lifetime and therefore the RO2 concentration will increase in the model, and slower HO2 production via RO2+NO means that the HO2 concentration will decrease – with autoxidation not expected to be a factor under the conditions at Weybourne.*

Additional comments:

1)  The authors conducted interference measurements during two different periods, finding that unknown interferences contributed less than 20% to the measured OH signal. It appears that these measurements occurred during both NW-SE and SW periods. Did the authors see a significant difference in the interference measurements from the different wind sectors?

*No, we did not see a significant difference from the different wind sectors.*

2)  The authors should consider highlighting the NW-SE and SW periods on Figure 5 to help illustrate the impact of the different air masses on the radical measurements.

*The top panel of Figure 2 shows the wind direction, from which the NW-SE and SW periods can be ascertained (either points high on the plot, or low on the plot). Fig. 5 is quite complex already, and we do not want to complicate it further, as there are quite a few changes in wind direction from NW-SE during the campaign. In the caption to Figure 5 we will point to Figure 2 for information about the time-series of wind direction.*

Given that the main focus of the paper is on the measurement/model discrepancy of the radical concentrations, there are several sections and figures in the paper that could be moved to a supplement to improve readability. In particular, sections 3.3 and 3.4 along with figures 9 and 10 could be moved to a supplement.

*We have merged the two companion papers, and moved significant sections into the SI. Included in the material moved to the SI are the sections mentioned above and the accompanying figures.*

3)  The authors could also condense much of the discussion of previous measurements by including a table summarizing the previous measurements/model agreement under the different NO conditions and referencing the table in the discussion.

*We have constructed a new table which summarises the previous measurements, whilst retaining key information in the text.*

4)  As mentioned above, the authors should consider adding the reduced RO2+NO model results to Figures 5-7 to illustrate how this model improves the agreement with the measurements. This illustration of the impact of the reduced rate is not included in the companion paper.

*A model with reduced RO2+NO was not run. Rather in the second paper the impact of reducing the RO2+NO on the budgets of RO2, HO2 and OH was considered. As the two papers have now been merged, we will merge together the material on reducing the RO2+NO rate, discussing the impact on the budget, and likely impact on their concentrations.*

[Figure]

Atmos. Chem. Phys. Discuss., referee comment RC1 https://doi.org/10.5194/acp-2022-213-RC1, 2022 © Author(s) 2022. This work is distributed under the Creative Commons Attribution 4.0 License.

**Comment on acp-2022-213**

Anonymous Referee #1

Referee comment on "Radical chemistry at a UK coastal receptor site - Part 2: experimental radical budgets and ozone production" by Robert Woodward-Massey et al., Atmos. Chem. Phys. Discuss., https://doi.org/10.5194/acp-2022-213-RC1, 2022.

*We thank the referee for their careful reading of the manuscript and for their helpful suggestions. Our responses are given in italics below each comment, together with any changes to the text. The main change is that the two papers have been merged into one paper, and a lot of material has been moved into the SI.*

This paper presents the experimental budget analysis of OH, HO2, RO2, and the sum of all three radicals (ROx) at a site on the east coast of UK in summer 2015 as a part of the ICOZA campaign. The data was split into two subsets, i.e. SW and NW-SE, according to the wind sectors. The ROx budget was closed within experimental uncertainty, indicating no significant missing primary sources or termination processes. The OH destruction rate was slightly larger than its production rate, indicating a missing OH production process. The most severe imbalance occurred in the HO2 and RO2 budget analysis. The HO2 production rate exceeded the destruction rate by the similar rate as the RO2 destruction rate exceeding the production rate. Such imbalance elevated with NO mixing ratios.

To reconcile the imbalance in HO2 and RO2 budget, the authors performed several sensitivity tests. The most efficient case was reducing the RO2+NO reaction rate constant by a factor of 5. However, the change in the reaction rate constant contradict with literature reports.

The gross in situ ozone production rate was also calculated from observed and modelled radical concentrations. Large discrepancy was found in the ozone production rate derived from measured and modelled radical concentrations.

The results are of interest to the atmospheric chemistry community, to show key processes of ROx radical chemistry and unveil missing chemical processes. The paper is well structured and nicely written. I am in favor of publication after some of the issue being addressed. Also, the author may consider to concise the writing so that the main message(s) can be more prominent.

*We have merged the two companion papers, and moved a considerable amount of material into the SI, and this has led to a more concise delivery of the main messages.*

General comments:

1. The uncertainty analysis is missing. The authors stated the experimental uncertainty of
   radical destruction rates. But what about the production rates? Was it possible that the ROx
   budget in the SW air was also balanced if uncertainty of P(ROx) was considered. It would be
   useful to add one vertical bar at the time when the discrepancy was largest. The paper
   discussed the possible mechanism behind the experimental budget imbalance, which was
   fine. But the imbalance in the HO2 and RO2 budget could also be explained by
   measurement interreference, for example, the RO2 measurement interference. An
   evaluation of possible interreference and its contribution to close/enlarge the imbalance of
   radical budget should be added in the revised version to complete the discussion.

*A discussion of possible interferences was part of the companion paper. The two papers are
now combined, and so interferences will be more prominent. The uncertainty of the
measurements and the interferences are not sufficient to explain the imbalance of the radical
budgets. We have added the uncertainty in the discussion.*

I would suggest to add the time series of radical budget in supplement at least. So the readers
could see the variability of the experimentally determined budget.

*Many thanks for the suggestion. We will add such a time series for the budgets in the
Supplementary Information.*

The selection of the amounts of X, Y, Z looks quite arbitrary. It would be good to reconstruct the
context to give the reason why such amounts of X, Y, Z were used. The amount of X was
chosen to be 100 ppt as suggested in another paper, which conducted at a very different
environment. The simplification of X mechanism was not really correct as the X species will have
impact on OH, HO2, and RO2 radical budgets. The RO2^OH conversion is possible, such as
RO2 unimolecular isomerization, which should be nominated to a different term other than 'X'.
Although I don't expect a big change to the HO2 and RO2 budget, it's more comparable to
previous papers.

*The amounts are arbitrary, and were chosen as there was some precedent from another paper
as the referee points out. It is just a hypothesis as a possible reason to explain the discrepancy,
and we realise that there are many assumptions and approximations, such as the one
suggested by the referee. We could work out a production rate of OH from RO2 isomerisation
(kisom[RO2]) that closes the OH budget – this would use the total RO2 concentration – and so
the assumption would have to be made that all RO2 radicals make OH. We do not feel there is
previous literature supporting that RO2 to OH would account for all the missing OH, as OH
generation tends to lead to a slowing down on the rate of autoxidation. Also, the modelled to
measured OH is in reasonable agreement, even if there are both missing sources and sinks of
OH which may in some cases cancel, although OH is underpredicted at low NO. On reflection,
and also considering the comment of the other referee on this paper, we have decided to take
out the section in the paper regarding the use of X, Y and Z species being postulated to help to
explain the differences between the model and measurements.*

As the authors proposed that reducing the reaction rate of RO2 to HO2 propagation was the most likely explanation to reconcile the HO2 and RO2 budgets, should the P(Ox) also be calculated with the reduced reaction rate constant?

*The impact on P(Ox) would be simply multiplying the value of P(Ox) by the ratio of the rate of RO2 to HO2 production with and without the reduction in rate coefficient for RO2+NO. The impact of this for the HO2and RO2 budgets was shown in Figure 11. However, we feel that there are already a large number of figures, and hence we will make a statement in the text that the value of P(Ox) will be impacted by a change in the rate coefficient, and refer to Figure 11.*

2. If HO2 uptake and Cl chemistry was not important radical processes, I would suggest moving the relevant figures to the supplement and reduce the context further.

*Yes, that is a good suggestion. We have done this and just retained a sentence or two that uptake and Cl chemistry was done, and the results are in the SI.*

3. The first part of the discussion seems to be a repetition of what have been done and belongs to the conclusion. It should be shortened massively.

*We have done this, and further reduction in material enabling a more concise discussion has been achieved by merging the two companion papers.*

4. The comparison to previous works looks interesting and informative. Could it be able to summarize into a table with the three studies side by side?

*This is a good idea, and we have introduced a Table to summarise the information that is in the text. There is also a Table for the other companion paper which will summarise information for comparison of measured OH, HO2 and RO2 with box-models. This will further aid the reduction in material to make more concise.*

5. The citations are not properly used. For example, Mehra et al. 2020, Slater et al. 2020, Whalley et al., 2020, the discussion version is cited while the final version is published. I also noticed the same problem occurs in the accompany paper. I do not have time go through the list, but I think mistake in the reference is not acceptable in any kind of scientific journals. Please carefully check all the reference list.

*We apologise for this. When the paper was being initially drafted these papers were still at the ACPD stage, but this was not updated to reflect they are now published in ACP, and this will be updated.*

Technical comments:

1. Throughout the paper, it should be the rates of production and destruction. Most of the sentences miss the word 'rate(s)'.

*Thanks for pointing that out, we have added the word "rate(s)"*

2. Line 106. It's not a good argument that H2O2, ROOH and etc. were not measured so they are not considered in the radical budget. I think the contribution of these species is not important. Please give a rough estimate of these species and contribution to the radical budget first and say why they are not included.

*We agree that the contribution of these species is not important. H2O2 and CH3OOH, and some other peroxides have been measured at coastal locations. For example, mean concentrations of H2O2 and CH3OOH at Mace Head were 0.23-1.58 ppbv and 0.1-0.15 ppbv, respectively (Morgan and Jackson 2002). At coastal locations peroxide photolysis was shown to be a minor source of OH or HO2 (via CH3O) (Sommariva et al., 2004; 2006)). Specifically at Cape Grim, the rate of OH production from CH3OOH was less than 5% of the rate of production from O(1D)+H2O (Sommariva et al., 2004), and hence peroxides were not included in the radical budget analyses. This will be added to the revised MS together with these references.*

*Morgan, R. B., and A. V. Jackson, Measurements of gas-phase hydrogen peroxide and methyl hydroperoxide in the coastal environment during the PARFORCE project, J. Geophys. Res., 107(D19), 8109, doi:10.1029/2000JD000257, 2002*

*Sommariva, R., Haggerstone, A.-L., Carpenter, L. J., Carslaw, N., Creasey, D. J., Heard, D. E., Lee, J. D., Lewis, A. C., Pilling, M. J., and Zádor, J.: OH and HO₂ chemistry in clean marine air during SOAPEX-2, Atmos. Chem. Phys., 4, 839–856, https://doi.org/10.5194/acp-4-839-2004, 2004.*

*Sommariva, R., Bloss, W. J., Brough, N., Carslaw, N., Flynn, M., Haggerstone, A.-L., Heard, D. E., Hopkins, J. R., Lee, J. D., Lewis, A. C., McFiggans, G., Monks, P. S., Penkett, S. A., Pilling, M. J., Plane, J. M. C., Read, K. A., Saiz-Lopez, A., Rickard, A. R., and Williams, P. I.: OH and HO₂ chemistry during NAMBLEX: roles of oxygenates, halogen oxides and heterogeneous uptake, Atmos. Chem. Phys., 6, 1135–1153, https://doi.org/10.5194/acp-6-1135-2006, 2006.*

Line 135. 'do not'

*Corrected*

3. Line 212. Could you give an estimation on how much radical can be produced from NO3+VOC?

*Measurements of NO3 were not measured during the campaign and a model calculation of the*

*NO3 reactivity calculation was not performed. So it is difficult to estimate the fraction of radicals which come from NO3 chemistry, but any radicals from this source would reduce the budget imbalance at night.*

Line 231. The number is different from the figure.

*Thanks for spotting this, this has been corrected.*

4. Line 247-250. I don't get the picture why to split the data into two sectors. How significant are these two air masses different? I cannot really see from the radical budget analysis.

*The two papers have now been combined, and the first paper contained a more detailed rationale of why the data were split into two sectors. Weybourne is a well characterised site, and typically experiences two major types of airmass – a cleaner one from the ocean sector, and another which derives from more polluted regions.*

5. Line 258. Be more specific on what is the difference in VOC.

*The difference is VOC composition between the two wind sectors was covered in the companion paper (which is referenced), but as the two paper are now combined it will be clearer what the difference in VOC composition was for the two sectors.*

6. Line 264-266. Better rephrase this sentence to be clearer.

*Again, part of this sentence is summarising what was in the companion paper, and so upon combining the two papers this will no longer be necessary.*

7. Line 302-310. The contribution of different species is better scaled to measured kOH rather than calculated kOH. The same applies to Table 2.

*The missing reactivity is ~ 50% of the measured values, and so scaling to measured kOH would in effect halve the values stated in the text and in Table 2. Rather then changing the Table and the values in the text, we have restated that the calculated reactivity is approximately half that of the measured value – to make it clearer regarding the contribution of each species.*

8. Line 306. 'Small but significant' sounds like a statistic phrasing, which may not fit in the current sentence.

*The words "but significant" have been removed*

9. Line 350. Is there a reason or reference to support this speculation?

   *Not to our knowledge, which is why we use the word speculate.*

10. Line 358. wrong citation format.

*Corrected.*

11. Line 403-404. Better to rephrase it to be clearer.

*The word "gross" has been removed to make this clearer*

12. Line 408. 'HO2+NO' looks confusing.

This has been reworded to *"….for the channel generating $HO_2$ + $NO_2$ products".*

13. Line 411 E13. The RONO2 formation should not be a O3/NO2 loss. I guess the authors are referring to PAN formation, i.e. RO2+NO2?

*If RONO2 is formed then this removed NO2, and hence O3 production via its photolysis.*

14. Line 430 and later on. 'FAGE-calculated' looks like a lab jargon. Maybe better to use derived from measurements or others.

*Thanks, that is a good suggestion. We have replaced FAGE by "derived from measurements using the FAGE instrument"*

15. Line 449-450. Not really needed this sentence.

*We have removed this sentence.*

16. Line 478-481. Better to rephrase it to be clearer.

*This section has been rephrased to: "Reducing the $RO_2$ + NO rate constant by a factor of 5 is not consistent with accepted laboratory measurements for which uncertainties in the range ~15–35% are reported (Orlando and Tyndall, 2012). It is therefore imperative that more laboratory studies are conducted to measure $RO_2$ + NO rate constants with a wide variety of $RO_2$ types."*

17. Line 491. What does the 'complete' mean?

*We have replaced "the lack of (complete) suppression of HOM signals" with "why HOMs species are not completely removed…."*

18. Line 502. How the ozone increase was calculated?

*Using equation E14, we have added this to the text.*

[Figure]

Atmos. Chem. Phys. Discuss., referee comment RC2
https://doi.org/10.5194/acp-2022-213-RC2, 2022
**Comment on acp-2022-213**
Anonymous Referee #2

Referee comment on "Radical chemistry at a UK coastal receptor site - Part 2: experimental radical budgets and ozone production" by Robert Woodward-Massey et al., Atmos. Chem. Phys. Discuss., https://doi.org/10.5194/acp-2022-213-RC2, 2022.

*We thank the referee for their careful reading of the manuscript and for their helpful suggestions. Our responses are given in italics below each comment, together with any changes to the text. The main change is that the two papers have been merged into one paper, and a lot of material has been moved into the SI.*

This manuscript reports an investigation of the ROx radical budget for the ICOZA 2015 field campaign. This is the follow-up of a first publication where the authors reported the measurements of OH, HO2 and RO2 and a comparison to zero-dimensional box modeling. In this companion paper, the authors provide a detailed description of the ROx radical budget (OH, HO2 and RO2 taken all together as a group of species), providing insights into initiation and termination processes of this group of radicals. An original aspect of this publication is that the authors also investigated individual budget closures for OH, HO2 and RO2, providing additional insights into propagation routes within these radicals. Another originality of this work is the use of ancillary measurements of radical sources and sinks to perform an experimental assessment of these radicals budgets.

It is shown that while a reasonable closure of the ROx budget is observed (only a small imbalance between production and destruction rates for air masses from SW origin), individual radical budgets highlight that our knowledge on radical propagation pathways is still incomplete, especially propagation routes between HO2 and RO2.

This reviewer thinks that this work is of interest for the scientific community and deserves publication. Individual HO2 and RO2 budgets are usually not investigated with this level of details and this publication highlights the benefits of assessing these radicals' budgets in addition to the budgets of ROX and OH. I therefore recommend publication in ACP after the authors address the following minor comments:

L89-91 : " Given the short lifetimes of OH, HO2, and RO2 radicals (on the order of seconds to minutes), we can assume that their concentrations are in steady-state and hence expect their production and destruction rates to be equal at a location such as the WAO where incoming air is homogeneous. " - It is not clear to this reviewer whether a lifetime of tens of seconds/minutes is not too long to assume steady-state. This aspect was discussed for ROx modeling in the nocturnal boundary lawer by Geyer et al. (J. Geophys. Res. 109,

doi:10.1029/2003JD004425, 2004). Could the authors comment on this?

*The assumption of steady-state is a central assumption in order to calculate the concentrations of radicals using a box-model. For OH, with a lifetime of < 1 sec, this is always satisfied, and at a clean site where there are no local sources, for example of NOx, which may perturb the NO2/NO ratio from its expected value, we expect this to be the case also for longer-lived radicals such as HO2 or RO2. Actually for these species, minutes is probably not entirely correct, with the levels of NOx encountered at Weybourne, the lifetime is likely to be < 1 minute. We have modified the text to 1 minute. We have also referenced the work of Geyer et al.*

L135-137: "In line with Tan et al. (2019), we did not explicitly consider equilibrium reactions of the type HO2 + NO2 aDD HO2NO2 and RO2 + NO2 aDD RO2NO2 (e.g, peroxyacetyl nitrate (PAN) formation and decomposition) in the budget analyses, and assume these processes result in no net gain or loss of the radical species" - Both of these equilibrium reactions can act as a source or a sink of peroxy radicals, depending on ambient and environmental conditions. What is the range of lifetimes for HO2NO2 and RO2NO2? Could neglecting these reactions lead to significant biases in production and destruction rates of HO2 and RO2?

*The HO2NO2 and CH3O2NO2 destruction and production balance in the model – so represent no net loss or production of peroxy radicals. Only at extremes in temperature would the equilibrium be skewed. For example at Hudson Bay in the Arctic, the formation of HO2NO2 was identified as an important radical reservoir, reducing HOx concentrations during the day and enhancing them at night (Edwards et al, 2011). However, these conditions are not relevant for Weybourne in summer. The acyl peroxy radicals is the only way to form PAN in the MCM, and we state that the acyl peroxy constitutes 7-8% of the RO2 pool. For typical temperatures of the campaign HO2NO2 and PAN (and other PANs) will be in equilibrium. We have made this clearer in the text and added this reference.*

*Edwards, P., et al. (2011), Hydrogen oxide photochemistry in the northern Canadian spring time boundary layer, J. Geophys. Res., 116, D22306, doi:10.1029/2011JD016390.*

Section 2.1.3: The reaction of OH with some VOCs can lead to the prompt formation of HO2 (e.g. isoprene, aromatics). The authors may want to comment on the potential bias introduced in P(HO2) calculations when assuming that VOC+OH reactions only lead to RO2 formation. Same question for P(RO2) in section 2.1.4 - What is the potential bias introduced in P(RO2) calculations?

*The yield of HO2 from OH oxidation of these species is explicitly contained in the MCM mechanism and so HO2 formation is included. For the Beijing field campaign the formation of HO2 from VOC + OH → HO2 versus the formation of RO2 from VOC+OH → RO2 was investigated during the AIRPRO campaign (Whalley et al., 2021) – and HO2 production was not insignificant, owing to the presence of VOCs like isoprene and aromatics in the Beijing in summer. We will modify the wording to make clear that prompt formation of HO2 is included.*

*Whalley, L. K., Slater, E. J., Woodward-Massey, R., Ye, C., Lee, J. D., Squires, F., Hopkins, J. R., Dunmore, R. E., Shaw, M., Hamilton, J. F., Lewis, A. C., Mehra, A., Worrall, S. D., Bacak, A., Bannan, T. J., Coe, H., Percival, C. J., Ouyang, B., Jones, R. L., Crilley, L. R., Kramer, L. J., Bloss, W. J., Vu, T., Kotthaus, S., Grimmond, S., Sun, Y., Xu, W., Yue, S., Ren, L., Acton, W. J. F., Hewitt, C. N., Wang, X., Fu, P., and Heard, D. E.: Evaluating the sensitivity of radical*

*chemistry and ozone formation to ambient VOCs and NO$_x$ in Beijing, Atmos. Chem. Phys., 21, 2125–2147, https://doi.org/10.5194/acp-21-2125-2021, 2021.*

L453-455: "In contrast, model-calculated P(Ox) starts to fall off a little above 1 ppbv NO in NW-SE air, but generally increases with NO in SW air." - For SW air, it seems that the increasing trend stated by the authors is very dependent on one data point at approximately 2 ppb NO. This reviewer thinks that this is a bit overstated.

*We agree. We have modified the text to: "In contrast, model-calculated P(O$_x$) starts to fall off a little above 1 ppbv NO in NW–SE air, but generally increases with NO in SW air, but the latter is largely due to a single point at 2 ppb NO".*

L466-73: The authors discuss the impact of the various recycling hypotheses (HO2+Y, RO2+X, RO2+Z) on the HO2 and RO2 budgets. The discussion focuses on the comparison of median diel profiles of production and destruction rates. Could the authors comment whether the NO-dependence of observed imbalances changes when the proposed recycling processes are accounted for?

*That is a good question, but we feel it is beyond the scope of the paper to further analyse the impact of X, Y and Z on the HO2 and RO2 budgets as a function of NO, given that the identity of X, Y, and Z is not known (we have adopted an approach used by other workers where the concentrations of X, Y and Z are chosen to represent an equivalent NOx concentration). On reflection, and also considering a comment by Referee 1 for this paper, we have decided to omit the section on X, Y, and Z from the paper.*

L566-567: "It is therefore recommended that more studies are conducted to measure RO2 + NO rate constants, in particular for more complex, functionalised RO2." - On the basis of the arguments discussed L536-541 to explain a lower-than-expected RO2-to-HO2 propagation rate, the authors may want to recommend to study the fate of RO radicals.

*A good suggestion, especially for complex RO which may be involved in autoxidation processes. We have added this recommendation.*

---

## Author Response (AR2)

We thank the two referees for their reviews of our revised MS. The Editor has asked us to respond to the comments in Report#1 from Anonymous Reviewer #3.

Below we reproduce the reviewer's comments in black normal type. Our response is given in blue type and changes to the MS are given in red type, unless the change is a major change (e.g. moving of material, removal of redundant material), in which case the revised MS with the tracked changes will indicate the changes that have been made (some minor changes were made once tracked changes were accepted).

**Report #1 (Anonymous referee #3)**

This manuscript describes observed OH, HO2, and RO2 radical concentrations and OH reactivity from the ICOZA campaign at Weybourne, UK, during July 2015 to investigate the degree of our process understanding regarding the atmospheric radicals and ozone formation. As this manuscript already experienced first round of reviews, I basically assumed or found that fundamental issues were mostly resolved. The budget analysis fully constrained by the observed terms is new and interesting. Nonetheless, I still identify two major issues as well as other minor points to consider for potential improvement, as listed below.

1. Consider removing redundant parts to achieve conciseness. I would recommend merging Discussion section to Results to avoid redundancy. For example, discussion about the RO2+NO reaction rate coefficient appears several times; from lines 796, 971, and 1076. Also, some auxiliary parts such as Section 4.3 could be moved to supplementary.

We agree that the combined manuscript can be significantly improved by further (and significant) shortening of the paper via removing redundancy as suggested by the reviewer. We have completely done away with the Discussion Section (Section 4) and renamed Section 3 "Results" as "Results and Discussion", and moved Section 4.3 to the SI as suggested (together with Table 4). In addition, upon moving the Discussion sections to the Results we were able to significantly shorten them (or in fact completely remove some parts). In fact we have gone further with some of the discussion, moving other sections to the SI, for example Section 4.1 and Section 4.2.

Please see the revised MS with the tracked changes showing to see how we have altered the document to achieve this. There is additional material now in the SI.

2. Considering CH3O2 is the main component of RO2 (line 1101, Figure S5), the k_14 rate coefficient of 2.3x10^-11 cm3 molecule-1 s-1 (Line 403) might be too fast; four times faster than that for CH3O2 + HO2 (5.2 x10^-12 cm3 molecule-1 s-1). This might be the main reason why the analyzed budget for RO2 and HO2 is quite open. I believe this is treated adequately in the box model based on MCM and thus the gaps between the observed and modeled radical concentrations are valid; so the main conclusion of the manuscript may remain unchanged. However, many plots (e.g., Figures 9, 10 and 11) and the stated degrees of discrepancy would become different. The necessary shift in the RO2 + NO reaction rate coefficient to explain the observations that the authors propose (Line 747, pointing to section 3.10.1 instead of 3.11?) might become milder. Discussion in the lines 1009-1014 might also be affected.

We agree that using a single value for the RO2+HO2 rate coefficient for the budget analysis is a simplification. The treatment of this rate coefficient in the MCM is indeed more sophisticated. The rate coefficient for reaction of CH3O2+HO2 to form CH3OOH is $4.74 \times 10^{-12}$ cm3 molecule-1 s-1 at 298K in the MCM. $k_{RO2+HO2} = 2.3 \times 20^{-11}$ cm3 molecule-1 s-1 is the generic RO2+HO2 rate coefficient which is used in the MCM for C3 RO2 and above (but this is scaled down by multiplying it by e.g. 0.52 for propane RO2, 0.63 for butane RO2 etc). Reducing $k_{RO2+HO2}$ in the budget analysis would reduce both DRO2 and DHO2, so would improve PRO2 and DRO2 agreement but would worsen agreement between PHO2 and DHO2. It would also likely close the PROx and DROx budget in the afternoon under SW conditions. Only reducing the rate of propagation of RO2 to HO2 can remedy both the PHO2 - DHO2 and PRO2-DRO2 discrepancies simultaneously and this is the focus of the discussion. Yes, by reducing this rate coefficient then the shift in the RO2+NO rate coefficient to explain the discrepancy in the budgets would in turn not need to be as much. We have altered the discussion accordingly in several places, and yes, it should have pointed to section 3.10.1 rather than to 3.11 (apologies, now corrected, although there is now only a Section 3.10).

For example in the Conclusion:

However, if the $RO_2 + HO_2$ rate coefficient were reduced then the reduction in the $RO_2$+NO rate coefficient to explain the discrepancy in the budgets would in turn not need to be by as much.

Minor but important issues:

3. Numbering of the equations and reactions need to be double checked. (E3) appears twice (Lines 293, 422). Line 452: k_4a, Line 456: the HO2+HO2 reaction appeared as R12 before and HO2+RO2 as R14. Line 693: E3 and E6? Line 703: E7 and 8 instead of E5-E6? Line 293: Some reaction rate coefficients are shown in a format as k_HO2+NO, instead of k_4a.

We apologise for equation (E3) appearing twice, this is a legacy of the merger of the two papers and not spotting all of the redundant material. We have checked for numbering of equations and reactions, and where they appear twice under different guises, or are incorrectly numbered, or are formatted inconsistently, we have harmonised in the revised MS. All equations now have $k_N$ rather than by the name of the reaction. Please see the tracked changes document to see the changes in the equations.

4. Line 265: The 800m boundary layer height is kept constant. May it vary over a day? How well were the observed HCHO concentration levels reproduced by the model simulations predicting HCHO, with this boundary layer height assumption? Nighttime radical concentrations might be sensitive to this assumption.

The reviewer is correct in that the boundary layer will vary over a 24 hour period at this site. However, as there were no measurements of this parameter during this study, we have kept it constant. Regarding the model simulations of HCHO, in the SI, we have stated "The MCM-base model performance in simulating these carbonyls was assessed, where it was found that there was reasonable agreement for MVK+MACR on a diel average basis, but that HCHO concentrations were significantly overpredicted in the afternoon (data not shown)." It is possible that the model overprediction of HCHO may be due to an unrealistic boundary layer height in the afternoon, where it is likely be larger than 800m. It is worth noting though that modelled HO2 is still overpredicted in both the base and carb-constrained model, the latter where HCHO is a constraint used by the MCM

and so is not subject to any uncertainty in HCHO caused by an inappropriate boundary layer height used in the model. Nighttime radical concentrations (which in general are very low) might be sensitive to this boundary layer assumption, and we have stated this in the revised MS.

Section 3.4. "The nighttime modelling results might also be sensitive to the choice of boundary layer height, which was kept constant at 800 m in the model."

5. Line 708: HO2 to multi-ppbv is correct? (pptv?)

Yes, multi-ppbv level is correct.

Technical issues:

6. Line 155: The authors replied to a reviewer as 99.8% rather than 99.95%.

Sorry, this has been corrected now to 99.8%.

7. Line 340: Two different instruments were used for HONO observations. Which data did the authors use for the analysis?

We used the measurements of HONO made with the Long path absorption photometry (LOPAP) instrument. We have added a footnote in Table 1 to this effect.

[d] Used to constrain the MCM model

8. Line 360: Though stating all reaction rate constants are from MCM, it seems k_13 and k14 were rather set arbitrarily (line 403).

We address the point about the RO2+HO2 reaction above (R14, which is now R15). For the RO2+RO2 reaction (R13, which is now R14), the MCM uses k(CH3O2+RO2) = 3.5x10$^{-13}$ cm3 molecule-1 s-1 (multiplied by a scaling factor (so at 298K is ~2.2E-13), then the scaling factor in the MCM varies with RO2. For the budget analysis a single value is indeed used, and we have stressed this further by adding to this sentence:

In this budget analysis, $RO_2$ radicals are treated as a single species, with generalised rate coefficients taken from the MCMv3.3.1: at 298 K and 1 atm.....

9. Line 688: Though the authors suggest missing ROx sources, this might be just from a too fast k_14 as mentioned above in the point 2.

We have added the following on this line to reflect this point (discussed above).

Alternatively, the rate constant for $RO_2$+$HO_2$, $k_{15}$, for which a single value is used in the MCM, may be too large for the mix of $RO_2$ present at the WAO.

10. Line 780. Where is the methane contribution to the OH loss?

We apologise that this was left out of the description of the OH loss, and also from Table 3 and Figure 13. A fixed concentration of 1900 ppm was used. The text has been modified as follows:

In terms of organic OH reactivity, methane (~ 10%, 12.5%, a constant mixing ratio of 1900 ppm was used)....

We have also been careful to be clear that missing OH reactivity is for the measured OH reactivity compared with the MCM modelled value, which includes unmeasured intermediates.

Figure 13 has been modified as follows:

[Figure]

**Figure 13.** Median diel profiles of the OH reactivity calculated from measured reactants and comparison to measured OH reactivity, split according to wind direction. Average daytime contributions are given in Table 2. For interpretation of colours, please see the figure legend. Reactants in the "Other" class are listed in Table 2. The shaded area on measured $k'_{OH}$ corresponds to the $1\sigma$ precision of ~1 s$^{-1}$. Model intermediates are not included here but their contributions are discussed in the text.

11. Line 855 and Section 3.11.2. How the relatively large uncertainties associated with the radical observations and with k_14 affect the conclusion here? If the estimated D(ROx) has large uncertainty, I cannot support this section.

We have decided on the basis of this comment, and also to further reduce the amount of material in the paper, to omit the section on the ozone production regime – Ln / Q, the analysis of which could be due to some uncertainty both due to the uncertainty in the radical measurements and in the chosen value of $k_{14}$ (now $k_{15}$).

12. Lines 975-976. Smaller number of cited literature may be enough because this is not the main point of the study.

We agree that the referencing here is rather over-zealous, and so have reduced the number of references. The text now reads as:

The model HO$_2$ was somewhat reduced but the observations could still not be reconciled after inclusion of both HO$_2$ aerosol uptake (using $\gamma_{HO2}$ = 1) and autoxidation chemistry (Bianchi et al.,

2019), which is now known to play a significant role in the gas phase oxidation of both BVOCs (e.g. Zha et al., 2017 and references therein) and anthropogenic VOCs (AVOCs) (Mehra et al., 2020 and references therein).

13. Figure 4, top panels. Can the authors point to the nighttime data for the observation panels too?

We thought about this, but for the observation panels the nighttime data are all clustered closer to the origin and given the density of points it is not possible to discern them easily, and no significant difference in slope is apparent (as we have already stated in the text). So it is not really possible to point to the nighttime data in a useful way. We have added to the caption the same wording as in the text:

For the model results, nighttime data exhibit a different $RO_2$ versus $HO_2$ slope (not observed in the observations).